# Specific CD4+ T cell phenotypes associate with bacterial control in people who 'resist' infection with *Mycobacterium tuberculosis*

Meng Sun [1,15], Jolie M. Phan [2,15], Nathan S. Kieswetter[2,15], Huang Huang[1], Krystle K. Q. Yu[2], Malisa T. Smith[2], Yiran E. Liu[1,3], Chuangqi Wang [4], Sanjana Gupta[1,5], Gerlinde Obermoser[6], Holden Terry Maecker [6], Akshaya Krishnan[6], Sundari Suresh[6], Neha Gupta [1,6], Mary Rieck[6], Peter Acs[6], Mustafa Ghanizada [1,7], Shin-Heng Chiou[8], Purvesh Khatri [1,5,9], W. Henry Boom[10], Thomas R. Hawn[2], Catherine M. Stein [11], Harriet Mayanja-Kizza[12], Mark M. Davis [1,13,14,15] ✉ & Chetan Seshadri [2,15] ✉

A subset of individuals exposed to *Mycobacterium tuberculosis* (*Mtb*) that we refer to as 'resisters' (RSTR) show evidence of IFN-γ⁻ T cell responses to *Mtb*-specific antigens despite serially negative results on clinical testing. Here we found that *Mtb*-specific T cells in RSTR were clonally expanded, confirming the priming of adaptive immune responses following *Mtb* exposure. RSTR CD4+ T cells showed enrichment of T_H17 and regulatory T cell-like functional programs compared to *Mtb*-specific T cells from individuals with latent *Mtb* infection. Using public datasets, we showed that these T_H17 cell-like functional programs were associated with lack of progression to active tuberculosis among South African adolescents with latent *Mtb* infection and with bacterial control in nonhuman primates. Our findings suggested that RSTR may successfully control *Mtb* following exposure and immune priming and established a set of T cell biomarkers to facilitate further study of this clinical phenotype.

Nearly 1.7 billion people who have been exposed to *Mycobacterium tuberculosis* (*Mtb*) are clinically asymptomatic but show positive results on tuberculin skin tests (TST) or interferon (IFN)-γ release assays (IGRA), indicating they may harbor a 'latent' *Mtb* infection (referred to here as LTBI)[1]. We have previously described a cohort of Ugandan household contacts who never developed a positive TST or IGRA, despite a high probability of exposure to *Mtb*. As these individuals did not develop active tuberculosis (TB) over a median of 9.5 years of follow-up, it was proposed that they 'resist' *Mtb* infection[2] (hereafter referred to as RSTR). Highly exposed individuals with negative TST and/or IGRA were also reported among healthcare workers, miners and other household contact cohorts[3], even when considering variations in timing and strength of exposure, duration of follow-up and

frequency of testing[4,5], but whether these individuals control *Mtb* infection is unknown.

Early events following *Mtb* infection are poorly understood, because most animal models seek to recapitulate active disease. Human TB occurs along a spectrum ranging from clinically apparent disease, asymptomatic infection, and possibly resistance to *Mtb* infection[6]. T cells probably mediate resistance to *Mtb* infection, as several studies have conclusively demonstrated their importance in protecting against disease in humans and animal models[2,7,8]. However, the specific role of T cell-derived IFN-γ has been debated. CD4+ T cells can control *Mtb* infection even without production of IFN-γ[9,10]. Although mendelian defects in IFN-γ-related pathway genes confer susceptibility to mycobacterial disease[11], human and mice data suggest that mechanisms

**Fig. 1 | IFN-γ-independent T cell responses are not detected in low-exposure controls. a**, Representative flow cytometry showing the expression of IFN-γ and CD154 in CD4⁺ T cells from TST⁻IGRA⁻ or LTBI in the low-exposure cohort in response to ESAT6/CFP10 stimulation. **b**, COMPASS-generated probability heat map of CD4⁺ T cell subsets including IFN-γ⁻IL-2⁺CD154⁺, IFN-γ⁺IL-2⁻CD154⁺, IFN-γ⁺IL-2⁺CD154⁻ and IFN-γ⁺IL-2⁺CD154⁺ cells from TST⁻IGRA⁻ or LTBI in the low-exposure cohort in response to ESAT6/CFP10. The depth of purple shading correlates to the probability of a participant-specific response above background for a given cell subset. White, no function; black/gray, presence of a function. IFN-γ⁺ cell subsets are noted in gray. **c**, The frequencies of COMPASS-identified CD4⁺ T cell subsets are shown, background corrected by subtracting the frequency of each subset after DMSO stimulation from the frequency after ESAT6/CFP10 stimulation as in **b. d**, The frequencies of polyfunctional CD4⁺ T cells that expressed two or more cytokines in response to ESAT6/CFP10 in CD4⁺ T cells as in **c. e**, Polyfunctionality score of CD4⁺ T cells specific to ESAT6/CFP10. In **c–e**, the statistical significance was calculated using the Wilcoxon rank-sum test. Two-sided P values are shown, while in **c** the reported P values were adjusted for multiple hypothesis testing using the Bonferroni method.

other than IFN-γ may be important for protection[9,12]. We identified *Mtb*-specific IFN-γ- CD4⁺ T cells in both RSTR and LTBI that express interleukin (IL)-2, tumor necrosis factor (TNF) and CD154 in the absence of IFN-γ after *Mtb* exposure[5].

Here, we asked whether *Mtb*-specific CD4⁺ T cells from RSTR expressed unique functional programs compared to LTBI. To address this question, we first analyzed a Ugandan cohort characterized by low *Mtb* transmission and found that *Mtb*-specific IFN-γ⁻CD4⁺ T cells were not detectable in the absence of *Mtb* exposure. Next, we identified several functional programs, including T_H17 cell-like, regulatory T cell and memory T cell phenotypes, that were selectively enriched in *Mtb*-specific CD4⁺ T cells from RSTR compared to LTBI. We also leveraged public datasets to demonstrate an association between these T cell phenotypes and South African adolescents with LTBI who control *Mtb* infection, as well as the protective efficacy of intravenous Bacillus Calmette–Guérin (BCG) in nonhuman primates (NHPs). Taken together, our results suggest that RSTR might control *Mtb* after initial exposure by priming an IFN-γ⁻CD4⁺ T cell response.

## Results

### *Mtb*-specific IFN-γ⁻CD4⁺ T cells are absent in low-exposure individuals

To investigate whether IFN-γ⁻CD4⁺ T cell responses were unique to household contacts, we enrolled a cohort of participants from Kampala, Uganda[5] who had low exposure risk based on community TB transmission rates (hereafter low-exposure cohort; Supplementary Tables 1 and 2)[13]. The low-exposure cohort contained participants

negative for both TST and IGRA (hereafter TST⁻IGRA⁻, *n* = 17, 9 males and 8 females, aged 18–26 years) (Extended Data Fig. 1a) and sex and age matched TST⁺IGRA⁺ individuals (hereafter LTBI, *n* = 19, 10 males and 9 females, aged 18–42 years). To study *Mtb*-specific responses, peripheral blood mononuclear cells (PBMC) from TST⁻IGRA⁻ and LTBI participants were stimulated for 6 h with overlapping peptide pools targeting the *Mtb*-specific proteins ESAT6 and CFP10, which are absent in BCG vaccines (Extended Data Fig. 1b). We used intracellular cytokine staining (ICS) to identify IFN-γ⁺ and IFN-γ⁻CD4⁺ T cell responses and the expression of cytokines IL-2, IL-4/IL-5/IL-13, IL-17A and TNF and we also assessed the expression of the activation and memory markers CD107a and CD154 (Extended Data Fig. 1c and Supplementary Table 3)[14]. Four CD4⁺ T cell functional profiles were detected using the combinatorial polyfunctionality analysis of antigen-specific T cell subsets (referred to as COMPASS)[15], three of which were IFN-γ⁺ (Fig. 1a,b). TST⁻IGRA⁻ participants did not have IFN-γ⁻ or polyfunctional IFN-γ⁺ T cell responses, while LTBI had both (Fig. 1b–e)[5]. Overall, IFN-γ⁻CD4⁺ T cells were absent in low-exposure individuals, indicating IFN-γ-independent T cell responses to ESAT6/CFP10 were not observed outside a high-exposure setting, such as household contacts[5].

### *Mtb*-specific RSTR CD4⁺ T cells have unique phenotypes
We next profiled the *Mtb*-specific CD4⁺ T cells in a cohort of TB household contacts (hereafter household contact cohort) in Kampala, Uganda, which contained 45 individuals who remained TST⁻IGRA⁻ for a median of 9.5 years after exposure[5] (RSTR; 22 males and 23 females, aged 14–67 years) and 45 individuals (23 males and 22 females, age

14–63 years) who were TST[+]IGRA[+] at the initiation of the cohort and did not progress to disease for a median of 9.5 years after exposure[5] (LTBI). Participants were matched for age, sex and exposure risk score (Supplementary Tables 4 and 5). In the discovery phase, we used PBMC collected at a median of 9.5 years after exposure[5] from three RSTR and four LTBI participants (hereafter referred to as the discovery cohort) and performed index sorting and single-cell multiplex targeted transcriptomics on CD69[+]CD154[+] and CD69[+]CD137[+] T cells (hereafter referred to as activated T cells or antigen-specific T cells according to the stimulation antigens) after stimulation of PBMC with ESAT6/CFP10 for 6 h (Extended Data Fig. 1b). This was followed by single-cell whole transcriptomics on clonally expanded activated T cells (the method is hereafter referred to as SELECT-seq[16]; Fig. 2a and Extended Data Fig. 2a). For T cell phenotyping, index sorting also measured the expression of lineage (CD4 and CD8), memory (CD45RA and CD127) and additional activation (CD25, HLA-DR and CD38) markers, while targeted transcriptomics measured the clonotypic T cell receptor (TCR) α and β sequences, the gene expression of canonical transcription factor usage (*TBX21* (T-bet), *RORC* (RORγt), *GATA3*, *FOXP3*, *RUNX1* and *RUNX3*), and characteristic cytokines (*IFNG, IL2, IL17A, GZMB, PERF, IL4, IL5, IL13* and *TGFB*). Using FlowSOM, a tool for clustering and visualizing flow cytometry data, these markers identified 19 cell subsets of ESAT6/CFP10-specific CD4[+] T cells (Fig. 2b)[17]. *t*-Distributed stochastic neighbor embedding (*t*-SNE) visualization indicated that some subsets, such as *RORC*[+]*IL17A*[+] cells (cluster 6), were enriched in RSTR, while some subsets, such as *RORC*[+]*IFNG*[+]*GZMB*[+]*PERF*[+] cells (cluster 11), were abundant in LTBI (Fig. 2c and Extended Data Fig. 2b). We also determined the T cell phenotypes in 16 RSTR (8 males and 8 females, aged 14–39 years) and 16 LTBI (9 males and 7 females, aged 15–46 years) after 12 h stimulation with whole *Mtb* lysate, which contains a broader set of antigens conserved across mycobacteria (Supplementary Tables 4 and 5). Index sorting and targeted transcriptomics did not identify qualitative differences in cluster enrichment in *Mtb* lysate-specific CD4[+] T cell phenotypes between RSTR and LTBI (Extended Data Fig. 2c). Clonal expansion of ESAT6/CFP10-specific CD4[+] T cells was observed in both RSTR and LTBI participants (Fig. 2d,e and Supplementary Tables 6 and 7), which confirmed the IFN-γ[–]CD4[+] T cell recall responses reported previously[5]. Clonally expanded CD4[+] T cells were focused within the *IFNG*[+] T$_H$1[*] subsets in LTBI (Fig. 2f), while they did not dominate in any particular T cell subsets in RSTR (Fig. 2f). These data suggested that RSTR harbored *Mtb*-specific T cell responses with unique functional and phenotypic profiles compared to LTBI.

## RSTR and LTBI *Mtb*-specific CD4[+] T cells have a stem memory T (T$_{SCM}$) cell phenotype

Next, we applied SELECT-seq[16] to deeply profile the clonally expanded ESAT6/CFP10-specific CD4[+] T cells. After sequencing quality control filtering, we analyzed 524 CD4[+] T cells (Extended Data Fig. 3a) and found that the major transcriptional variance was between the RSTR and LTBI group (Extended Data Fig. 3b). Differential gene expression analysis identified 582 genes upregulated in RSTR compared to LTBI and 156 genes upregulated in LTBI compared to RSTR (Methods, Fig. 3a and Supplementary Table 8). The top upregulated genes in LTBI were proinflammatory (*IFNG, TNF* and *IL2*) and cytotoxic (*GZMB* and *PERF*) genes, aligning with an inflammatory response and IFN-γ production identified by Gene Ontology (GO) analysis (Supplementary Tables 9–11). The top upregulated genes in RSTR were linked with T cell activation (*SASH3, TNFRSF4* and *IKZF1*) and cell membrane receptor (*CCR7* and *ITGAL*) (Fig. 3a). GO enrichment in RSTR identified translation, cell–cell adhesion, cell movement, mitochondrial electron transport (Fig. 3b and Supplementary Tables 9–11), a process associated with oxidative phosphorylation and early T cell differentiation[18], as well as T cell activation. Aligning with these, ESAT6/CFP10-specific CD4[+] T cells from RSTR had higher expression of *TCF7, FOXP1* and *CCR7*, suggesting a less differentiated and naive-like phenotype, higher *ITGAL* (encodes LFA1)

and *ITGA1* (VLA1) expression, suggesting higher trafficking mobility and homing to lymph nodes and airways[19,20], and lacked expression of cytokines such as *IFNG, IL2* and *TNF* compared to LTBI (Fig. 3c,d). The enrichment of T cell activation genes and the lack of cytokine production after short-term ESAT6/CFP10 stimulation suggested that RSTR ESAT6/CFP10-specific CD4[+] T cells were likely to be T$_{SCM}$ cells, rather than naive T cells. The T$_{SCM}$ cell phenotype was further supported by the expression of the transcription factor *TCF7* and an enrichment in genes belonging to the Wnt signaling pathway (Fig. 3b,d)[21]. Overall, SELECT-seq indicated that RSTR CD4[+] T cells were less differentiated, had less inflammatory cytokine activities and more cell trafficking mobility and cell–cell adhesions compared to LTBI, suggesting a CD4[+] T$_{SCM}$ cell phenotype.

To validate the CD4[+] T$_{SCM}$ cell phenotype, we assessed 18 additional RSTR (8 males and 10 females, aged 15–39 years) and 20 LTBI (10 males and 10 females, aged 15–63 years) from the household contact cohort who were not included in the discovery cohort (hereafter referred to as the validation cohort; Supplementary Tables 4 and 5). Flow cytometry (17 RSTR and 20 LTBI) on PBMC from the validation cohort stimulated with ESAT6/CFP10 for 12 h or whole *Mtb* lysate for 12 h (Fig. 3e and Extended Data Fig. 4a) indicated a trend toward a higher proportion of naive-like CD45RA[+]CCR7[+]CD4[+] T cells in RSTR ESAT6/CFP10-specific CD4[+] T cells compared to LTBI (16.85% versus 13.21%, *P* = 0.069) (Fig. 3f), while in the same participants, *Mtb* lysate-specific CD4[+] T cells had similar frequencies of naïve-like CD45RA[+]CCR7[+]CD4[+] T cells in RSTR (23.11%) versus LTBI (24.01%) (Extended Data Fig. 4b).

To explore whether the CD45[+]CCR7[+]CD4[+] T cells in RSTR had a naïve or T$_{SCM}$ cell phenotype, PBMC from RSTR (*n* = 12) and LTBI (*n* = 12) from the validation cohort were stained with carboxyfluorescein succinimidyl ester (CFSE) and stimulated with ESAT6/CFP10 or whole *Mtb* lysate overnight and activated naive-like CD69[+]CD154[+]CCR7[+]CD45RA[+]CD45RO[–] lymphocytes were sorted by flow cytometry and cultured for 7 days to assess antigen-specific expansion[22] (Extended Data Fig. 5a–c and Supplementary Table 3). At the time of sorting, we observed a significantly higher proportion of naive-like CCR7[+]CD45RA[+]CD45RO[–] T cells coexpressing the T$_{SCM}$ cell marker CD95 in LTBI compared to RSTR participants, regardless of stimulation (Extended Data Fig. 5d). The frequency of naive-like CD95[+] T cells (hereafter T$_{SCM}$ cells)[22] that proliferated in response to whole *Mtb* lysate or ESAT6/CFP10 was similar in RSTR and LTBI (Extended Data Fig. 5c,e). However, the *Mtb* lysate-specific T$_{SCM}$ cells in both RSTR and LTBI participants preferentially differentiated into CCR7[+]CD45RA[–] central memory T cells, while ESAT6/CFP10-stimulated T$_{SCM}$ cells resulted in a higher frequency of CCR7[–]CD45RA[–] effector memory T cells (Extended Data Fig. 5f,g). Taken together, *Mtb*-specific T$_{SCM}$ cells were found at similar frequencies in RSTR and LTBI, and proliferated and differentiated into various memory phenotypes after antigen re-encounter.

## *Mtb*-specific CD4[+] T cells in RSTR have a T$_{reg}$ phenotype

Next, we investigated the activation phenotypes of the clonally expanded ESAT6/CFP10-specific CD4[+] T cells identified by SELECT-seq. GO analysis on these T cells identified T cell activation pathways, including MAPK signaling, TCR signaling, TNF signaling and noncanonical NF-κB signaling, that were enriched in RSTR compared to LTBI in the household contact cohort (Fig. 4a and Supplementary Tables 9–11). Genes encoding costimulatory molecules such as IL2RB, FAS (CD95), TNFRSF4 (OX40), ICOS, CTLA4 and IKZF1 were significantly upregulated in clonally expanded ESAT6/CFP10-specific CD4[+] T cells in RSTR compared to LTBI (Fig. 4b). To further explore the CD4 signaling interactions in these RSTR cells, we conducted a protein–protein interaction network analysis on 119 immune-related genes that were upregulated in RSTR compared to LTBI. Three network clusters were identified from the genes upregulated in RSTR: a CD4 activation network centered on *CTLA4*, which encodes for a T cell checkpoint inhibitor, an innate immunity network, and a lymphocyte trafficking network (Supplementary

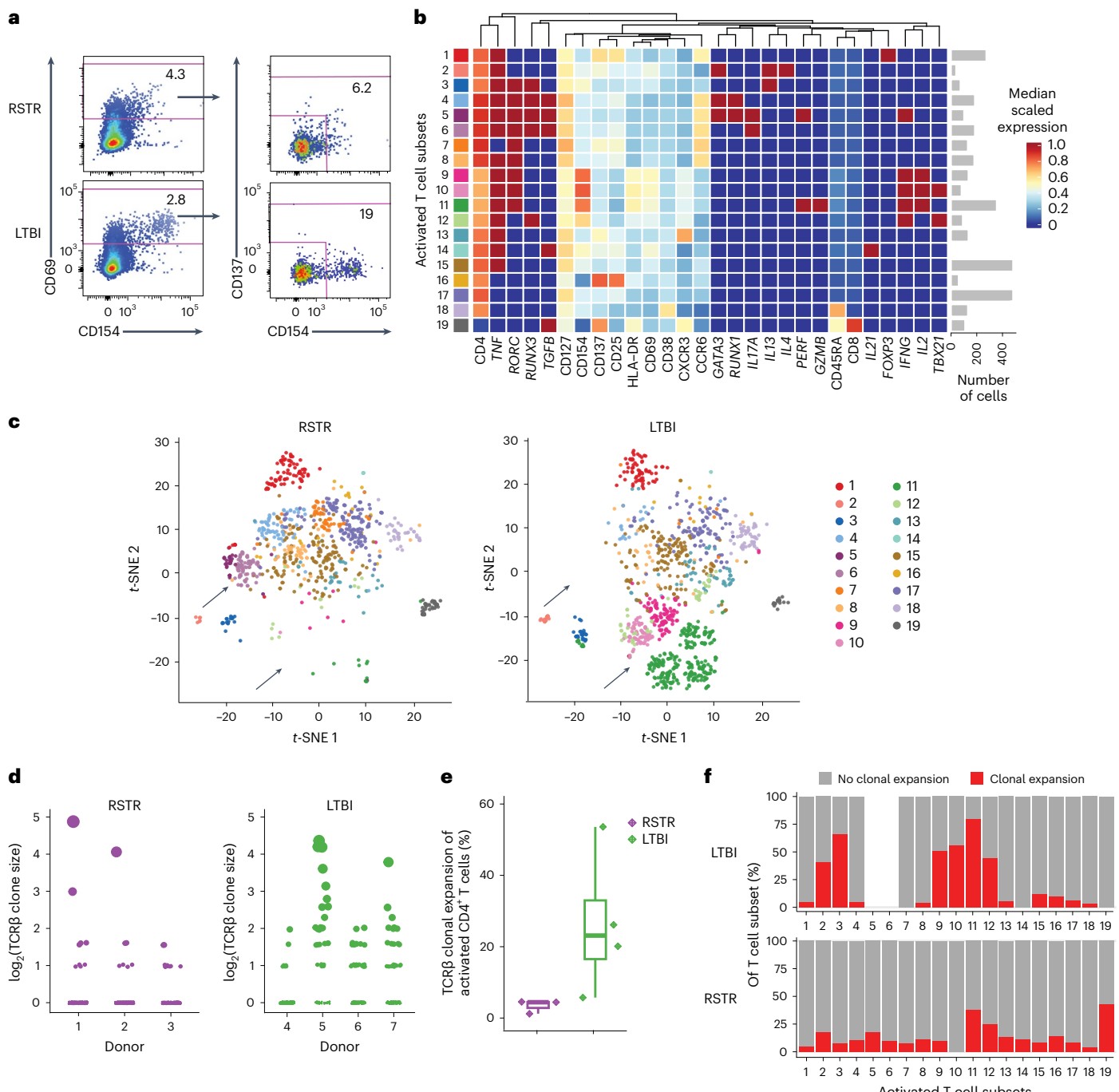

**Fig. 2 | *Mtb*-specific T cells exhibit unique phenotypes and are clonally expanded in RSTR. a**, The gating strategy is shown for the sorting of ESAT6/CFP10-specific T cells from RSTR and LTBI in the discovery household contact cohort, based first on the expression of the activation marker CD69 and then CD154 or CD137. **b**, A heat map showing the median marker expression of ESAT6/CFP10-specific CD4⁺ T cell subsets from three RSTR and four LTBI participants in the discovery household contact cohort. Clustering was performed on flow cytometry mean fluorescence intensities and binarized read counts of profiled genes. **c**, Dimensionality reduction (*t*-SNE) projection of ESAT6/CFP10-specific CD4⁺ T cell subsets, as in **b**. The arrows highlight cluster 6 and cluster 11, which were preferentially detected in RSTR or LTBI participants, respectively. **d**, Distribution of clonal expansion based on TCRβ chain from single-cell targeted transcriptomics in ESAT6/CFP10-specific T cells, as in **b**. Each dot represents a clone as defined by the TCRβ chain CDR3 sequence. The size of the dot is proportional to the frequency, which is also depicted as log₂(counts) on the *y* axis. **e**, A box plot showing the median and interquartile range of frequency of TCRβ clonal expansion in ESAT6/CFP10-specific CD4⁺ T cells with whiskers representing minima and maxima. No statistical test was performed due to the small sample sizes. **f**, A histogram indicating the proportion of clonally expanded cells in ESAT6/CFP10-specific CD4⁺ T cell clusters (clusters 1–19), as in **b**, which were detected more than once in participants among the household contact cohort.

Tables 12–14). The CD4 activation network identified molecules with extensive interactions with other molecules, including *CTLA4*, *IKZF1*, *CD5*, *CCR7*, *IL2RB* and *BATF*, which were defined as key signaling hubs (Fig. 4c and Supplementary Tables 15 and 16). Based on the identification of CTLA4 as a key signaling hub in RSTR, we investigated whether the SELECT-seq clonally expanded ESAT6/CFP10-specific CD4⁺ T cells exhibited a regulatory T (T_reg) cell phenotype. We found higher expression of several T_reg cell-associated genes, including *IKZF2* (Helios)

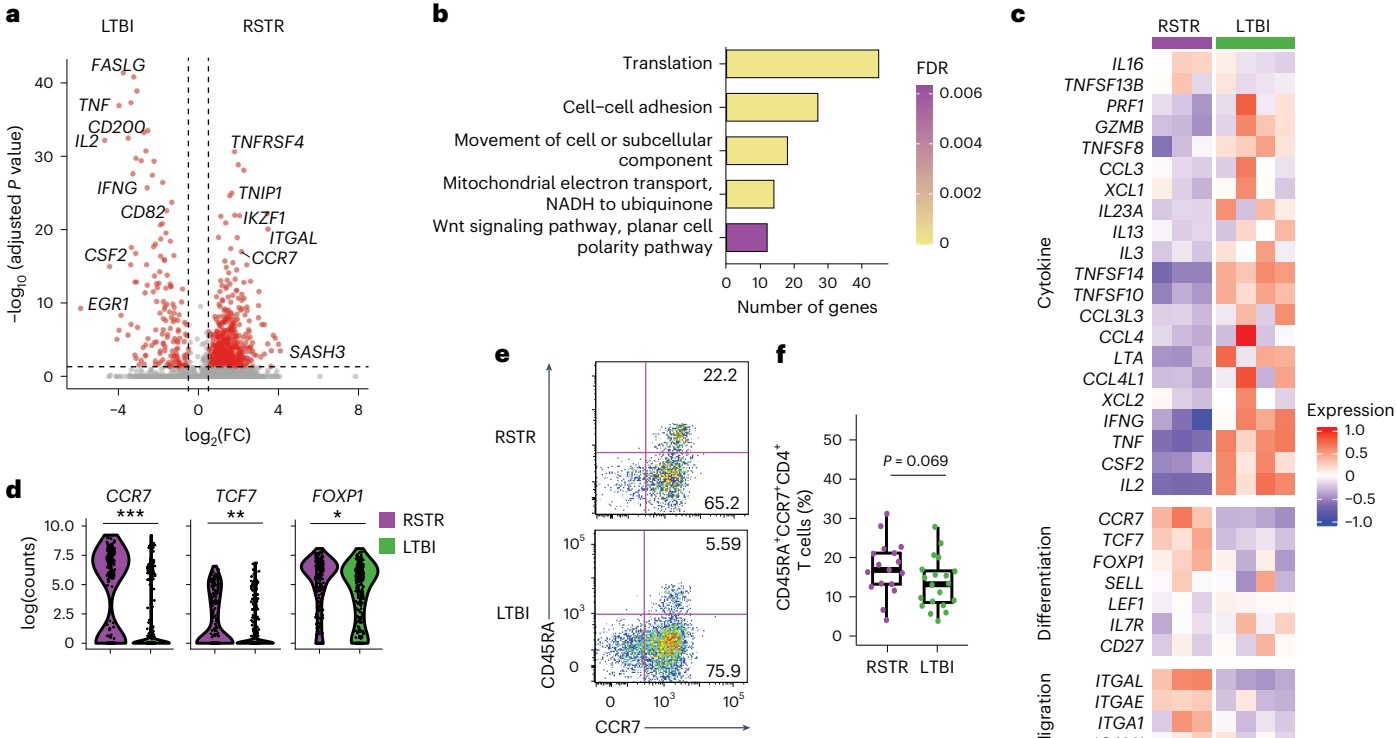

**Fig. 3 | *Mtb*-specific T cells exhibit a T_SCM cell-like phenotype in RSTR and LTBI. a**, A volcano plot depicting DEGs of clonally expanded activated T cells between RSTR (cell number, 231) and LTBI (cell number, 293) from the discovery household contact cohort in response to ESAT6/CFP10 based on SELECT-seq. Red, genes with |log₂(FC)| >0.5 and *P* value <0.05. **b**, GO Biological Process terms related to metabolic and basic activities based on upregulated DEGs from RSTR, as in **a**. False discovery rate (FDR) was calculated by a modified Fisher's exact test with FDR correction. **c**, A heat map displaying the mean expression of genes involved in migration, adhesion or cytokine production in each RSTR and LTBI participant in the discovery household contact cohort. The mean expression level was calculated as the mean of the scaled log-normalized gene counts.

**d**, Violin plots depicting the expression of stem memory like T cell genes (*CCR7*, *FOXP1* and *TCF7*) in RSTR and LTBI, as in **c**. Statistical significance was calculated using two-sided Wilcoxon rank-sum tests with the Bonferroni method. *Adjusted *P* value <0.05, **adjusted *P* value <0.005, ***adjusted *P* value <0.001. **e**, Gating strategy for naive-like CD45RA⁺CCR7⁺ T cells in ESAT6/CFP10-specific CD4⁺ T cells from RSTR (*n* = 17) and LTBI participants (*n* = 20) in the validation household contact cohort. **f**, A box plot showing the median and interquartile range of frequencies of naive-like CD45RA⁺CCR7⁺CD4⁺ T cells in ESAT6/CFP10-specific CD4⁺ T cells, as in **e**, with whiskers representing minima and maxima. Statistical significance was determined by the two-sided Wilcoxon rank-sum test, and the unadjusted *P* value is shown.

and *TNFRSF18* (GITR), in clonally expanded ESAT6/CFP10-specific CD4⁺ T cells from RSTR compared to LTBI (Fig. 4d)[23]. Targeted transcriptomics also suggested a higher frequency of ESAT6/CFP10-specific *FOXP3*⁺ CD25⁺CD4⁺ T cells among RSTR compared to LTBI (Fig. 4e).

To assess whether *Mtb*-specific CD4⁺ T_reg cells were enriched in RSTR compared to LTBI, we performed flow cytometry and multiplex cytokine analysis (ProcartaPlex) on the validation cohort to determine the concentrations of 28 secreted cytokines in conditioned supernatants from cultured PBMC poststimulation with ESAT6/CFP10 or *Mtb* lysate for 6, 12, 24 and 48 h (Extended Data Figs. 7 and 8). The frequency of CD25⁺ cells in total CD4⁺ T cells and ESAT6/CFP10-specific CD4⁺ T cells was higher in RSTR than in LTBI (total, 17.26% versus 14.94%, *P* = 0.028; ESAT6/CFP10-specific, 66.83% versus 63.54%, *P* = 0.065) (Fig. 4f,g and Extended Data Fig. 6a). ESAT6/CFP10-specific CD4⁺ T cells coexpressing protein markers Foxp3 and CD25 were detectable in both RSTR and LTBI at equal frequencies (25.89% versus 24.00%, *P* = 0.424) (Fig. 4f,h). However, in the same participants, we also observed a higher frequency of both CD25⁺CD4⁺ T cells and FoxP3⁺CD25⁺CD4⁺ T cells in RSTR compared to LTBI (CD25⁺, 62.98% versus 57.16%, *P* = 0.052; FoxP3⁺CD25⁺, 19.44% versus 16.13%, *P* = 0.013) after *Mtb* lysate stimulation (Fig. 4g,h). While the frequency of IL-10⁺ cells in ESAT6/CFP10-specific CD4⁺ T cells was comparable between RSTR and LTBI using flow cytometry (0.37% versus 0.33%, *P* = 0.532), the concentration of IL-10 in conditioned supernatants post-ESAT6/CFP10 stimulation was higher in LTBI compared to RSTR (Fig. 4h,i). Within

the same participants, IL-10 concentrations in supernatants after *Mtb* lysate stimulation trended higher in RSTR than LTBI (Fig. 4i). The concentration of TGF-β in supernatants post-ESAT6/CFP10 stimulation was similar between RSTR and LTBI (Extended Data Fig. 6b). Finally, the ratio of anti-inflammatory FoxP3⁺CD25⁺ T_reg cell fraction to pro-inflammatory T cell fraction, which included RORγt⁺T-bet⁺ T_H1* cells, RORγt⁺T-bet⁻ T_H17 cells and RORγt⁻T-bet⁺ T_H1 cells, was significantly higher in RSTR than LTBI (Extended Data Fig. 6c). Together, these data indicated that RSTR *Mtb*-specific T cells exhibited a distinct activation profile, enriched for a T_reg cell phenotype, compared to LTBI.

**Mtb-specific RSTR CD4⁺ T cells have a T_H17 cell-like phenotype**
Next, we investigated the polarization of ESAT6/CFP10-specific CD4⁺ T cell functions in PBMC from RSTR from the discovery household contact cohort. Targeted transcriptomics revealed that *RORC*⁺*TBX21*⁺ T_H1* or *RORC*⁻*TBX21*⁺ T_H1 CD4⁺ T cells were more abundant in ESAT6/CFP10-specific CD4⁺ T cells among LTBI than RSTR (Fig. 5a), which is consistent with known LTBI phenotypes[24], while *RORC*⁺*TBX21*⁻ T_H17 CD4⁺ T cells were enriched in RSTR (Fig. 5a and Extended Data Fig. 3c). SELECT-seq showed increased expression of several T_H17 cell-associated genes, including *BATF*, *RORA* and *STAT3*, in clonally expanded ESAT6/CFP10-specific CD4⁺ T cells from RSTR compared to LTBI (Fig. 5b,c; GO:0072539 from MSigDB database)[25,26], while the expression of T_H1 and T_H1* cell-associated genes *IL2*, *TBX21* and *CXCR3* was increased in LTBI (Fig. 5b,c).

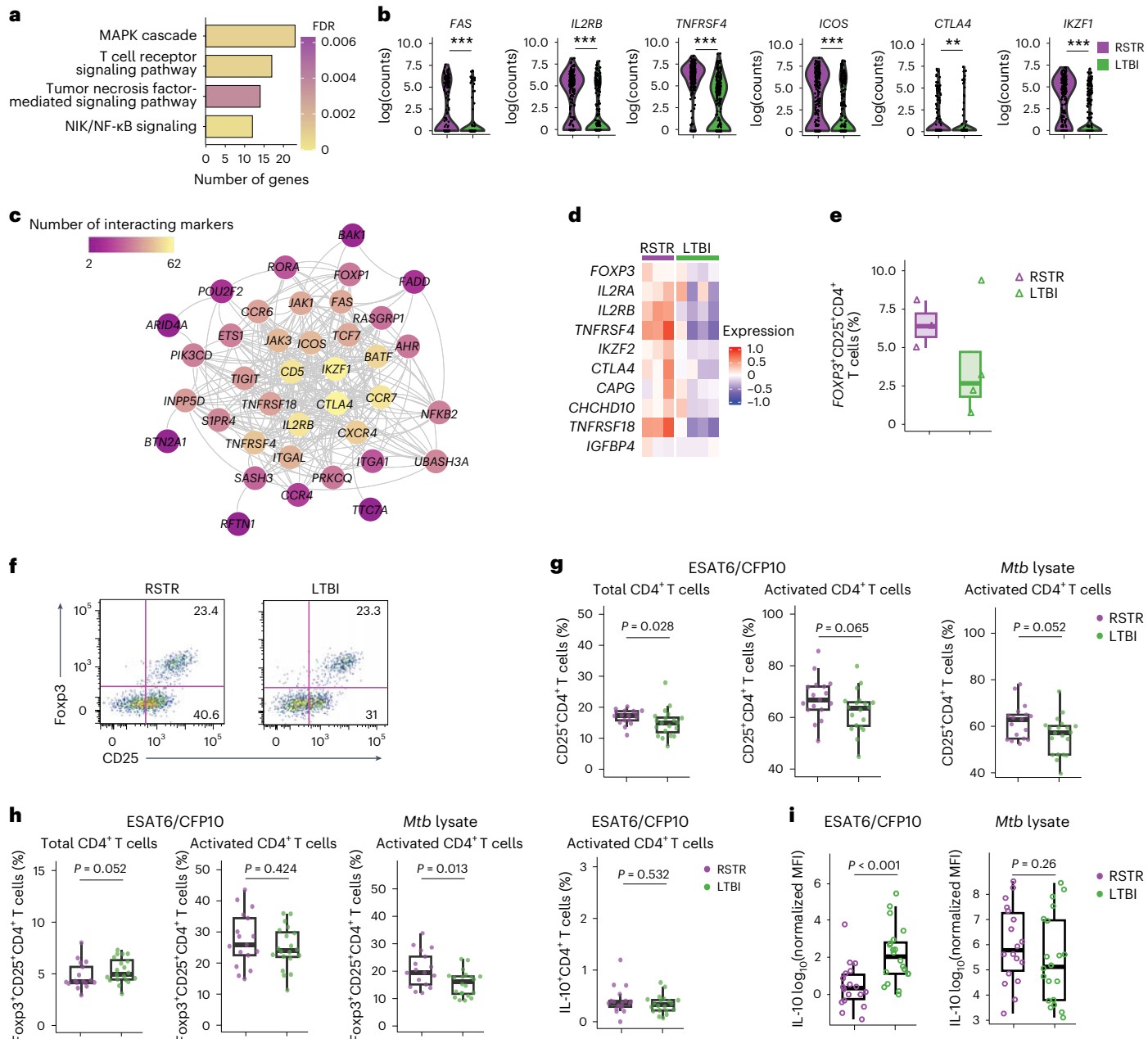

**Fig. 4 | *Mtb*-specific T cells exhibit distinct activation phenotypes in RSTR compared to LTBI. a**, GO terms related to T cell activation in clonally expanded ESAT6/CFP10-specific CD4[+] cells among RSTR (cell number, 231) compared to LTBI (cell number, 293) from the discovery household contact based on SELECT-seq. **b**, Expressions of activation and costimulation genes in RSTR and LTBI, as in **a**. Statistical significance was calculated by two-sided Wilcoxon rank-sum tests with the Bonferroni method. \*\*Adjusted $P$ value <0.005, \*\*\*adjusted $P$ value <0.001. **c**, Interaction network of DEGs related to T cell activation upregulated in RSTR compared to LTBI, as in **a**. **d**, Mean expression of $T_{reg}$ cell-associated genes within each RSTR and LTBI, as in **a**. **e**, A box plot showing the median and interquartile range of frequencies of *FOXP3*[+]CD25[+] $T_{reg}$ cells in ESAT6/CFP10-specific CD4[+] T cells from the same RSTR ($n$ = 3) and LTBI

($n$ = 4) participants as in **a** using index sorting and targeted transcriptomics. The whiskers represent minima and maxima. No statistical test was performed due to small sample sizes. **f**, Gating strategy for CD25[+] T cells and Foxp3[+]CD25[+] $T_{reg}$ cells in ESAT6/CFP10-specific CD4[+] T cells from RSTR ($n$ = 17) and LTBI participants ($n$ = 20) in the validation household contact cohort. **g**, Frequencies of CD25[+]CD4[+] T cells in ESAT6/CFP10-specific CD4[+] T cells are shown as in **f**. **h**, Frequencies of Foxp3[+]CD25[+] $T_{reg}$ and IL-10[+] CD4[+] T cells in ESAT6/CFP10-specific CD4[+] T cells are shown as in **f**. **i**, MFIs of IL-10 in supernatants after 48 h ESAT6/CFP10 or 12 h *Mtb* lysate stimulation based on multiplex cytokine analysis. Significance was determined by two-sided Student's *t*-test. Significance in **g** and **h** was determined by two-sided Wilcoxon rank-sum tests.

To test whether *Mtb*-specific $T_H$17-like cells were enriched in RSTR compared to LTBI, we performed flow cytometry and multiplex cytokine analysis on the validation cohort. After ESAT6/CFP10 stimulation, we found an increased frequency of RORγt[+]T-bet[+] or CXCR3[+]CCR6[+] $T_H$1\* and RORγt[−]T-bet[+] $T_H$1 cells in LTBI *Mtb*-specific CD4[+] T cells compared to RSTR (RORγt[+]T-bet[+], 15.6% versus 12.9%, $P$ = 0.020; CXCR3[+]CCR6[+], 16.7% versus 12.3%, $P$ < 0.001;

and RORγt[−]T-bet[+], 41.9% versus 30.22%, $P$ < 0.001) compared to RSTR (Fig. 5d,e and Extended Data Fig. 9a,b)[24]. Conversely, we detected an increased frequency of RORγt[+]T-bet[−] $T_H$17 T cells in RSTR compared to LTBI (5.86% versus 2.36%, $P$ < 0.001) (Fig. 5e). Similarly, after *Mtb* lysate stimulation, the frequency of RORγt[+]T-bet[+] or CXCR3[+]CCR6[+] $T_H$1\*CD4[+] T cells was higher in LTBI (RORγt[+]T-bet[+], 20.3% versus 15.3%, $P$ = 0.039 and CXCR3[+]CCR6[+], 20.2% versus 13.5%, $P$ = 0.003), while the frequency

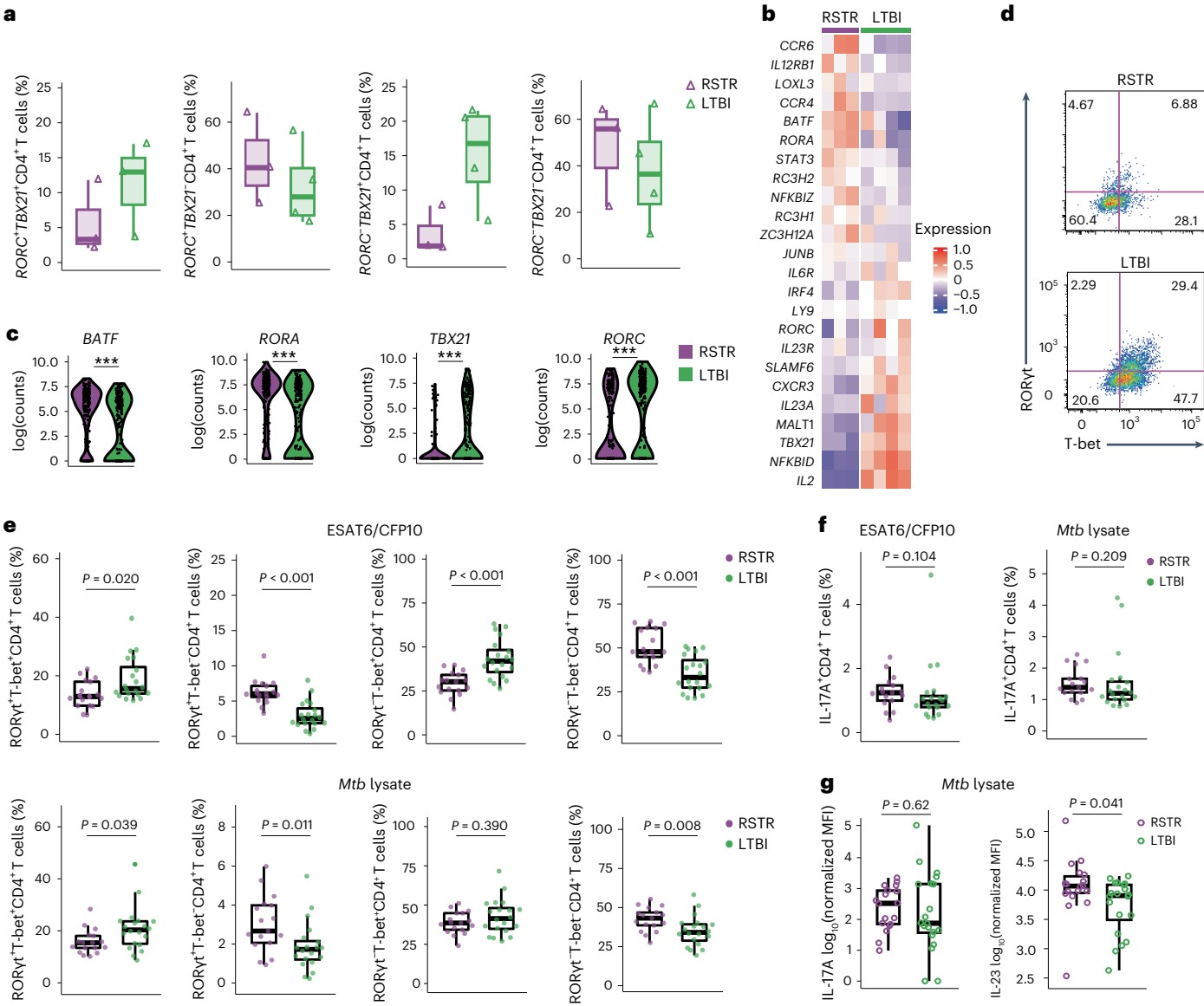

**Fig. 5 | Enrichment of *Mtb*-specific T_H17-like cells among RSTR. a**, Box plots showing the median and interquartile range of frequencies of *RORC⁺TBX21⁺*, *RORC⁺TBX21⁻*, *RORC⁻TBX21⁺* and *RORC⁻TBX21⁻* cells in ESAT6/CFP10-specific CD4⁺ T cells in RSTR (*n* = 3) and LTBI (*n* = 4) in the discovery household contact cohort using targeted transcriptomics. The whiskers represent minima and maxima. Statistical testing was not performed due to small sample sizes. **b**, SELECT-seq showing the mean expression of genes associated with T_H1 or T_H17 phenotypes from GO:0072539 in the MSigDB database in clonally expanded ESAT6/CFP10-specific CD4⁺ T cells within both RSTR and LTBI in the household contact cohort as in **a**. **c**, Expressions of T_H1 or T_H17 cell-associated genes in RSTR and LTBI in the household contact cohort, as shown in **a**. Significance was determined using two-sided Wilcoxon rank-sum tests with the Bonferroni method. ***Adjusted *P* value <0.001. **d**, Representative flow cytometry showing the expression of RORγt and T-bet in ESAT6/CFP10-specific CD4⁺ T cells from RSTR (*n* = 17) and LTBI participants (*n* = 20) in the validation household contact cohort. **e**, Frequencies of RORγ⁺T-bet⁺, RORγt⁺T-bet⁻, RORγt⁻T-bet⁺ and RORγt⁻T-bet⁻ cells in ESAT6/CFP10-specific CD4⁺ T cells or *Mtb* lysate-specific CD4⁺ T cells from RSTRs and LTBI, as in **d**. **f**, Frequencies of IL-17A⁺ cells in ESAT6/CFP10-specific CD4⁺ T cells or *Mtb* lysate-specific CD4⁺ T cells from RSTRs and LTBI as in **d**. **g**, MFIs of IL-17A or IL-23 in supernatants of PBMC are shown after 24 h or 6 h stimulation, respectively, with *Mtb* lysate. Significance was determined by two-sided Student's *t*-test. Significance in **e** and **f** was determined by two-sided Wilcoxon rank-sum tests.

of RORγt⁺T-bet⁻ T_H17 CD4⁺ T cells was higher in RSTR (2.66% versus 1.72%, *P* = 0.011) (Fig. 5e and Extended Data Fig. 9b). ICS and multiplex cytokine analysis found higher production of IFN-γ in PBMC from LTBI than RSTR after both stimulations (Extended Data Fig. 7, 8 and 9b). IL-17A⁺ cells trended higher in RSTR *Mtb*-specific CD4⁺ T cells compared to LTBI after both stimulations, although this was not statistically significant (ESAT6/CFP10, 1.24% versus 0.95%, *P* = 0.104 and *Mtb* lysate, 1.39% versus 1.20%, *P* = 0.209) (Fig. 5f). Multiplex cytokine analysis showed that expression of IL-17A increased in both RSTR and LTBI after ESAT6/CFP10 and *Mtb* lysate stimulations (Fig. 5g and Extended Data Fig. 7). We also noted higher amounts of the T_H17 cell-promoting cytokine IL-23 in RSTR conditioned supernatants post-*Mtb* lysate stimulation than LTBI (Fig. 5g). Thus, RSTR *Mtb*-specific T cells were biased toward a T_H17 cell-like functional program, rather than the T_H1* cell program observed in LTBI.

## *Mtb*-specific RSTR CD4⁺ T cell phenotypes associate with bacterial control

To further contextualize our findings, we analyzed public data from cohorts with clinically relevant endpoints. We leveraged data from the Adolescent Cohort Study (ACS), a longitudinal study of LTBI adolescents in South Africa, in which blood was collected every 6 months

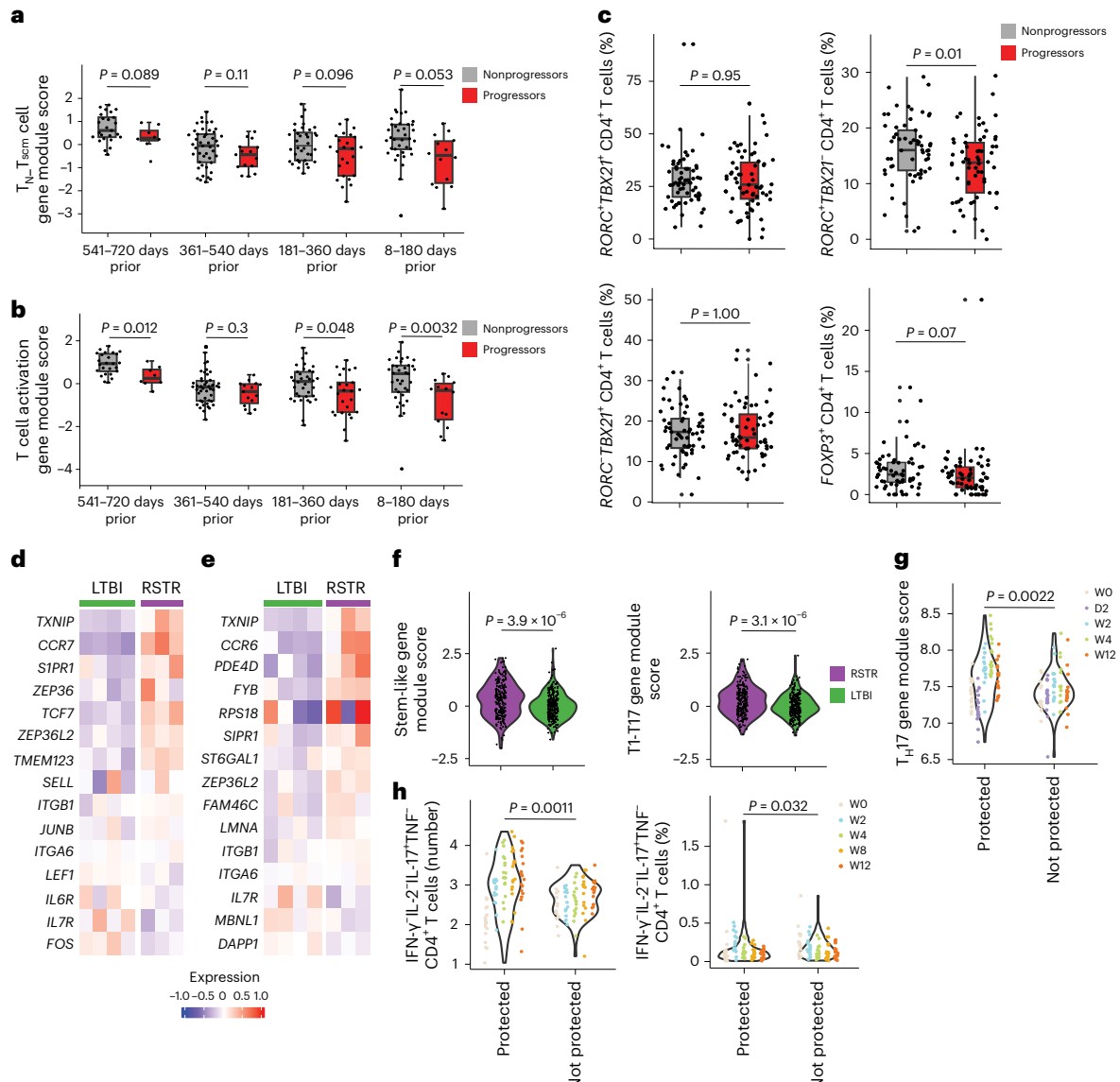

**Fig. 6 | RSTR CD4+ T cell phenotypes associate with lack of progression to active TB in the ACS cohort and bacterial control in NHPs. a,b,** Box plots showing the median and interquartile range of RSTR-associated $T_N$_$T_{SCM}$ cell module (**a**) or T cell activation module scores (**b**) from whole-blood bulk transcriptomics at serial time points before TB diagnosis among nonprogressors ($n = 101$) and progressors ($n = 43$) in the ACS (GSE79362) (ref. 28). The whiskers represent minima and maxima. Scores were computed as geometric mean of gene expressions. Significance was determined by two-sided Student's $t$-test. **c,** Frequencies of *Mtb*-specific CD4+ T cells expressing *RORC* and/or *TBX21* or *FOXP3* among TB nonprogressors ($n = 35$) and progressors ($n = 35$) in ACS[31] using single-cell targeted transcriptomics. Significance was determined by two-sided Wilcoxon rank-sum tests. **d,e,** Heat maps showing the mean expression of the

top 15 enriched genes in a stem-like T cell subset (**d**) and a T1–T17 cell subset (**e**) identified in granulomas from cynomolgus macaques in clonally expanded ESAT6/CFP10-specific CD4+ T cells in RSTR (cell number, 231) and LTBI (cell number, 293) in the household contact cohort. **f,** Stem-like T cell- and T1–T17 T cell-associated gene module scores as in **d**. **g,** RSTR-associated $T_H$17 cell gene module score from whole-blood bulk transcriptomics in rhesus macaques at day 0 (W0), day 2 (D2), week 2 (W2), week 4 (W4) and week 12 (W12), postintravenous vaccination with BCG. **h,** Flow cytometry showing cell count and frequency of IFNγ-IL-2-IL-17+TNF- $T_H$17 CD4+ cells in the bronchoalveolar lavage of BCG-vaccinated rhesus macaques as in **g**. For **c** and **f–h**, significance was calculated using the two-sided Wilcoxon rank-sum tests.

over 2 years to identify blood correlates of TB risk[27,28]. To determine whether the T cell transcriptional programs identified in RSTR were associated with a lack of progression to active TB disease, we studied 144 ACS participants who had whole-blood transcriptomic data, among which 43 participants progressed to microbiologically confirmed TB (progressors) and 101 did not develop TB (nonprogressors), and calculated gene module scores across all time points as the mean expression level of functional associated genes. A '$T_N$_$T_{SCM}$ module score' composed of seven genes related to the naive-like ($T_N$) or $T_{SCM}$ cell phenotype we describe here, including *CCR7* and *TCF7* (Methods)[29], was higher in

nonprogressors than progressors up to 180 days before the onset of TB disease (Fig. 6a). There was a trend toward a higher score in nonprogressors compared to progressors as far back as 2 years before disease onset (Fig. 6a). A 'T cell activation module score', composed of 24 genes identified through network analysis[30] in clonally expanded ESAT6/CFP10-specific CD4+ T cells among RSTR (Methods, Supplementary Tables 15 and 16 and Supplementary Fig. 4c), including *CTLA4*, *IL2RB* and *IKZF1*, was enriched in nonprogressors compared to progressors up to 180 days before disease onset (Fig. 6b). This association was also significant up to 2 years before disease onset (Fig. 6b). Thus, we

found an enrichment of naive-like and T cell activation transcriptional programs among a subset of South African adolescents with LTBI who did not progress to active TB disease compared to those who did progress.

As the associations within whole-blood transcriptomes did not necessarily implicate T cells directly, we analyzed the frequency of CD4[+] T cell subsets using publicly available targeted transcriptomic data[31] from the ACS cohort in response to stimulation with whole *Mtb* lysate (35 progressors and 35 nonprogressors, with all time points collapsed). The frequency of *RORC*[+]*TBX21*[+] $T_H1^*$ or *RORC*[−]*TBX21*[+] $T_H1$ cells in *Mtb*-specific CD4[+] T cells was similar between progressors and nonprogressors (Fig. 6c). However, the frequencies of *RORC*[+]*TBX21*[−] $T_H17$ and *FOXP3*[+] $T_{reg}$ were higher in nonprogressors than progressors ($T_H17$, 16.0% versus 13.3%, $P = 0.01$ and $T_{reg}$, 3.01% versus 2.40%, $P = 0.07$) (Fig. 6c), indicating that RSTR-defined $T_{reg}$ cell-like and $T_H17$ cell-like phenotypes were enriched in South African adolescents with LTBI who did not progress to active TB disease.

Finally, we sought to examine whether RSTR CD4[+] T cell phenotypes were associated with bacterial control by analyzing published NHP studies in which the *Mtb* burden was quantified after natural infection or experimental vaccination. We first leveraged a study of 26 granulomas from four cynomolgus macaques assessed at 10 weeks after low-dose infection with *Mtb* (<10 colony forming units (c.f.u.)) using single-cell RNA sequencing (RNA-seq)[32] in which several T cell clusters, including *GZMB*[+]*S100A10*[+] cytotoxic, *MKI67*[+] proliferating, *CCR7*[+]*TCF7*[+] stem-like and *TXNIP*[+]*CCR6*[+] T1−T17 cell populations, were associated with reduced bacterial burden[32]. We found higher expression of the top genes in the macaque stem-like cell subset (such as *CCR7* and *TCF7*; Fig. 6d) and T1−T17 population 1 subset genes (such as *CCR6* and *TXNIP*; Fig. 6e) in clonally expanded activated CD4[+] T cells in RSTR compared to LTBI. The associated gene module scores composed of the top 15 genes from each subset were significantly higher in RSTR than LTBI (Methods and Fig. 6f). We also examined the blood transcriptomes and T cell phenotypes in 34 rhesus macaques that were vaccinated intravenously with BCG, challenged with *Mtb* 24 weeks postvaccination, and assessed for *Mtb* burden by necropsy at 36 weeks postvaccination or upon development of humane endpoint clinical signs (c.f.u. <100 designated as protected and c.f.u. >100 nonprotected) to identify correlates of protective immunity[33,34]. Expression of a $T_H17$ cell gene module based on genes enriched in clonally expanded activated CD4[+] T cells from RSTR in Fig. 5b (*CCR6*, *RORA*, *CCR4* and *BATF*) was higher in protected macaques than nonprotected ones (Methods and Fig. 6g). Similarly, the absolute number, but not the frequency of purified protein derivative (PPD)-specific IFN-γ[−]IL-2[−]TNF[−]IL-17[+] CD4[+] T cells in bronchoalveolar lavage was higher in protected macaques than nonprotected ones (Fig. 6h). As such, $T_H17$ cell-like functional programs that were enriched in *Mtb*-specific T cells in RSTR were also associated with the control of *Mtb* infection in NHP models.

## Discussion

Here, we showed that *Mtb*-specific IFN-γ[−] T cell responses to the *Mtb* antigens ESAT6/CFP10 were not detectable in a low-exposure cohort in Uganda, but were clonally expanded in individuals in a household contact cohort who 'resist' *Mtb* infection, indicating the IFN-γ[−] T cell functional profile may be a reliable measure of *Mtb* exposure and adaptive immune priming. RSTR *Mtb*-specific T cells expressed $T_{reg}$ cell and $T_H17$ cell-like phenotypes compared to LTBI, who were characterized by $T_H1$ and $T_H1^*$ phenotypes. $T_H17$-like and $T_{reg}$ cell transcriptional programs were also observed in a publicly available cohort of adolescent LTBI nonprogressors, suggesting an association with protection along the clinical spectrum of human TB. Finally, The $T_H17$ cell-like functional programs enriched in RSTRs were also associated with early *Mtb* control in natural and experimental infections in NHP models. Our findings suggest that RSTR may control *Mtb* after initial exposure and define a set of T cell biomarkers that could be used to identify RSTR in other populations after high-intensity exposure to *Mtb*.

Our data support an important role for *Mtb*-specific T cells expressing IL-17 in the absence of IFN-γ. Studies in mice that received an ESAT6 subunit vaccine indicated a role for IL-17A-producing CD4[+] T cells in recruiting $T_H1$ cells, which directly restricted bacterial growth[35]. Consistent with this, *RORC* loss-of-function mutations in humans result in impaired IFN-γ response to mycobacteria[36]. A longitudinal study of *Mtb*-exposed household contacts in Peru found a reduction in $T_H17$-like effector cells in individuals who previously developed active TB disease compared to those who did not[37]. In a repeated limiting-dose challenge model in NHPs, mucosal $T_H17$ cells correlated with protective immunity[38]. Group 3 innate lymphoid cells mediated protection against *Mtb* through IL-17 and IL-22 induction of CXCL13 and the formation of lymphoid follicles within granulomas[39]. Here, we found an expansion of $T_H17$ cell-like transcriptional program in LTBI adolescents who did not to progress to active TB over 2 years of observation compared to progressors in a South African cohort. These data support the notion that LTBI is heterogeneous and consists of individuals at risk for and individuals protected from *Mtb* disease. Future work may help replace the terms RSTR and LTBI, which are defined by IFN-γ[+] immunity, with more clinically informative definitions.

Our results are not inconsistent with a model in which *Mtb* is recognized and eliminated without the assistance of T cells at the earliest stages of exposure. Alveolar macrophages are among the first airway immune cells to encounter *Mtb* and are generally permissive to *Mtb* growth[40]. Studies of blood-derived myeloid cells have revealed differences in the transcriptional response between RSTR and LTBIs in household contacts in Uganda[41,42] and a cohort of miners in South Africa[43]. A study of *Mtb*-exposed household contacts in Indonesia reported higher IL-6 concentrations after in vitro stimulation with *Escherichia coli* among those who remained persistently IGRA[−] compared to those who became IGRA[+][44]. Notably, IL-6 and TGF-β are required to prime $T_H17$ responses[45]. These data support a model whereby differences in the early innate immune response to *Mtb* infection might result in differences in T cell priming. Antibodies also contribute to bacterial control through a variety of mechanisms, including activating antibacterial programs within macrophages[46]. Monoclonal antibodies derived from TST[−] *Mtb*-exposed healthcare workers confer protection against *Mtb* challenge in mice[47]. Thus, non-T cell mechanisms may act in concert to reduce the bacterial load and tune the inflammatory environment to prime the T cell phenotypes that we observed.

*Mtb*-specific IFN-γ[+] CD4 T cells may be a reliable proxy for established infection with *Mtb*. The concentration of IFN-γ in IGRA supernatants is associated with progression to active TB[48,49]. The frequency of *Mtb*-specific IFN-γ[+] CD4 T cells increases along the spectrum of IGRA nonconverters (persistently IGRA[−]), reverters (previously IGRA[+], but now IGRA[−]) and persistently IGRA[+] South African adolescents[50]. Our study characterized the phenotypes of T cells targeting ESAT6/CFP10, which are specific for *Mtb* and incorporated into IGRA and whole *Mtb* lysates, which contain many more antigens that are shared across many mycobacterial species. However, we did not identify the additional antigens present in *Mtb* lysate driving this T cell response. In a study of *Mtb*-infected mice and BCG-vaccinated humans, *Mtb* infection drove ESAT6-specific T cells to become more differentiated than T cells specific for Ag85B, consistent with the observation that *Mtb* restricts expression of Ag85B, but not ESAT6, during chronic infection[51–53]. The LTBI nonprogressors that we studied here were shown to preferentially target PE13 and CFP10 when compared to progressors[31]. Similar efforts may define antigens present in Mtb lysate that are preferentially targeted by RSTR compared to LTBI.

## Online content

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

[1]Institute for Immunity, Transplantation and Infection, School of Medicine, Stanford University, Stanford, CA, USA. [2]Department of Medicine, University of Washington School of Medicine, Seattle, WA, USA. [3]Department of Epidemiology and Population Health, School of Medicine, Stanford University, Stanford, CA, USA. [4]Department of Immunology and Microbiology, University of Colorado, Anschutz Medicine Campus, Aurora, CO, USA. [5]Center for Biomedical Informatics Research, Department of Medicine, Stanford University, Stanford, CA, USA. [6]Human Immune Monitoring Center, Institute for Immunity, Transplantation and Infection, School of Medicine, Stanford University, Stanford, CA, USA. [7]Department of Immunology and Microbiology, Faculty of Health and Medical Sciences, University of Copenhagen, Copenhagen, Denmark. [8]Division of Medical Oncology, Rutgers Cancer Institute of New Jersey, Department of Medicine, Rutgers Robert Wood Johnson Medical School, Rutgers University, New Brunswick, NJ, USA. [9]Department of Biomedical Data Sciences, School of Medicine, Stanford University, Stanford, CA, USA. [10]Department of Medicine, Case Western Reserve University, Cleveland, OH, USA. [11]Department of Population and Quantitative Health Sciences, Case Western Reserve University, Cleveland, OH, USA. [12]Department of Medicine, Makerere University School of Medicine, Kampala, Uganda. [13]Department of Microbiology and Immunology, Stanford University School of Medicine, Stanford, CA, USA. [14]Howard Hughes Medical Institute, Stanford University School of Medicine, Stanford, CA, USA. [15]These authors contributed equally: Meng Sun, Jolie M. Phan, Nathan S. Kieswetter, Mark M. Davis, Chetan Seshadri. ✉e-mail: mmdavis@stanford.edu; seshadri@uw.edu

## Methods

### Low-exposure cohort

The low-exposure cohort was enrolled from a low TB incidence district that was identified by the Kampala Capital City Authority based on low TB transmission rates[13]. Subjects were screened and enrolled for health assessment and blood draws between 2017 and 2018 in Uganda (Extended Data Fig. 1). All healthy, nonpregnant participants were eligible. A total of 247 individuals were approached in this district, of which 230 consented and were screened for previous TB treatment, pregnancy, medications and serious illnesses. A total of 220 healthy, nonpregnant individuals were enrolled, and 211 of the enrolled participants were found to be noninfected with human immundeficiency virus (HIV). All participants reported no known contact with a TB case. Based on TST and IGRA (QuantiFERON-TB (QFT) Gold) concordance, we were able to classify 196 HIV-noninfected individuals as either TST+IGRA+ concordant, TST+IGRA− discordant, TST−IGRA− concordant or TST−IGRA+ discordant. For the present study, we selected 17 TST−IGRA− subjects (mean age of 21.59 years; 9 males and 8 females) and 19 TST+IGRA+ (LTBI) subjects (mean age of 24.47 years; 10 males and 9 females) after matching for age and sex.

### Household contact cohort

The full details of this cohort have been previously published[54]. Initially, household contacts of sputum culture-positive cases of pulmonary TB were enrolled between 2002 and 2012 as part of the Kawempe Community Health Study. At baseline, individuals who had no active *Mtb* infection determined by sputum culture and radiology were enrolled. Upon enrollment, individuals were longitudinally screened during a 2-year follow-up period by TST (Mantoux method, 0.1 ml of 5 tuberculin units of PPD, Tubersol; Connaught Laboratories), in which a positive TST was defined as an induration of >10 mm for individuals noninfected with HIV and >5 mm for individuals infected with HIV. In this initial study, a total of 2,585 individuals were enrolled in the household contact cohort. Of these individuals, 198 (10.7%) remained persistently TST negative over the 2-year follow-up period upon their enrollment.

Between 2014 and 2017, 691 household contacts from the initial study were identified as eligible for retracing according to the epidemiologic risk score criteria previously published[54]. Of these individuals, 441 (63.8%) were enrolled in a subsequent longitudinal follow-up retracing study. The mean time between enrollment in the initial study and completion of the retracing study was 9.5 years. During the retracing study, individuals completed three QFT assays over 2 years. On their final visit, individuals also underwent the TST (positive TST defined above). RSTR were classified as such if all TST assays (five from the initial study and one at the end of the retracing study) and the three QFTs from the retracing study were concordantly negative, while LTBI participants were classified as such if all TST and QFT assays were positive. PBMC were isolated from whole blood collected during the retracing study by Ficoll–Hypaque density centrifugation and cryopreserved until use.

For the present study, we selected PBMC from a subset of 45 LTBI (mean age of 23.9 years; 23 males and 22 females) and 45 RSTR (mean age of 22.8 years; 22 males and 23 females) subjects from the retracing study after matching for age, sex, exposure risk score and documented lack of HIV co-infection (Supplementary Tables 4 and 5). No power calculations were performed to predetermine sample sizes, but our sample sizes are similar to those reported in our published studies of this cohort[5,41]. All study subjects gave written, informed consent, approved by the National AIDS Research Committee, the Uganda National Council for Science and Technology and the institutional review board at University Hospitals Cleveland Medical Center.

### ACS

The ACS is a published cohort that followed healthy adolescents (*n* = 6,363; aged 12–18 years) from South Africa and monitored them for progression from latent TB infection to active disease over

2 years[28]. We obtained publicly available bulk RNA-seq data measured in whole-blood samples every 6 months for 2 years from 144 individuals (aged 12–18 years; 48 males and 96 females)[28,31]. At baseline, adolescents were diagnosed with LTBI by a positive QFT assay (>0.35 IU ml⁻¹) and/or a positive TST (>10 mm). Progression to active TB was determined by evidence of intrathoracic disease (two positive sputum smears or one positive Mycobacteria Growth Indicator Tube liquid culture). Of these 144 individuals, 43 progressed to active TB (referred to here as progressors), while the remaining 101 did not develop active TB during the study period (referred to here as nonprogressors). Progressors only include individuals who developed active TB over 6 months after enrollment (or the first positive TST or positive QFT assay) to exclude early asymptomatic disease. From the same study, we also obtained publicly available data on single-cell targeted transcriptomics performed on sorted CD4+ T cells of PBMC after stimulation with *Mtb* lysate obtained at the same time points mentioned above, in which there were 70 participants with 35 progressors and 35 nonprogressors[31]. All individuals were HIV-uninfected.

### NHPs

We obtained publicly available gene sets from single-cell sequencing data of granuloma homogenate from cynomolgus macaques (*n* = 4; mean age of 7 years, 3 males and 1 female)[32]. The cynomolgus macaques were infected with low-dose *Mtb* via bronchoscopic instillation (<10 c.f.u., Erdman strain). Infection was confirmed at 4 weeks by positron emission tomography computed tomography and monitored until necropsy 10-weeks postinfection by clinical and radiographic examinations. We utilized gene sets from single-cell sequencing data obtained from 26 granulomas sampled 10-weeks postinfection at necropsy.

Additionally, we obtained whole-blood transcriptome data and flow cytometry data of bronchoalveolar lavage fluid from rhesus macaques (*n* = 34; median age of 4.4 years, 16 males and 18 females)[33,34]. The rhesus macaques were randomized into six vaccine groups (based on birth colony, gender and prevaccination CD4 T cell responses to *Mtb* PPD) and received intravenous BCG vaccination of varying doses (4.5–7.5 log₁₀ c.f.u. in half-log increments). The rhesus macaques received *Mtb* challenge via bronchoscope (average 12 c.f.u., Erdman strain) 24 weeks postvaccination. We analyzed bulk RNA-seq data from whole blood collected at baseline, 2 days, 2 weeks, 4 weeks and 12 weeks postvaccination. We analyzed flow cytometry data from bronchoalveolar lavage fluid collected at baseline, 2 weeks, 4 weeks, 8 weeks and 12 weeks postvaccination. The rhesus macaques were euthanized either 36 weeks postvaccination or when they developed clinical signs of humane endpoints, and total *Mtb* c.f.u. was measured upon necropsy.

### Antigens

Overlapping peptide pools targeting ESAT6 and CFP10 were used to stimulate T cells for these studies (BEI Resources). Peptides were 15 or 16 mers with 11 or 12 amino acid overlaps for ESAT6 protein and 11 amino acid overlaps for CFP10. *Mtb* whole-cell lysate from H37Rv was also used to stimulate T cells (BEI Resources). Dimethyl sulfoxide (DMSO) (Sigma-Aldrich) was used as a negative control. *Staphylococcus enterotoxin* B (List Biological Laboratories) was used as a positive control for the low-exposure cohort.

### ICS

ICS was performed on samples from the low-exposure cohort as we have previously described[5]. The same ICS assay and flow cytometry acquisition method was performed on samples from the household contacts with minor modifications. Before staining, samples from the household contacts were divided in half to be analyzed using two multiparameter flow cytometry panels, one for the analysis of T_reg subsets and one for T_H subsets (panel details in Supplementary Table 3).

Cells were permeabilized and underwent intracellular staining using the eBioscience Foxp3/Transcription Factor Staining Buffer Set (eBioscience) according to the manufacturer's directions to allow for the analysis of transcription factors. In the household contacts data, PBMC from two RSTR subjects were excluded from data acquisition and analysis due to bacterial contamination of the samples after overnight rest. The investigators were blinded to group allocation during acquisition of flow cytometry data in the validation household contact cohort.

### SELECT-seq on *Mtb*-reactive T cells

We conducted single-cell whole transcriptomics using the SELECT-seq protocol, which includes stimulation and sorting procedures[16,31]. In brief, PBMC from RSTR and LTBI samples were stimulated as described in the ICS methods above. Notably, before the stimulation, the cells were cultured in 1 µg ml⁻¹ anti-CD154 antibody for 30 min to prevent CD154 downregulation. After stimulation, cells were first stained (Supplementary Table 3) and index sorted on live CD3⁺/TCRαβ⁺ cells positive for CD69 and CD154 and/or CD137 on a BD FACSAria Fusion. These antigen-specific T cells were fluorescence-activated cell sorted (FACS) into individual wells of a 96-well PCR plate with lysis buffers. To assess the technical variability, we mixed the lysis buffers with the external RNA control−ERCC RNA Spike-In (a final estimate of ~5,000 molecules per well, Thermo Fisher Scientific). We used the modified Smart-seq2 protocol (Clontech Laboratories) to generate the complementary DNA library. From the library, we took a small aliquot (1 µl) for the nested PCR to amplify and sequence 15 targeted RNAs (*TBX21*, *RORC*, *GATA3*, *FOXP3*, *RUNX1*, *RUNX3*, *IFNG*, *IL2*, *IL17A*, *GZMB*, *PERF*, *IL4*, *IL5*, *IL13* and *TGFB*) and the CDR3 regions of both TCRα and TCRβ chains. Based on the TCR sequences, the Smart-seq2-generated complementary DNAs of the clonally expanded activated T cells were manually selected only on the ESAT6/CFP10-reactive cells for the high-coverage in-depth single-cell full transcriptomic sequencing. We applied the tagmentation and indexing protocol as in the Smart-seq2 protocol and amplified the tagmented DNA. The final pooled library was prepared using the Nextera XT library prep kit (96 index primers; Illumina) protocol and was sequenced on an Illumina HiSeq 2500.

### T cell proliferation assay

PBMC from RSTR (*n* = 12) and LTBI (*n* = 12) were thawed and enumerated as described above. After resting the samples for 2 h, the cells were washed with cold CFSE buffer (PBS supplemented with 5% FBS) and adjusted to 5 × 10⁶ cells ml⁻¹. The samples were then stained with 5 µM CFSE (BioTracker 488 Green CSFE Cell Proliferation Kit; Sigma-Aldrich) for 5 min at room temperature (RT), followed by three successive washes using CFSE buffer. After the third wash, the cells were reconstituted in media and incubated with anti-CD154 (1 µg ml⁻¹) (Miltenyi) for 30 min at 37 °C to prevent internalization of surface CD154. The cells were then stimulated overnight with DMSO, the whole *Mtb* lysate (100 µg ml⁻¹) or ESAT6/CFP10 peptide pool (1 µg ml⁻¹). Following stimulation, the cells were stained with LIVE/DEAD Fixable Aqua (Thermo Fisher Scientific) for 15 min at RT. Next, the cells were washed with FACS buffer and stained with CCR7 (phycoerythrin (PE)) for 30 min at 37 °C. The cells were centrifuged and washed with FACS buffer and stained with CD154 (BV711), CD69 (BV450), CD95 (PE−Dazzle 594), CD45RA (PE−Cyanine7) and CD45RO (PerCP-Cyanine5.5) (panel details in Supplementary Table 3) for 30 min at 4 °C. The live CD154⁺/CD69⁺CCR7⁺CD45RA⁺CD45RO⁻ populations were sorted into 5-ml FACS tubes using a BD FACSAria III Cell Sorter. These cells were then transferred to a 96-well U-bottom plate (Corning) and cultured in sterile-filtered RPMI 1640 (Gibco) supplemented with 10% FBS (HyClone) and 50 units ml⁻¹ recombinant human IL-2 (Prometheus Pharmaceuticals through UWMC Clinical Pharmacy) at 37 °C. After 7 days in culture, the cells were stained with LIVE/DEAD Fixable Aqua (Thermo Fisher Scientific) for 15 min at RT. Next, the cells were washed and stained with CCR7 (PE) for 30 min at 37 °C. The cells were then washed and stained with

additional phenotypic markers (Supplementary Table 3) for 30 min at 4 °C. Finally, the cells were fixed using 1% paraformaldehyde (Electron Microscopy Sciences) for 15 min at 4 °C and acquired using a BD LSRFortessa Cell Analyzer. Data were analyzed using FlowJo version 10.10.00 (BD Biosciences).

### Multiplex cytokine analysis and ELISA

Cytokine profiles were assessed for 18 RSTR and 20 LTBI participants from the household contact cohort using custom 27-plex ProcartaPlex Panel kits (Invitrogen) and enzyme-linked immunosorbent assay (ELISA). Briefly, 100 µl of supernatant from PBMC stimulated with DMSO or ESAT6/CFP10 peptide pool for 6, 12, 24 and 48 h was collected and stored at −80 °C for downstream analyses. For assessment via multiplex cytokine analysis, 50 µl of undiluted culture supernatant was incubated with magnetic capture beads conjugated to analyte-specific antibodies in 96-well plates. The plates were then washed, and the wells containing samples and beads were incubated with detection antibodies followed by streptavidin conjugated to phycoerythrin. Cytokine secretion data (Granulocyte-macrophage colony-stimulating factor (GM-CSF), IFN-α, IFN-γ, IL-1α, IL-1β, IL-10, IL-12p70, IL-13, IL-15, IL-17A, IL-18, IL-1RA, IL-2, IL-21, IL-22, IL-23, IL-27, IL-3, IL-4, IL-5, IL-6, IL-7, IL-8, IL-9, IP-10, TNF-α and TNF-β) were then acquired using the Bio-Plex 200 suspension array system (Bio-Rad). Only samples with bead counts >50 were considered. Due to the concentration of several analytes falling outside the linear range of their respective standard curve, the mean fluorescence intensity (MFI) value for each cytokine per sample was extracted and analyzed in R.

Due to incompatibility with ProcartaPlex chemistry, TGFβ-1 was detected separately via ELISA. Briefly, 96-well high-binding ELISA plates (Millipore) were coated overnight with mouse anti-human TGFβ-1 IgG (BioLegend) in carbonate coating buffer at 4 °C. Then, the plates were washed and blocked using 2% BSA in PBS for 3 h at 37 °C. After blocking, the plates were washed, 50 µl of the experimental sample or TGFβ-1 standard was added to each well, and the plates were incubated overnight at 4 °C. The supernatant was collected, the plates were washed and 50 µl of biotinylated mouse anti-human TGFβ-1 IgG (BioLegend) was added to each well, and the plates were incubated at 37 °C for 3 h. Next, the plates were washed, and 50 µl of streptavidin conjugated to horseradish peroxidase (BD Biosciences) was added to each well. The plates were incubated at room temperature for 30 min and then washed, and 50 µl of substrate buffer (*o*-phenylenediamine dihydrochloride (Thermo Fisher Scientific) and Pierce Stable Peroxide Buffer) was added to each well. The plates were developed for 10–15 min at RT before plate absorbance was read at 450 nm using a CLARIOstar Plus Microplate Reader (BMG LabTech).

### Computation/statistics

**Flow cytometry.** ICS data were compensated and gated using FlowJo (v9.9.6) (BD Biosciences). Representative gating trees of the low-exposure controls and the household contacts are shown in Extended Data Fig. 2. The data were then processed using the OpenCyto framework (V2.16.1) in the R programming environment (V4.1.2)[55]. With the data from the low-exposure controls COMPASS (V1.19.4) was used to achieve a comprehensive and unbiased analysis of the activation profiles of antigen-specific T cells as previously described[5,15]. For a given subject, COMPASS was also used to compute a polyfunctionality score that summarizes the entire functionality profile into a single number in which greater weight is given to subsets with more than one function. COMPASS was performed on data from the antigen stimulations for CD4⁺ T cells to assess T cell subsets expressing IFN-γ, IL-17A, IL-4/5/13, CD107a, TNF, IL-2 and CD154. Poor-quality samples were identified by low CD3 (<10,000 cells) or CD4 (<3,000 cells) counts and were excluded from downstream analysis. The R package ComplexHeatmap (V1.15.1) was used to visualize COMPASS posterior probabilities of response[56]. For all flow cytometry data, magnitudes of T cell responses

were calculated as the proportion of gated events. We used Wilcoxon rank-sum tests to compare magnitudes of T cell responses between sample groups.

**Index sort and targeted RNA-seq.** The index sort flow cytometry MFI and targeted RNA-seq count dataset were analyzed in R using the packages FlowSOM (V2.1.11)[17] and CATALYST (V1.14.1)[57]. First, all MFI values were shifted by a minimal constant to ensure they were above zero. The non-negative MFI values helped prevent the data overspill. Next, the MFIs underwent an arcsinh5 transformation. The RNA-seq count data were binarized with a cutoff of 5 due to its binary property[58]. The combined transformed data matrix was standardized to have a mean of 0 and a standard deviation of 1 within each marker across all cells. The CATALYST workflow conducted two series of clustering for the computational efficiency with many cells. The cells were clustered by FlowSOM into 100 groups with a grid size of 15. These 100 groups were meta-clustered by the agglomerative hierarchical clustering to a preset cluster number 40 to identify rare subsets, such as $T_{reg}$ and $T_H1^*$. We visualized these cell phenotypes in the $t$-SNE plot calculated in CATALYST. Next, we manually merged and annotated these cell phenotype clusters. We used Wilcoxon rank-sum tests to compare the cell phenotype subset composition between the two groups and corrected for multiple hypothesis testing using the Bonferroni method.

**Full transcriptomic scRNA-seq.** Python (V3.9) was used for preprocessing of raw sequencing data. To convert the sequencing read fastq files into single-cell count tables, we used STAR[59] to align with the reference GRCh38.genome.ERCC.fa and an index length of 150 bp, and featureCounts[60] to calculate the count tables. The single-cell count tables were imported as a Seurat object using the Seurat package (V4) in R (V4.1.2) for downstream analysis. We filtered cells with unique featureCounts over 4,000, total counts below $5 \times 10^5$ and mitochondrial counts >5% for doublets, dropouts and dying cells (Extended Data Fig. 4a). CD8+ T cells were removed due to a low cell number. We gated CD4+ T cells based on their CD4 and CD8 flow MFIs. The principal component analysis reduced the expressed genes into 30 components. To study cell phenotypic heterogeneity, these CD4+ T cells were clustered using the graphical-based clustering method and visualized in the Uniform Manifold Approximation and Projection plot. To identify significantly differentially expressed genes (DEGs) between the two groups, we used the Wilcoxon rank-sum test with cutoffs of 0.05 in the Benjamini–Hochberg-adjusted $P$ values and 0.5 in the absolute $\log_2$ of fold change (FC) of the average expression between the two groups. The significantly upregulated genes in RSTR were used to run the GO analysis using DAVID GO[61,62]. For the network analysis, we screened the immune-related genes (GO0002376: immune_system_process) and ran the STRINGdb (V2.16.4) network clustering algorithm in R[30]. The gene regulatory network inference and motif discovery were conducted using SCENIC (V1.1.2)[63].

**Mutiplex cytokine analysis and ELISA.** The multiplex cytokine and TGFβ ELISA data were exported as comma-separated value (CSV) files from the Bio-Plex 200 CLARIOstar Plus Microplate Reader, respectively. All the cytokine data were merged, cleaned and analyzed in R. Due to the low concentration of cytokines detectable by the Procartaplex kit, the MFI data for each cytokine were extracted and assessed. Background-corrected signals were computed by subtracting the signal from the DMSO condition from the antigen stimulation condition. To choose the precise time points with which to compare protein secretion (measured by ProcartaPlex and ELISA) between the RSTR and LTBI groups, we identified samples exhibiting the most favorable signal-to-noise ratio in response to stimulation (ESAT6/CFP10 or $Mtb$ lysate) versus DMSO. This selection was based on applying the $t$-statistic at each time point assessed. Subsequently, statistical testing between the two groups was performed on a single time point for each analyte

using the Student's $t$-test (Extended Data Figs. 7 and 8). $P$ values less than 0.05 were considered significant. Data distribution was assumed to be normal but this was not formally tested.

**ACS cohort analysis.** We summarized a 7-gene differentiation module (*CCR7*, *SELL*, *LEF1*, *TCF7*, *FOXP1*, *IL7R* and *CD27*) and the 24-gene activation module (*CTLA4*, *IKZF1*, *CD5*, *IL2RB*, *BATF*, *TNFRSF4*, *CXCR4*, *ICOS*, *JAK3*, *TNFRSF18*, *FAS*, *JAK1*, *UBASH3A*, *S1PR4*, *NFKB2*, *PIK3CD*, *PRKCQ*, *ETS1*, *AHR*, *RASGRP1*, *SASH3*, *POU2F2*, *TTC7A* and *RFTN1*) as the geometric mean of the module gene expression level, and we evaluated whether these modules were differentially expressed between progressors and nonprogressors at all time points before sputum conversion[28]. The phenotypic CD4+ T cell fractions were calculated from the non-mucosal-associated invariant T (MAIT) CD4+ T cells previously published[31]. The targeted transcriptional profiling count data were binarized with a cutoff of 5 as we have done previously[36] to study cells expressing *RORC*, *TBX21* and *FOXP3*.

**NHP cohort analysis.** We derived the gene expression patterns identified in 26 granulomas from 4 cynomolgus macaques obtained 10 weeks after low-dose $Mtb$ infection and analyzed using single-cell RNA-seq[32]. We selected the top 15 enriched genes in each of the stem-like cell subset and T1–T17 population 1 cell subset and calculated their mean expression using scaled log-normalized gene counts in RSTR and LTBI subjects using the SELECT-seq dataset. Genes that were not expressed or not found were excluded from the analysis. We used the AddModuleScore function from Seurat to calculate the associated gene module scores.

We examined the expression of the $T_H17$ gene module in the whole-blood transcriptome of 34 rhesus macaques from a dose-ranging study of intravenous BCG vaccination followed by $Mtb$ challenge. Data collection and preprocessing have been described in detail previously[33]. We calculated the RSTR-associated $T_H17$ module score using the geometric mean of genes identified in Fig. 5b that could be mapped to the rhesus macaque genome (*CCR6*, *CCR4*, *BATF* and *RORA*). We then compared this module score across all time points between macaques that were protected ($n = 18$) versus not protected ($n = 16$) against $Mtb$ challenge, as determined by total $Mtb$ c.f.u. upon necropsy, using the two-sided Wilcoxon rank-sum test. Flow cytometry was performed on freshly collected bronchoalveolar lavage fluid obtained from these same macaques[34]. From these data, we extracted the IL-17-monofunctional T cell phenotype measured by IFNγ⁻IL-2⁻IL-17+TNF⁻ among CD4+ T cells after PPD stimulation.

**Reporting summary**

Further information on research design is available in the Nature Portfolio Reporting Summary linked to this article.

## Data availability

All the validation flow cytometry data are available for download via ImmPort at https://www.immport.org under study accession number SDY2277 and via Fairdomhub at https://fairdomhub.org/studies/1179. The processed Seurat object generated from the SELECT-seq data is available via Zenodo at https://zenodo.org/records/7946277 (ref. 64). The raw and processed SELECT-seq data is available at Gene Expression Omnibus (GEO; accession number GSE267774). The gene sets GO:0072539 and GO:0002376 from MSigDB were used to analyze SELECT-seq data. From the ACS cohort, whole-blood bulk transcriptomics data is available at GEO (accession number GSE79362)[28] and single-cell targeted transcriptomics data can be found in supplementary materials in the published study from Musvosvi et al.[31].

## Code availability

Code to complete multiplex cytokine and flow cytometry analyses can be found via GitHub at https://github.com/seshadrilab/cd4-phenotypes-sun-2024-procartaplex and https://github.

com/seshadrilab/cd4-phenotypes-sun-2024-flow. Code to complete SELECT-seq analysis and analyses of ACS and NHP cohort data can be found via GitHub at https://github.com/ttsunmeng/TB_RSTR_ESAT6CFP10_SelectSeq_analysis_pipeline.

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

## Acknowledgements

We acknowledge the invaluable contribution made by the Kawempe study team's medical officers, health visitors, laboratory and data personnel in Uganda and the USA. The present study would not have been possible without the generous participation of the Ugandan TB patients and their families. We thank D. Lauffenberger from MIT in collaborating to analyze the intravenous BCG data. We thank members of the Human Immune Monitoring Center for facilitating this work, especially A. Cheruku. Furthermore, we thank J. Wilhelmy, A. McSween, Q. Xia and C. Wang from the Davis lab for their input on assay development and data analysis. Cell sorting/flow cytometry analysis for this project was done on instruments in the Stanford Shared FACS Facility. We thank the Hi-IMPAcTB Data Management team for organizing the data associated with this project for FAIR sharing, namely C. Demurjian, E. Koo, S. Levine (MIT BioMicro Center) and D. Mugahid (Harvard School of Public Health). This work was supported by the US National Institutes of Health (R01-AI124348 to W.H.B., C.M.S. and T.R.H.; U01-AI115642 to W.H.B., T.R.H., C.M.S. and H.M.-K.; K24-AI137310 to T.R.H.; R01-AI125189 and R01-AI146072 to C.S.; and 75N93019C00071 to C.S. and W.H.B.), the Bill and Melinda Gates Foundation (OPP1151836 and OPP1109001 to W.H.B., T.R.H., C.M.S. and H.M.-K. and OPP1113682 Center for Human Systems Immunology award to M.M.D.) and the Howard Hughes Medical Institute (M.M.D.). The funders had no role in study design, data collection and analysis, decision to publish or preparation of the manuscript. Additionally, we acknowledge E.D. Layton for technical assistance. Finally, we would like to thank E. Nemes and K. Urdahl for providing critical feedback on the paper before submission.

## Author contributions

C.S. and M.M.D. conceived the study. M.S., J.M.P. and N.S.K. performed the experiments, analyzed the data, and generated the figures and tables. M.T.S., H.H., S.G., S.-H.C., M.G. and P.K. facilitated computational analyses. K.K.Q.Y., G.O., H.T.M., A.K., S.S., N.G., M.R. and P.A. facilitated SELECT-seq experiments at the Stanford Human Immune Monitoring Core. C.M.S., H.M.-K., T.R.H. and W.H.B. contributed to epidemiologic analyses and established the clinical cohorts. Y.E.L., C.W. and P.K. contributed the analyses in the NHP study. M.S., J.M.P., N.S.K. and C.S. wrote the paper with contributions from all authors. C.M.S. and W.H.B. facilitated access to biospecimens. C.M.S., W.H.B., T.R.H., M.M.D. and CS provided funding and oversight for the work. All authors read and approved the final paper.

## Competing interests

The authors declare no competing interests.

## Additional information

**Extended data** is available for this paper at https://doi.org/10.1038/s41590-024-01897-8.

**Correspondence and requests for materials** should be addressed to Mark M. Davis or Chetan Seshadri.

**Peer review information** *Nature Immunology* thanks Fatoumatta Darboe and Jayne Sutherland for their contribution to the peer review of this paper. Primary Handling Editor: Ioana Staicu, in collaboration with the *Nature Immunology* team. Peer reviewer reports are available.

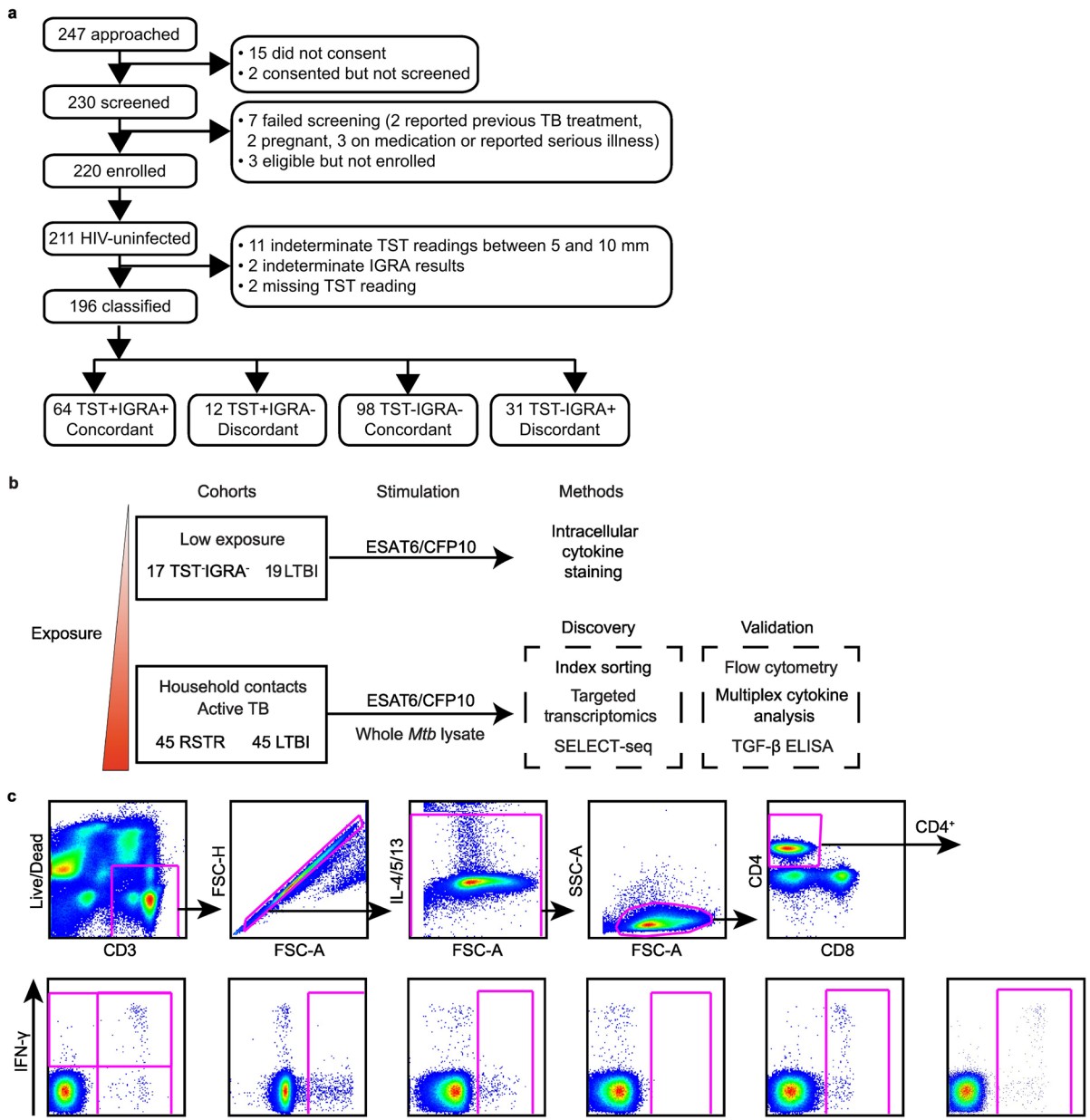

**Extended Data Fig. 1 | Enrollment of low exposure cohort, study schema and gating strategy. (a)** Screening and enrollment of low exposure cohort participants for health assessment and blood draw. Enrollment took place in Uganda from July 2017 to March 2018. All healthy non-pregnant participants were eligible. TB, tuberculosis; HIV, human immunodeficiency virus; TST, tuberculin skin test; IGRA, IFN-γ release assay. **(b)** Schematic of the low exposure cohort, containing 19 LTBI and 17 TST⁻IGRA⁻ participants, and the household contact

cohort, containing 45 RSTR and 45 LTBI participants. T cell responses to ESAT6/CFP10 or whole *Mtb* lysate were assessed using intracellular cytokine staining, targeted transcriptomics and SELECT-seq, and validated using flow cytometry and multiplex cytokine analysis on independent samples. **(c)** Flow cytometry gating strategy for CD4⁺ T cell subsets with cytokine and activation marker expressions from PBMCs in the low exposure control cohort stimulated with ESAT6/CFP10 peptide pool for 6 hours.

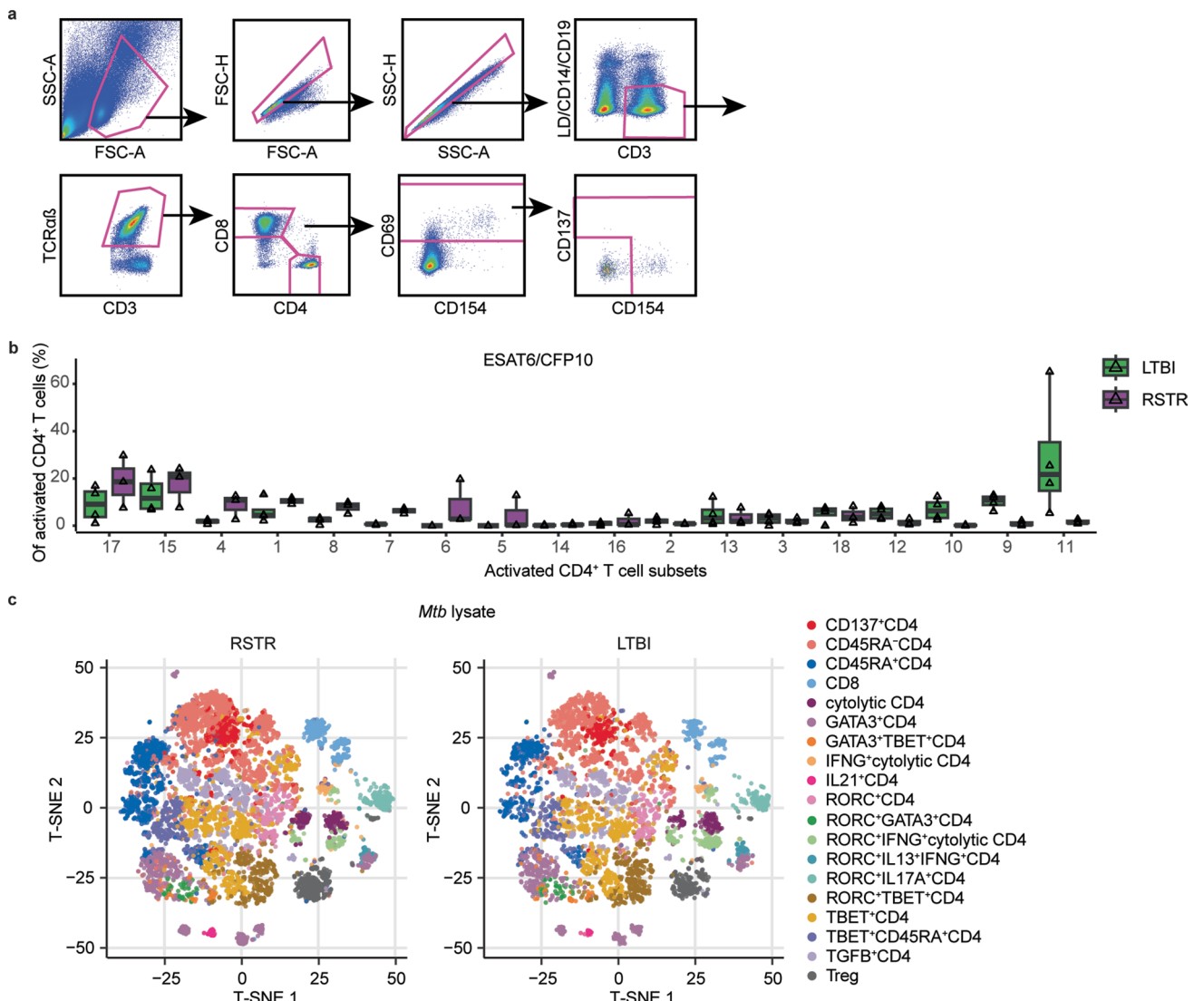

**Extended Data Fig. 2 | The composition of *Mtb*-specific T cell subsets.**
**(a)** Gating strategy of ESAT6/CFP10-specific CD4⁺ T cells among PBMCs from the household contact cohort using index sort. T cells were identified as CD3⁺CD14⁻/CD19⁻. T cells positive for the TCRαβ marker were gated on and separated into CD4⁺ or CD8⁺ subsets. Activated CD3⁺TCRαβ⁺ T cells were gated on using CD69, CD137, and CD154 markers. **(b)** Box plots showing the median and interquartile range of frequencies of ESAT6/CFP10-specific CD4⁺ T cell clusters

defined in Fig. 2b among RSTR (n = 3) and LTBI (n = 4) donors in the household contact cohort using index sort and targeted transcriptomics. Whiskers represent minima and maxima. Statistical testing was not performed due to the small sample sizes. **(c)** Dimensionality reduction (t-SNE) projection of activated CD4⁺ T cell subsets sorted from PBMCs among the household contact cohort after stimulation with *Mtb* lysate for 12 hours.

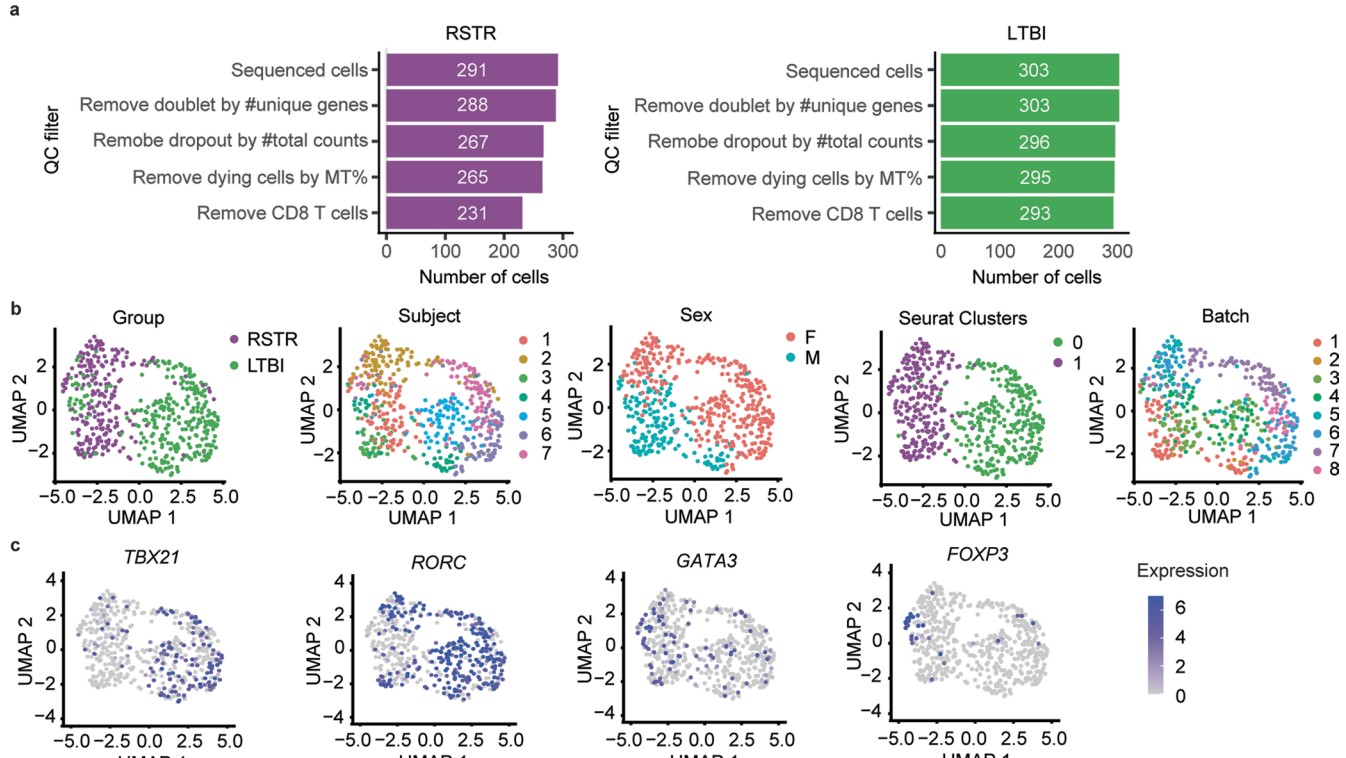

**Extended Data Fig. 3 | Quality control filtering and key T cell transcriptional factor gene visualization of the SELECT-Seq data. (a)** Filtered cell number by quality control parameters, including the number of unique expressed genes, total gene counts, and percentage of mitochondrial (MT) gene expression, from the whole transcriptomic data of clonally expanded ESAT6/CFP10-specific CD4+ T cells in the household contact cohort. **(b)** Dimensionality reduction (UMAP) projection of these clonally expanded CD4+ T cells showing group, subject, sex, sequencing batches, and subset clusters. **(c)** Expression of transcriptional factor genes essential for T cell polarization on these clonally expanded CD4+ T cells. The expression levels were calculated as the log-normalized gene counts from the whole transcriptomic data.

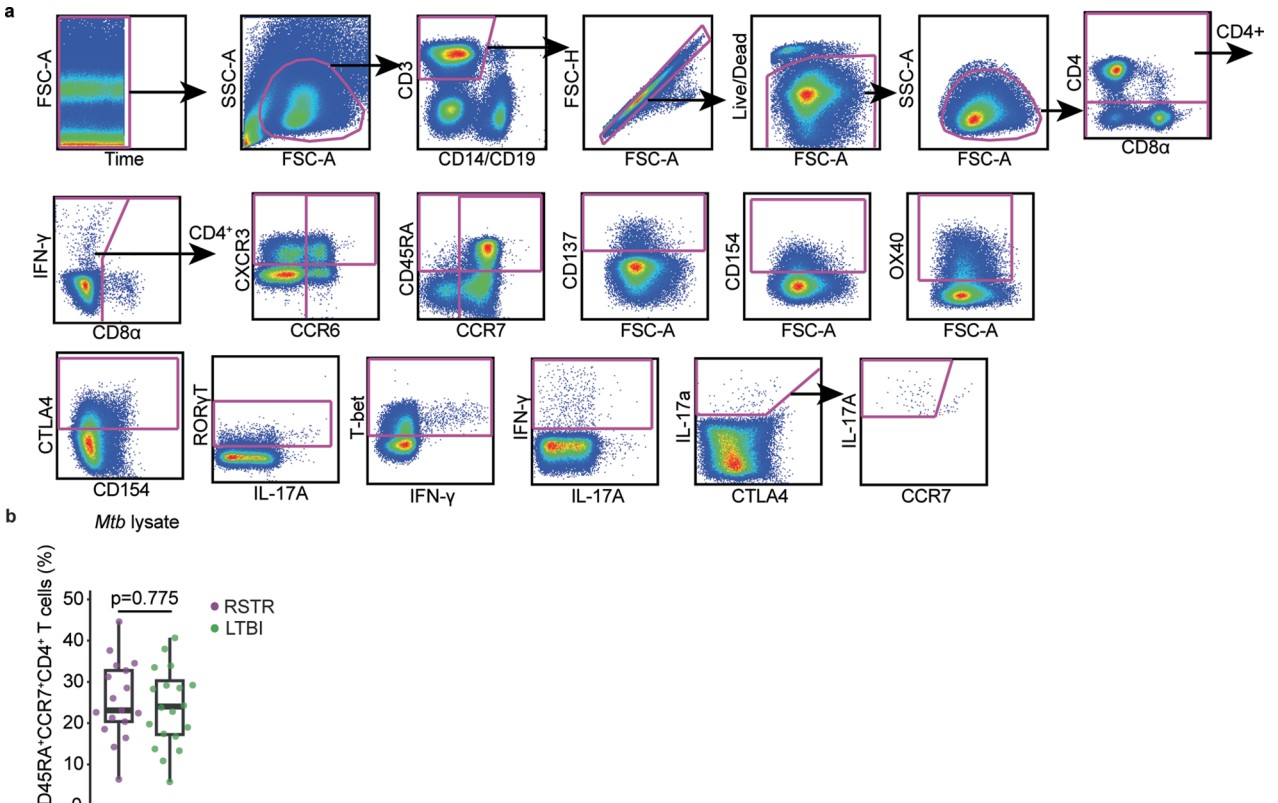

**Extended Data Fig. 4 | Gating strategy for T_H panel and naïve-like *Mtb* lysate-specific cell frequency in the validation cohort. (a)** Flow cytometry gating strategy for cells expressing memory markers, transcriptional factors and intracellular cytokines associated with T_H1 and T_H17 among the validation household contact cohort in response to stimulation with ESAT6/CFP10. A time gate was applied to exclude events affected by sample acquisition aberrations, followed by T cell identification by lymphocyte size, CD3, CD14, and CD19 markers. A singlet gate was then applied, followed by the identification of viable cells and a second lymphocyte gate. A secondary gate was applied to identify IL-17A⁺ cells due to spillover spreading. The same gating strategy was applied to samples stimulated with whole *Mtb* lysate (not shown). **(b)** Flow cytometry showing the median and interquartile range of frequencies of CD45RA⁺CCR7⁺ naïve-like cells in *Mtb* lysate-specific CD4⁺ T cells among validation household contact cohort RSTR (n = 17) and LTBI (n = 20). Whiskers represent minima and maxima. Statistical significance was determined by the two-sided Wilcoxon rank-sum test.

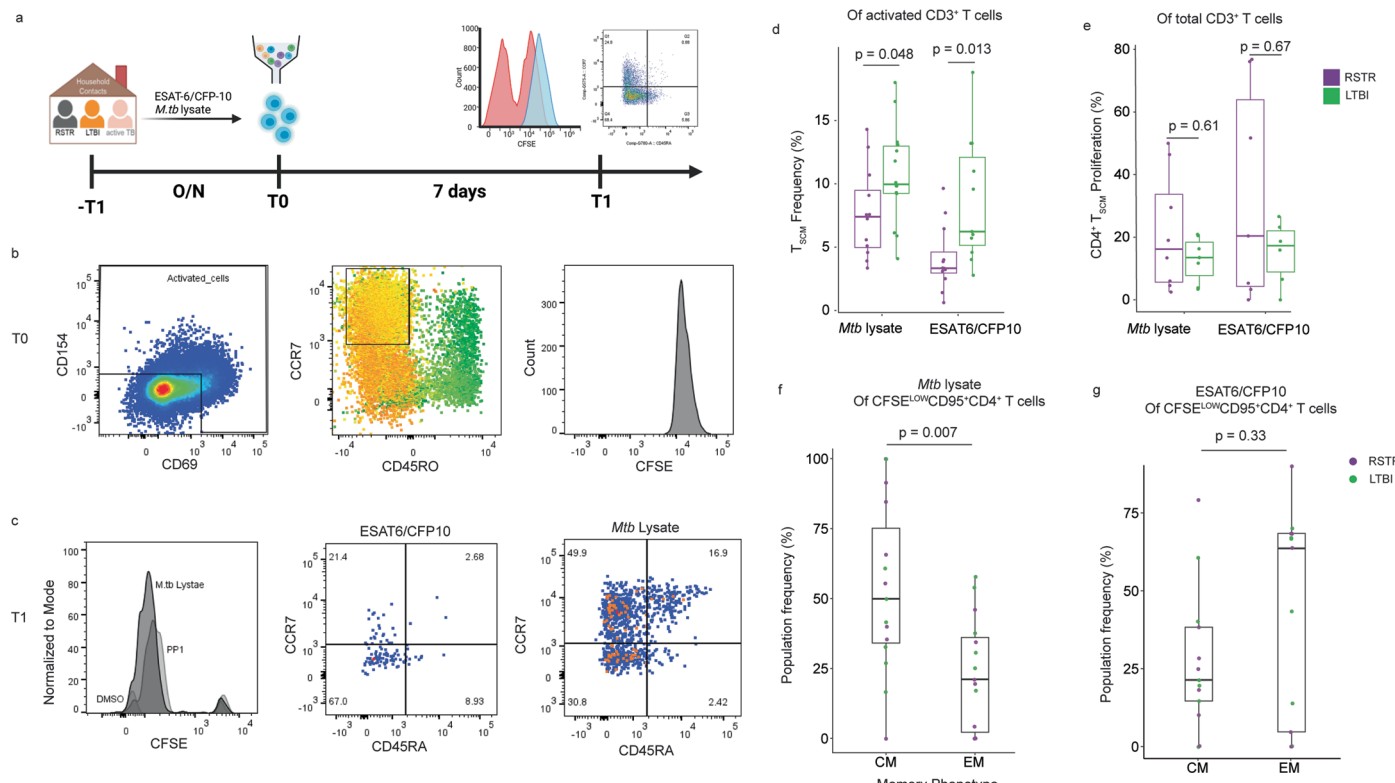

**Extended Data Fig. 5 | RSTR T$_{SCM}$ cells proliferate similarly to LTBI T$_{SCM}$ cells in response to *Mtb* antigen. (a)** Schematic of T$_{SCM}$ study where RSTR (n = 12) and LTBI (n = 12) PBMC from the household contact cohort were thawed, rested for 2 hours, stained with 5 μM CFSE, and stimulated with ESAT6/CFP10 or *Mtb* lysate overnight. **(b)** Gating strategy of activated naive-like CD69$^+$CD154$^+$CCR7$^+$CD45RA$^+$CD45RO$^-$ lymphocytes. **(c)** Representative flow cytometry showing sorted T$_{SCM}$ cells from RSTR and LTBI expressing the proliferation marker CFSE after 7-day culture. Representative CD45RA and CCR7 staining of the CFSE$^{low}$ population in

response to ESAT6/CFP10 and *Mtb* lysate 7-day post-stimulation **(d)** Box plots showing the median and interquartile range of frequencies of T$_{SCM}$ cells (CD95$^+$ CCR7$^+$CD45RA$^+$CD45RO$^-$) at the time of sorting from PBMC stimulated with *Mtb* lysate or ESAT6/CFP10. Whiskers represent minima and maxima. **(e)** Frequencies of CFSE$^{low}$CD95$^+$ CD4$^+$ T$_{SCM}$ cells after 7-day culture. **(f-g)** Frequencies of proliferated cells in response to **(f)** *Mtb* lysate and **(g)** ESAT6/CFP10 stimulation. CM, central memory; EM, effector memory. Significance in **d-g** was calculated using the two-sided Wilcoxon rank-sum tests.

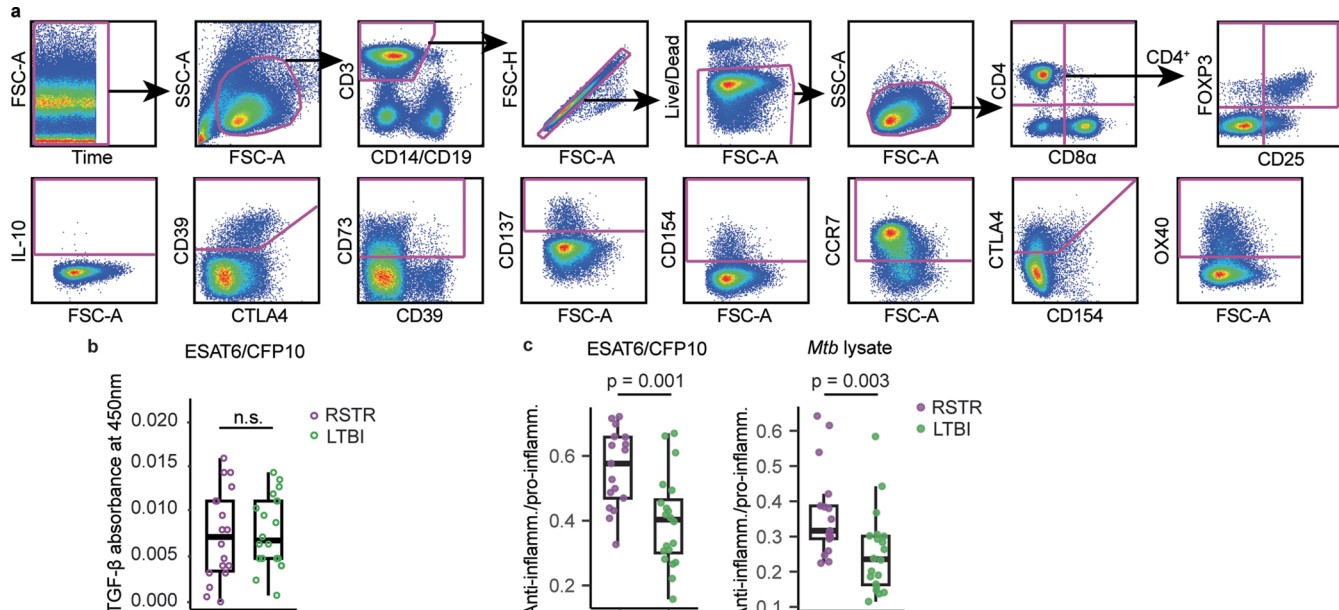

**Extended Data Fig. 6 | Gating strategy of T$_{reg}$ and its functional profiles.**
**(a)** Flow cytometry gating strategy for cells expressing T$_{reg}$ cell- and activation-associated markers, transcriptional factors and intracellular cytokines among the validation household contact cohort in response to stimulation with ESAT6/CFP10. A time gate was applied to exclude events affected by sample acquisition aberrations, followed by T cell identification by lymphocyte size, CD3, CD14, and CD19 markers. A singlet gate was then applied, followed by the identification of viable cells and a second lymphocyte gate. The same gating strategy was applied to samples stimulated with whole *Mtb* lysate (not shown). **(b)** ELISA showing the median and interquartile range of background-corrected absorbance level of TGF-β in conditioned supernatants from PBMCs in response to ESAT6/CFP10 stimulation among RSTR (n = 18) and LTBI (n = 20) in the validation household contact cohort. Whiskers represent minima and maxima. **(c)** Ratio of anti-inflammatory FoxP3$^+$CD25$^+$ T$_{reg}$ cell frequency to pro-inflammatory CD4 T cell frequency, including RORγt T-bet$^+$ T$_H$1, RORγt$^+$T-bet$^-$T$_H$17, and RORγt$^+$T-bet$^+$ T$_H$1* cells, among validation donors in response to ESAT6/CFP10 (left) or *Mtb* lysate (right) from flow cytometry data. Significance in **b,c** was determined by the two-sided Wilcoxon rank-sum tests.

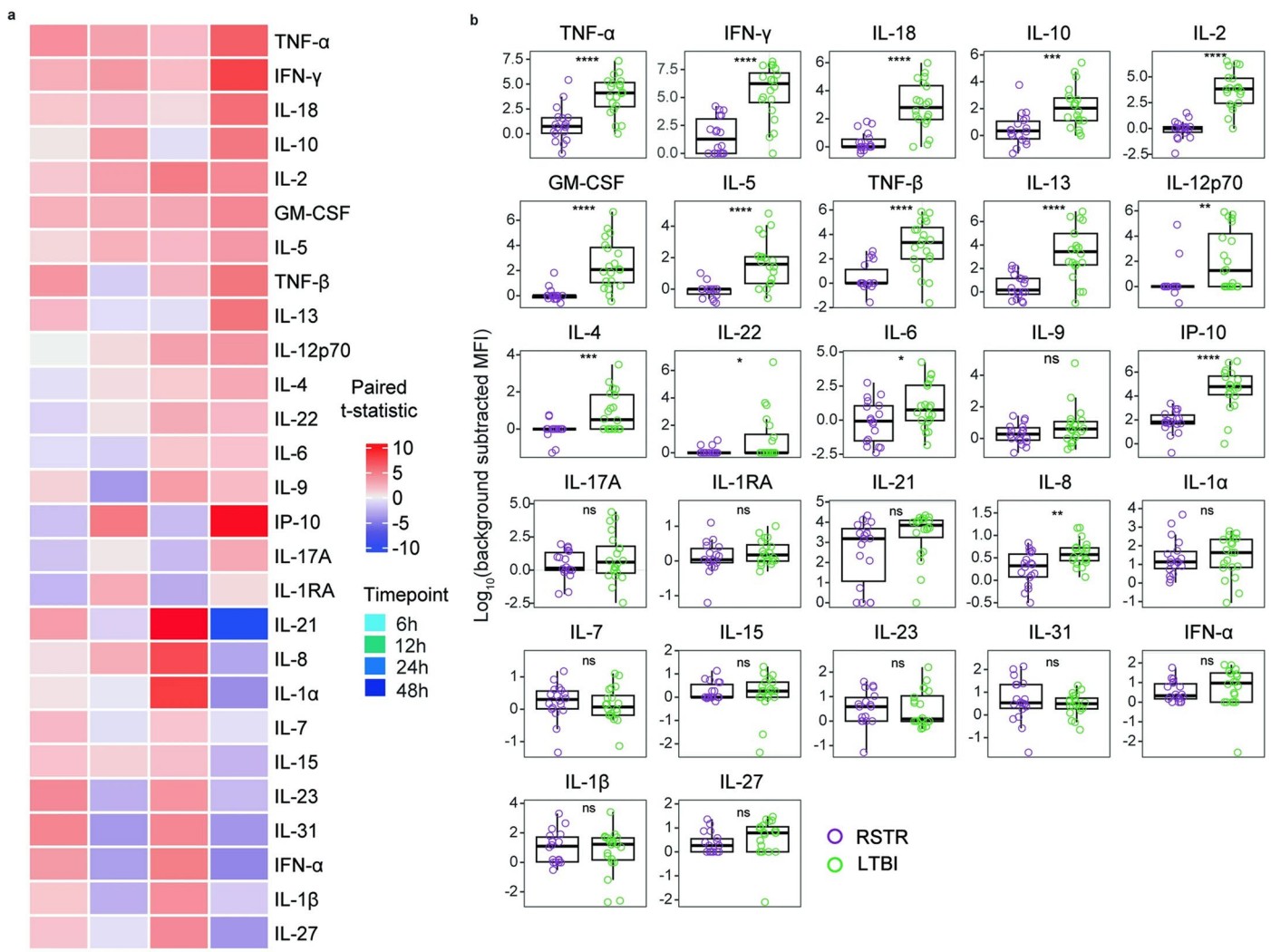

**Extended Data Fig. 7 | Cytokine secretion from LTBI and RSTR PBMC following ESAT6/CFP10 stimulation. (a)** Heatmap showing paired t-statistic of cytokine expression levels between ESAT6/CFP10 and DMSO stimulations from supernatants among RSTR (n = 18) and LTBI (n = 20) donors at 6, 12, 24, and 48 hours after stimulation. **(b)** Box plots showing the median and interquartile range of cytokine mean fluorescence intensities at the timepoint that exhibited the highest signal identified as in **a**. Whiskers represent minima and maxima. Significance was calculated by two-sided student's t-test. n.s. (not significant), p-value > 0.05; *, p-value < 0.05; **, p-value < 0.01; ***, p-value < 0.001.

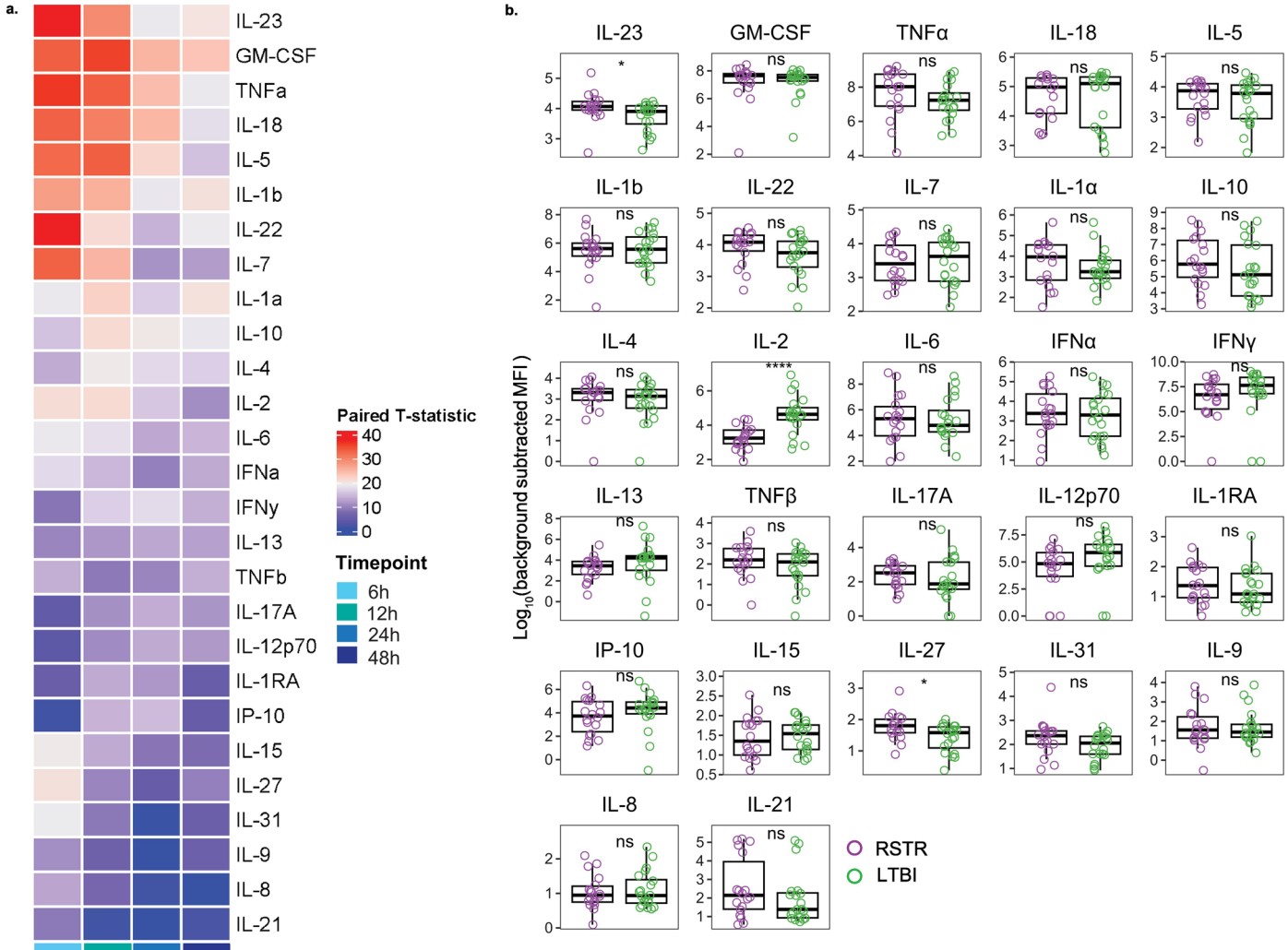

**Extended Data Fig. 8 | Cytokine secretion from LTBI and RSTR PBMC following *Mtb* lysate stimulation. (a)** Heatmap showing paired t-statistic of cytokine expression levels between *Mtb* lysate and DMSO stimulations from supernatants among RSTR (n = 18) and LTBI (n = 20) donors at 6, 12, 24, and 48 hours after stimulation. **(b)** Box plots showing the median and interquartile range of cytokine mean fluorescence intensities at the timepoint that exhibited the highest signal identified as in **a**. Whiskers represent minima and maxima. Significance was calculated by two-sided student's t-test. n.s. (not significant), p-value > 0.05; *, p-value < 0.05; **, p-value < 0.01; ***, p-value < 0.001.

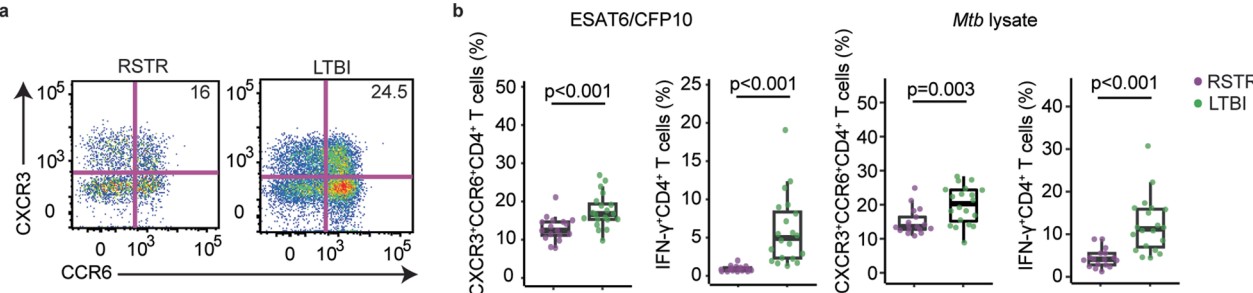

**Extended Data Fig. 9 | Representative staining of CXCR3⁺CCR6⁺ TH1 cells and IFN-γ⁺ CD4⁺ T cell frequencies. (a)** Representative staining for CXCR3⁺CCR6⁺ $T_H$1 cells in ESAT6/CFP10-specific CD4⁺ T cells from a participant in the validation household contact cohort. **(b)** Median and interquartile range of frequencies of CXCR3⁺CCR6⁺ $T_H$1 and IFN-γ⁺ CD4⁺ T cells in ESAT6/CFP10-specific CD4⁺ T cells or Mtb lysate-specific CD4⁺ T cells among RSTR (n = 17) and LTBI (n = 20). Whiskers represent minima and maxima. Significance was determined by two-sided Wilcoxon rank-sum tests.

# Reporting Summary

## Statistics

For all statistical analyses, confirm that the following items are present in the figure legend, table legend, main text, or Methods section.

| n/a | Confirmed | |
|---|---|---|
| ☐ | ☒ | The exact sample size (*n*) for each experimental group/condition, given as a discrete number and unit of measurement |
| ☐ | ☒ | A statement on whether measurements were taken from distinct samples or whether the same sample was measured repeatedly |
| ☐ | ☒ | The statistical test(s) used AND whether they are one- or two-sided *Only common tests should be described solely by name; describe more complex techniques in the Methods section.* |
| ☒ | ☐ | A description of all covariates tested |
| ☐ | ☒ | A description of any assumptions or corrections, such as tests of normality and adjustment for multiple comparisons |
| ☐ | ☒ | A full description of the statistical parameters including central tendency (e.g. means) or other basic estimates (e.g. regression coefficient) AND variation (e.g. standard deviation) or associated estimates of uncertainty (e.g. confidence intervals) |
| ☐ | ☒ | For null hypothesis testing, the test statistic (e.g. *F*, *t*, *r*) with confidence intervals, effect sizes, degrees of freedom and *P* value noted *Give P values as exact values whenever suitable.* |
| ☒ | ☐ | For Bayesian analysis, information on the choice of priors and Markov chain Monte Carlo settings |
| ☒ | ☐ | For hierarchical and complex designs, identification of the appropriate level for tests and full reporting of outcomes |
| ☒ | ☐ | Estimates of effect sizes (e.g. Cohen's *d*, Pearson's *r*), indicating how they were calculated |

*Our web collection on statistics for biologists contains articles on many of the points above.*

## Software and code

Policy information about availability of computer code

Data collection | The index-sorting data were collected using BD FACSDiva 8.0.1. Single-cell sequencing used Illumina instrumentation and softwares described in the method session. The flow cytometry data in the validation study were collected using a BD LSRFortessa. The multiplex cytokine data were acquired using the Bio-Plex 200 suspension array system (Bio-Rad, Hercules, CA). The ELISA plates were read at 450 nm using a CLARIOstar Plus Microplate Reader (BMG LabTech, Ortenberg, Germany).

| Data analysis | The index-sort and targeted PCR data in SELECT-seq were analyzed in CATALYST package (V2.1.11) in R (V4.1.2). For scRNA-seq in SELECT-seq, the raw data was first processed in Python (V3.9) and then analyzed using the Seurat package (V4) in R (V4.1.2). For the network analysis, we screened the immune-related genes (GO0002376:immune_system_process) and ran the STRINGdb (V2.16.4) network clustering algorithm. The gene regulatory network inference and motif discovery were conducted using SCENIC (V1.1.2). Code to complete SELECT-seq analysis and analyses of ACS and NHP cohort data can be found at https://github.com/ttsunmeng/TB_RSTR_ESAT6CFP10_SelectSeq_analysis_pipeline.<br><br>The flow cytometry data were compensated and gated using FlowJo (v9.9.6) (BD Biosciences, San Jose, CA). Representative gating trees of the low exposure cohort and the household contact cohort are shown in Extended Data Fig. 1, 2, 4, and 6. The data were then processed using the OpenCyto framework (V2.16.1) in the R programming environment (V4.1.2). With the data from the endemic controls, Combinatorial Polyfunctionality Analysis of Antigen-Specific T Cell Subsets (COMPASS) (V1.19.4) was used to achieve a comprehensive and unbiased analysis of the activation profiles of antigen-specific T cells. The R package ComplexHeatmap (V1.15.1) was used to visualize COMPASS posterior probabilities of response. The multiplex cytokine and TGFβ ELISA data were exported as CSV files from the Bio-Plex 200 CLARIOstar Plus Microplate Reader, respectively. All the cytokine data were merged, cleaned, and analyzed in R. The code to complete multiplex cytokine and flow cytometry analyses can be found at https://github.com/seshadrilab/cd4-phenotypes-sun-2024-procartaplex and https://github.com/seshadrilab/cd4-phenotypes-sun-2024-flow. |
|---|---|

For manuscripts utilizing custom algorithms or software that are central to the research but not yet described in published literature, software must be made available to editors and reviewers. We strongly encourage code deposition in a community repository (e.g. GitHub). See the Nature Portfolio guidelines for submitting code & software for further information.

## Data

Policy information about availability of data

All manuscripts must include a data availability statement. This statement should provide the following information, where applicable:

- Accession codes, unique identifiers, or web links for publicly available datasets
- A description of any restrictions on data availability
- For clinical datasets or third party data, please ensure that the statement adheres to our policy

All the validation flow cytometry data are available for download from ImmPort at https://www.immport.org under study accession number SDY2277 and at Fairdomhub at https://fairdomhub.org/studies/1179. The processed Seurat object generated from SELECT-Seq data is available at Zenodo at https://zenodo.org/records/7946277. The raw and processed SELECT-Seq data is available at Gene Expression Omnibus (GEO; accession number GSE267774). The gene sets GO:0072539 and GO:0002376 from MSigDB were used to analyze SELECT-Seq data. From the ACS cohort, whole blood bulk transcriptomics data is available at GEO (accession number GSE79362) from Zak et al. 2016 and single-cell targeted transcriptomics data can be found in supplementary materials in Musvosvi et al. 2023.

## Human research participants

Policy information about studies involving human research participants and Sex and Gender in Research.

| Reporting on sex and gender | Samples from study participants were selected after matching for sex. In the low exposure control cohort, subjects included in this study consisted of 19 male subjects and 17 female subjects (Supplementary Table 2). In the household contact cohort, subjects included in this study consisted of 38 male subjects and 39 female subjects (Supplementary Tables 4-5). We analyzed 524 T cells from the SELECT-Seq data after quality control filtering and found that the major axes of transcriptomic variance were group assignment (LTBI or RSTR) and sex, which were not mutually exclusive (Extended Data Fig. 3). |
|---|---|
| Population characteristics | For the low exposure cohort, the median age of the concordant positive (LTBI) group was 23 years and the median age of the concordant negative (TST-/IGRA-) group was 22 years. The participants of the LTBI and TST-/IGRA- group were 52.9% male and 52.6% male, respectively. All of the participants were HIV negative.<br><br>For the household contact cohort, the median age of the RSTR group was 23.2 years and the median age of the LTBI group was 24.8 years. The participants of the RSTR and LTBI groups were 48.6% male and 50% male, respectively. All of the participants were HIV negative. |
| Recruitment | The low exposure cohort was enrolled from a low TB incidence district that was identified by the Kampala Capital City Authority based on low TB transmission rates. Subjects were screened and enrolled for health assessment and blood draws between 2017 and 2018 in Uganda (Extended Data Fig. 1). All healthy, non-pregnant participants were eligible. A total of 247 individuals were approached in this district, of which 230 consented and were screened for previous TB treatment, pregnancy, medications, and serious illnesses. A total of 220 healthy, non-pregnant individuals were enrolled, and 211 of the enrolled were found to be non-infected with HIV. All participants reported no known contact with a TB case.<br><br>As we have previously described (Stein et al. 2018), household contacts of sputum culture positive cases of pulmonary TB were enrolled between 2002 and 2012 as part of the Kawempe Community Health Study. Adults with pulmonary TB were recruited from clinics at the Uganda National TB and Leprosy Program treatment center at Mulago Hospital, referred to the TB research clinic at Mulago Hospital, or recruited through community sensitization efforts in the Kawempe division of Kampala. Selection bias is primarily driven by referral to clinical care for which an effort was made to recruit through community sensitization. At baseline, individuals had no active Mtb infection determined by sputum culture and radiology. Upon enrollment, individuals were longitudinally screened during a two-year follow-up period by TST (Mantoux method, 0.1 ml of 5 tuberculin units of purified protein derivative (PPD), Tubersol; Connaught Laboratories), in which a positive TST was defined as an induration of >10 mm for individuals non-infected with HIV and >5mm for individuals infected with HIV. In this initial study, a total of 2,585 individuals were enrolled in the household contact cohort. Of these individuals, 198 (10.7%) remained persistently TST negative over the two-year follow-up period upon their enrollment. Between 2014 and 2017, 691 household contacts from the initial study were identified as eligible for retracing according to the epidemiologic risk score criteria previously published. Of these individuals, 441 (63.8%) were enrolled in a subsequent longitudinal follow-up retracing |

study. The mean time between enrollment in the initial study and completion of the retracing study was 9.5 years. During the retracing study, individuals completed three QFT assays over two years. On their final visit, individuals also underwent the TST (positive TST defined above). Resisters (RSTR) were classified as such if all TST assays (five from the initial study and one at the end of the retracing study) and the three QFTs from the retracing study were concordantly negative, while latent TB (LTBI) participants were classified as such if all TST and QFT assays were positive. All study subjects gave written, informed consent, approved by the National AIDS Research Committee, the Uganda National Council for Science and Technology, and the institutional review board at University Hospitals Cleveland Medical Center.

| Ethics oversight | The household contact retracing study protocol was reviewed and approved by the National AIDS Research Committee, The Uganda National Council on Science and Technology, and the institutional review board at University Hospitals Cleveland Medical Center. |
| --- | --- |

Note that full information on the approval of the study protocol must also be provided in the manuscript.

# Field-specific reporting

Please select the one below that is the best fit for your research. If you are not sure, read the appropriate sections before making your selection.

☒ Life sciences ☐ Behavioural & social sciences ☐ Ecological, evolutionary & environmental sciences

For a reference copy of the document with all sections, see nature.com/documents/nr-reporting-summary-flat.pdf

# Life sciences study design

All studies must disclose on these points even when the disclosure is negative.

| Sample size | No power calculations were performed to pre-determine sample sizes, but our sample sizes are similar to those reported in our published studies of this cohort (Lu et al. 2019, Simmons et al. 2021). |
| --- | --- |
| Data exclusions | In the validation flow cytometry study, PBMC from two RSTR subjects were found to have bacterial contamination after overnight rest and were excluded from data acquisition and analysis. |
| Replication | Verification of reproducibility of the low exposure control cohort findings could not be performed because samples from an independent cohort of LTBI and TST-/IGRA- subjects from the low exposure control cohort were not available. Verification of SELECT-Seq findings of the household contact cohort was performed by validation flow cytometry experiments with an independent cohort of RSTR and LTBI subjects from the household contact cohort. |
| Randomization | Participant allocation was not randomized nor applicable to the study. PBMC from a subset of LTBI and RSTR subjects were selected after matching for age, sex, exposure risk score, and documented lack of HIV co-infection. |
| Blinding | The investigators were blinded to group allocation during acquisition of flow cytometry data in the validation household contact cohort. |

# Reporting for specific materials, systems and methods

We require information from authors about some types of materials, experimental systems and methods used in many studies. Here, indicate whether each material, system or method listed is relevant to your study. If you are not sure if a list item applies to your research, read the appropriate section before selecting a response.

## Materials & experimental systems

| n/a | Involved in the study |
| --- | --- |
| ☐ | ☒ Antibodies |
| ☒ | ☐ Eukaryotic cell lines |
| ☒ | ☐ Palaeontology and archaeology |
| ☒ | ☐ Animals and other organisms |
| ☒ | ☐ Clinical data |
| ☒ | ☐ Dual use research of concern |

## Methods

| n/a | Involved in the study |
| --- | --- |
| ☒ | ☐ ChIP-seq |
| ☐ | ☒ Flow cytometry |
| ☒ | ☐ MRI-based neuroimaging |

## Antibodies

| Antibodies used | The following antibody information is also reported in Supplementary Table 3, along with their titers.

The ICS results in Fig. 1 utilized the "IPEC" panel which consisted of the following antibodies/markers: anti-CD107a PE-Cy7 (clone H4A3, BD Biosciences, catalog no. 561348); anti-CD3 ECD (clone UCHT1, Beckman Coulter, catalog no. IM2705U); anti-CD4 APC-Cy7 (clone 13B8.2, Beckman Coulter, catalog no. A94685); anti-CD8 PerCP-Cy5.5 (clone SK1, BD Biosciences, catalog no. 341051); anti-IL-2 PE (clone MQ1-17H12, BD Biosciences, catalog no. 559334); anti-IL-4 APC (clone MP4-25D2, BD Biosciences, catalog no. 554486); |
| --- | --- |

anti-IL-5 APC (clone TRFK5, BioLegend, catalog no. 504306); anti-IL-13 APC (clone JES10-5A2, BioLegend, catalog no. 501907); anti-IFN-g V450 (clone B27, BD Biosciences, catalog no. 560371); anti-TNFa FITC (clone MAb11, BD Biosciences, catalog no. 554512); anti-IL-17A AF700 (clone BL168, BioLegend, catalog no. 512318); anti-CD154 PE-Cy5 (clone TRAP1, BD Biosciences, catalog no. 555701); LIVE/DEAD™ Fixable Aqua Dead Cell Stain (Invitrogen, catalog no. L34966).

The index-sort results in Fig. 2 utilized the "Index sort" panel which consisted of the following antibodies/markers: LIVE/DEAD™ Fixable Aqua Dead Cell Stain (Invitrogen, catalog no. L34966); anti-CD3 BV786 (clone UCHT1, BioLegend, catalog no. 300472); anti-CD4 BV605 (clone RPA-T4, BioLegend, catalog no. 300556); anti-CD8a BUV496 (clone RPA-T8, BD Biosciences, catalog no. 564804); anti-TCRab PE-Cy7 (clone IP26, BioLegend, catalog no. 306720); anti-CD14 BV510 (clone M5E2, BioLegend, catalog no. 301842); anti-CD19 BV510 (clone HIB19, BioLegend, catalog no. 302242); anti-CD16 PE-Cy5 (clone 3G8, BioLegend, catalog no. 302010); anti-CD45RA FITC (clone HI100, BioLegend, catalog no. 304106); anti-CD154 PE (clone TRAP1, BD Biosciences, catalog no. 555700); anti-CD137 APC-Fire 750 (clone 4B4-1, BioLegend, catalog no. 309834); anti-HLA-DR AF700 (clone LN3, BioLegend, catalog no. 327014); anti-CD38 BV711 (clone HIT2, BioLegend, catalog no. 303528); anti-CD69 BUV396 (clone FN50, BD Biosciences, catalog no. 564364); anti-CD127 BUV737 (clone HIL-7R-M21, BD Biosciences, catalog no. 564300); anti-CXCR3 AF647 (clone G025H7, BioLegend, catalog no. 353712); anti-CCR6 BV421 (clone G034E3, BioLegend, catalog no. 353408); anti-CD25 PE-Dazzle 594 (clone M-A251, BioLegend, catalog no. 356126).

The ICS results in Figs. 3 and 5 utilized the "Th" panel which consisted of the following antibodies/markers: Zombie Yellow Fixable Viability Kit (BioLegend, catalog no. 423103); anti-CD3 BUV395 (clone UCHT1, BD Biosciences, catalog no. 563546); anti-CD4 BB515 (clone L200, BD Biosciences, catalog no. 564419); anti-CD8a BV510 (clone RPA-T8, BD Biosciences, catalog no. 563256); anti-CD14 BV785 (clone M5E2, BioLegend, catalog no. 301840); anti-CD19 BV785 (clone SJ25C1, BioLegend, catalog no. 363028); anti-CD45RA BUV737 (clone HI100, BD Biosciences, catalog no. 612846); anti-CCR7 BV711 (clone 150503, BD Biosciences, catalog no. 566602); anti-CD154 PE-Cy5 (clone TRAP1, BD Biosciences, catalog no. 310802); anti-CD137 BV605 (clone 4B4-1, BioLegend, catalog no. 309821); anti-OX40 PE-Cy7 (clone ACT35, BioLegend, catalog no. 350012); anti-CTLA-4 BB700 (clone BNI3, BD Biosciences, catalog no. 566901); anti-CXCR3 PE-Dazzle 594 (clone G025H7, BioLegend, catalog no. 353735); anti-CCR6 APC-Cy7 (clone G034E3, BioLegend, catalog no. 353431); anti-T-bet PE (clone 4B10, BioLegend, catalog no. 644809); anti-RORyT AF647 (clone Q21-559, BD Biosciences, catalog no. 563620); anti-IFN-g V450 (clone B27, BD Biosciences, catalog no. 560371); anti-IL-17a AF700 (clone BL168, BioLegend, catalog no. 512318).

This ICS results in Fig. 4 utilized the "Treg" panel which consisted of the following antibodies/markers: Zombie Yellow Fixable Viability Kit (BioLegend, catalog no. 423103); anti-CD3 BUV395 (clone UCHT1, BD Biosciences, catalog no. 563546); anti-CD4 BB515 (clone L200, BD Biosciences, catalog no. 564419); anti-CD8a BV510 (clone RPA-T8, BD Biosciences, catalog no. 563256); anti-CD14 BV785 (clone M5E2, BioLegend, catalog no. 301840); anti-CD19 BV785 (clone SJ25C1, BioLegend, catalog no. 363028); anti-CCR7 BV711 (clone 150503, BD Biosciences, catalog no. 566602); anti-CD154 PE-Cy5 (clone TRAP1, BD Biosciences, catalog no. 310802); anti-CD137 BV605 (clone 4B4-1, BioLegend, catalog no. 309821); anti-OX40 PE-Cy7 (clone ACT35, BioLegend, catalog no. 350012); anti-CTLA-4 BB700 (clone BNI3, BD Biosciences, catalog no. 566901); anti-FoxP3 eFluor 660 (clone PCH101, eBiosciences, catalog no. 50-4776-41); anti-CD25 BV421 (clone 2A3, BD Biosciences, catalog no. 612813); anti-CD39 APC-Cy7 (clone A1, BioLegend, catalog no. 328225); anti-CD73 BUV737 (clone AD2, BD Biosciences, catalog no. 612813); anti-IL-10 PE (clone JES3-19F1, BD Biosciences, catalog no. 559330).

The Tscm results in Extended Data Fig. 5 utilized the "Tscm" panel which consisted of the following antibodies/markers: anti-CD45RO PerCP Cy5.5 (clone UCHL1, BioLegend, catalog no. 304222); anti-CD45RA PE Cy7 (clone HI100, BioLegend, catalog no. 304126); anti-CCR7 PE (clone G043H7, BioLegend, catalog no. 353204); anti-CD95 PE-Dazzle 594 (clone DX2, BD Biosciences, catalog no. 305633); anti-CD154 BV711 (clone 24-31, BioLegend, catalog no. 310837); anti-CD69 BV421 (clone FN50, BioLegend, catalog no. 310930); anti-CD4 APC-H7 (clone L200, BD Biosciences, catalog no. 560837); anti-TCRa/b APC (clone IP26, BioLegend, catalog no. 306718); anti-CD62L BV785 (clone DREG-56, BioLegend, catalog no. 304830); anti-CD8 BV650 (clone RPA-T8, BioLegend, catalog no. 301042); anti-CD3 BV421 (clone SP34-2, BD Biosciences, catalog no. 562877); anti-CD95 BUV395 (clone DX2, BD Biosciences, catalog no. BDB740306); LIVE/DEAD™ Fixable Aqua Dead Cell Stain (ThermoFisher, catalog no. L34957); BioTracker 488 Green CSFE (SigmaAldrich, catalog no. SCT110)

| Validation | All antibodies were purchased from commercial suppliers (BD Biosciences, BioLegend, Invitrogen, and Beckman Coulter). The manufacturers state that these antibodies are research use only (ROU) and have been tested for flow cytometry application using human samples. Fluorescence minus one experiments were performed and and antibody titers were validated experimentally prior to performing the flow cytometry experiments in this study to determine the optimal staining volumes.. Gating strategies can be found in Extended Data Fig. 1, 2, 4, and 6. |
|---|---|

# Flow Cytometry

## Plots

Confirm that:

☒ The axis labels state the marker and fluorochrome used (e.g. CD4-FITC).

☒ The axis scales are clearly visible. Include numbers along axes only for bottom left plot of group (a 'group' is an analysis of identical markers).

☒ All plots are contour plots with outliers or pseudocolor plots.

☒ A numerical value for number of cells or percentage (with statistics) is provided.

## Methodology

| Sample preparation | Human peripheral blood mononuclear cells (PBMCs) were isolated from whole blood by Ficoll-Hypaque density centrifugation and cryopreserved until use. Cryopreserved PBMC were thawed in warm, sterile-filtered RPMI 1640 (Gibco, Waltham, MA) supplemented with 10% fetal bovine serum (FBS) (HyClone, Logan, UT) and 2 µL/mL Benzonase (Millipore, |
|---|---|

Burlington, MA). Centrifugation was performed at 300xg for 10 minutes. The cells were enumerated using the Guava easyCyte (Millipore, Burlington, MA) with guavaSoft 2.6 software and centrifuged again at 300xg for 10 minutes. The cells were then resuspended in a 50 mL conical at a density of 2 × 106 cells/394 mL in RPMI/10% FBS with caps loosely secured and allowed to rest overnight at 37°C/5% CO2. The following day, the cells were enumerated using the Guava easyCyte and resuspended at a density of 5 × 106 cells/mL. To observe ICS following antigen stimulation, approximately 1 × 106 cells/well were plated into a 96-well U-bottom plate and stimulated in the presence of ESAT-6/CFP-10 peptide pool (final concentration of 1 μg/mL of each peptide, BEI Resources), 100 μgml–1 M.tb whole cell lysate (H37Rv, BEI Resources), DMSO (0.5% for the endemic controls and 0.18% for the household contacts) (Sigma, St. Louis, MO). In addition to antigen, each stimulation cocktail consisted of 1 μg/mL anti-CD28/49d (BD Biosciences, San Jose, CA), 10 μg/mL Brefeldin A (Sigma, St. Louis, MO), and GolgiStop (BD Biosciences, San Jose, CA) prepared according to manufacturer's instructions was added to each sample. Cells were incubated at 37oC/5% CO2. EDTA (Thermo Fisher Scientific, Waltham, MA) was then added to disaggregate cells at a final concentration of 2 mM. Samples were stored overnight at 4C and stained the following day.

| Instrument | A BD LSRFortessa was used to acquire the ICS data, equipped with a high-throughput sampler and configured with blue (488 nm), green (532 nm), red (628 nm), violet (405 nm) and ultraviolet (355 nm) lasers using standardized good clinical laboratory practice procedures to minimize the variability of data generated. The index-sorting data were collected using BD FACSDiva 8.0.1. |

| Software | The index-sort data were analyzed using the CATALYST package in R (v4.2.1). ICS data were compensated and gated using FlowJo (v9.9.6) (BD Biosciences, San Jose, CA). Representative gating trees of the low exposure cohort and the household contact cohort are shown in Extended Data Fig. 1, 2, 4, and 6. The data were then processed using the OpenCyto framework in the R programming environment. With the data from the endemic controls, Combinatorial Polyfunctionality Analysis of Antigen-Specific T Cell Subsets (COMPASS) was used to achieve a comprehensive and unbiased analysis of the activation profiles of antigen-specific T cells. The code to complete the validation flow cytometry analyses can be found at https://github.com/seshadrilab/rstr-ins-validation. |

| Cell population abundance | In the index-sort data, the median frequencies of activated (CD69+/CD154+ and CD69+/CD137+) T cells in RSTR and LTBI samples were 0.30% (range 0.21% - 0.35%) and 0.46% (range 0.27% - 0.89%), respectively. |

| Gating strategy | Flow cytometry gating strategies can be seen in Extended Data Fig. 1-2 and 4-6.

For the low exposure cohort ICS experiments, gating began with viable CD3+ cells to identify T cells, followed by a singlet gate (FSC-A vs. FSC-H) and an IL-4/5/13 keeper gate to clean up events. CD4 T cells were identified, and cytokine and activation marker expressions were visualized by gating against IFN-γ.

For the index-sort in the household contact cohort, a lymphocyte gate (FSC-A vs. SSC-A) and a singlet gate (FSC-A vs. FSC-H) were applied. T cells were identified as CD3+/CD14-/CD19-. T cells positive for the TCRαβ marker were gated on and separated into CD4 or CD8 subsets. Activated cells were gated on using CD69, and sorting of activated CD4 and CD8 T cells was done using CD137 and CD154 markers, respectively. Gating of activation and functional markers were determined by DMSO controls.

For the Treg panel in the validation ICS experiments, a time gate (Time vs. FSC-A) was applied to exclude events affected by sample acquisition aberrations, followed by T cell identification by lymphocyte size (FSC-A vs. SSC-A), CD3, CD14, and CD19 markers. A singlet gate (FSC-A vs. FSC-H) was then applied, followed by the identification of viable cells and a second lymphocyte gate (FSC-A vs. SSC-A). CD4 T cells were identified, and positive populations for Treg phenotypic, functional, and activation makers were determined by DMSO controls.

For the Th panel in the validation ICS experiments, the gating strategy from the time gate (Time vs. FSC-A) to the second lymphocyte gate (FSC-A vs. SSC-A) was the same as the Treg panel gating strategy. CD4 T cells were then identified, and positive populations for Th1 and Th17 phenotypic, functional, and activation makers were determined by DMSO controls. A secondary gate was applied to identify IL-17A+ T cells due to spillover spreading. |

☒ Tick this box to confirm that a figure exemplifying the gating strategy is provided in the Supplementary Information.

