## [Peer Review File · Nature Immunology]

Peer Review Information

Journal: Nature Immunology

Manuscript Title: T cell phenotypes in M. tuberculosis 'resisters' are associated with bacterial control

Corresponding author name(s): Dr Chetan Seshadri; Dr Mark Davis

Editorial Notes:

Redactions – transferred manuscripts (mention of the other journal) This manuscript has been previously reviewed at another journal. This document only contains reviewer comments, rebuttal and decision letters for versions considered at Nature Immunology.

Reviewer Comments & Decisions:

Decision Letter, initial version:
--

11th Aug 2023

Dear Dr. Seshadri,

Thank you for your response to the referees' comments on your article "T cell signatures of bacterial control among M. tuberculosis 'resisters'". While we find your work of potential interest, the reviewers have raised substantial concerns that must be addressed. As such, we cannot accept the current version of the manuscript for publication, but would be happy to consider a revised version that addresses these concerns, as long as novelty is not compromised in the interim.

Please revise the manuscript to address all issues raised by the referees and according to your response. In addition please revise the manuscript to clearly define the cohorts and discriminate the various patients and samples (as requested by the referees), with a special attention to avoiding assumptions about their status. As such, we recommend designation based on experimental readouts, for example Xtest+ Ytest+, rather than assumed status (resistant, latent). At resubmission, please include a point-by-point "Response to referees" detailing how you have addressed each referee comment (please specify page and figure number where the new data can be found in the revised manuscript). This response will be sent back to the referees along with the revised manuscript.

In addition, please include a revised version of any required reporting checklist. It will be available to

referees (and, potentially, statisticians) to aid in their evaluation if the manuscript goes back for peer review. A revised checklist is essential for re-review of the paper.

The Reporting Summary can be found here:

When submitting the revised version of your manuscript, please pay close attention to our [href="https://www.nature.com/nature-portfolio/editorial-policies/image-integrity">Digital Image Integrity Guidelines](https://www.nature.com/nature-portfolio/editorial-policies/image-integrity). and to the following points below:

We hope to receive a suitably revised manuscript within 6 months. If you cannot send it within this time, please let us know. We will be happy to consider your revision so long as nothing similar has been accepted for publication at Nature Immunology or published elsewhere.

Nature Immunology is committed to improving transparency in authorship. As part of our efforts in this direction, we are now requesting that all authors identified as 'corresponding author' on published papers create and link their Open Researcher and Contributor Identifier (ORCID) with their account on the Manuscript Tracking System (MTS), prior to acceptance. ORCID helps the scientific community achieve unambiguous attribution of all scholarly contributions. You can create and link your ORCID from the home page of the MTS by clicking on 'Modify my Springer Nature account'. For more information please visit www.springernature.com/orcid.

Thank you for the opportunity to review your work.

Sincerely,

Ioana Visan, Ph.D.

Senior Editor
Nature Immunology

Tel: 212-726-9207
Fax: 212-696-9752
www.nature.com/ni

Reviewers' Comments:

Reviewer #1:

Remarks to the Author:

This study analyzed individuals with a high probability of exposure to *M. tuberculosis* (*M.tb*) who appear to 'resist' infection. While these 'resisters' (RSTR) display IFN- γ -independent T cell responses to the *M.tb*-specific antigens ESAT-6 and CFP-10, it is currently unknown whether specific T cell functional programs are associated with this clinical outcome. The data presented outlines results from 3 different human cohorts and a NHP cohort to determine genes and cell pathways involved in this 'protection'. The authors found that *M.tb*-specific T cells derived from RSTRs showed an early differentiation phenotype as well as enrichment of Th17-like transcriptional programs compared to LTBI, which were characterized by Th1*-like effector programs. The paper was interesting, very well written and the data presented built on nicely from previous research from this group. The inclusion of community controls, and comparison with other models, provides valuable information on potential mechanisms of protection in this unique cohort.

Outlined below are some minor concerns/points for clarity:

Introduction:

1. You mention other cohorts of 'resisters' but these are all quite different from your long-term resister cohort. I would suggest differentiating between 'early clearance' and resistance mechanisms as they are likely to involve different cell types.
2. You mention the SA cohort were long-term non-progressors but they were non-progressors from disease, not infection. This needs to be clarified as it's very likely different mechanisms will be at play.

Methods:

1. How did you determine the community controls were low exposure?
2. Why did you focus solely on antigen-specific T cells and not B cells etc?
3. Why did you use peptide pools rather than WCL for eg. Or Mtb300 – this would allow a much broader response to be analysed.

Results:

1. In all these studies, analysis is done only after classification is known. It would be good to know if there is prognostic risk for development of LTBI as opposed to simply analyzing the differences once converted. It would be good to determine the cell types in individuals at baseline time-point ie in those who convert and those who don't convert prior to any evidence of infection. This would indicate if the same cell types are dominant in RSTR at future time-points.
2. For the HIV infected individuals, were they on ART? What were their CD4 counts and what were their QFT and TST readings?

3. Do you think protection from disease progression (ie the ACS cohort) would require the same cell types? Given progressors are infected then I would hypothesise that different mechanisms are required, and likely involve both IFN-dependent and independent responses.
4. Suggest looking at ratio of Treg to other cell types as it's normally the overall response with a balance of pro and anti-inflammatory that determines outcome.
5. Could the transcriptional profiles have changed following sorting (and thus activation)?
6. You say cluster 6 'appears to be' enriched. Please be more precise as to whether it was or not...
7. Interesting results for RSTR in terms of clonal expansion. Perhaps this lack of skewing to a certain type of response may aid protection depending on which antigen is presented?
8. Results for higher levels of naïve phenotype in RSTR are not significant, particularly if you were to adjust for multiple comparisons. Please change wording to 'shows a trend'.
9. Which time-points were analyzed for luminex? Could these same time-points be analysed using PBMC in future studies? Note, I am not suggesting further analysis for this manuscript.
10. The NHP data are very interesting. What about data from iv BCG? Do those animals who were protected show similar gene profiles? That data should be publicly available.

Discussion:

This was really short and perhaps didn't provide enough detail on the limitations of each model. The South African cohort were not resistant to infection and whilst the data provides interesting insight into protection from disease, it's not necessarily the same mechanisms/cell types at play. In addition, adolescents will be closer to the time of BCG vaccination and likely will have some influence on protection from infection due to vaccine status.

I think the RSTR cohort is unique in that it is very long term after exposure and therefore would benefit from more discussion on this long-term protection versus protection at early stages of exposure versus protection from disease. In addition, the obvious omission in human studies is lack of analysis at the site of infection and this limitation should also be discussed in relation to the NHP findings.

Methods:

1. Why were different staining methods used for the endemic controls and the HHC? Are the results generated therefore comparable?
2. Please explain why no multiple comparison corrections were used?

Reviewer #2:

Remarks to the Author:

Sun et al set out to understand early events of Mtb infection with the main question been if resisters have any unique anti-bacterial functional programs relative to latently infected persons. A long outstanding question in the field of TB is understanding the phenomenon of resisters and whether failure to convert TST or IGRA has a biological association with Mtb clearance. They aimed to measure IFN γ -independent T cell signatures specific to household exposure, and to determine genes selectively enriched among Mtb-specific T cells derived from resisters compared to controls. This will fill in knowledge gaps about whether resisters have actual protective immunity or failed to convert TST/IGRAs in addition to the known Epi data. They observed a role of Th17 cells and lower polyfunctionality scores in TST-/IGRA relative to LTBI, in addition, they showed important information on T cell quality which is novel. They used other cohorts to confirm this and non-human primates. The

manuscript is well-written, experiments were well described as well as the results and conclusions which were precise and concise. The study is important and would have a major impact on the field, however, some of the claims are not confirmed resulting in several limitations. I'm requesting for a major revision due to importance of the paper in the field.

Major limitations

The study is important and will have a major impact on the field, however these points should be addressed before the paper can be considered for publication in Nature Medicine.

- The authors claim that the observed Th17 phenotype (a known correlate of protection both in human and animal studies); they show is reflective of bacterial burden. However, they have not proven this. I would recommend they measure bacterial burden in the study cohorts to confirm this claim. How do the bacterial burden correlate with the responses observed?
- TB has a diverse set of antigens, the authors used only two antigens, ESAT-6 and CFP which are used in IGRAs. The study participants were classified based on IGRAs and TSTs, and persons who were IGRA negative were classified as resisters. IGRAs measure IFN γ responses from T cells, thus their observed results that the quality of IFN γ response in resisters is low is expected. Is the response observed after ESAT⁶ and CFP10 stimulation the same as in other TB antigens such as MTB300? I'm recommending that the authors test other TB antigens in the same way as ESAT-6/CFP10 to confirm their results.
- The resistor/LTBI cohort used for targeted sequencing is small. Sample size should be increased to increase the statistical observations made in Figure 2 after index sorting and targeted sequencing.
- SELECT-Seq was used to characterize responses observed in the targeted sequencing, cells were sorted on TCR α and activation induced markers. I recommend showing data from the TCR observed in the study to provide more information on the diversity of the T cells involved in protection.
- The authors claim that the T cells from the resisters are more stem cell naïve-like based on expression of a few stem naïve genes. This has not been functionally validated and should be validate via flow using naïve like stem cell markers.
- There are no mechanistic studies which are important in comprehending the underlying mechanisms for T cell protection in TB disease. Proliferative responses of clonally expanded cells are needed to show functional ability of the cells after stimulation.

Minor comment

- The study was validated in a cohort of adolescents based on gene expression data, which is excellent. Cellular data is also available from the same cohort (Scriba et al., 2017, Plos Pathogens) and this should be analyzed to confirm the T cell responses observed in the resisters.

[REDACTED]

Author Rebuttal to Initial comments

See inserted PDF

Point-by-Point Response to Reviewer Comments

Reviewer #1:

This study analyzed individuals with a high probability of exposure to *M. tuberculosis* (*M.tb*) who appear to 'resist' infection. While these 'resisters' (RSTR) display IFN- γ -independent T cell responses to the *M.tb*-specific antigens ESAT-6 and CFP-10, it is currently unknown whether specific T cell functional programs are associated with this clinical outcome. The data presented outlines results from 3 different human cohorts and a NHP cohort to determine genes and cell pathways involved in this 'protection'. The authors found that *M.tb*-specific T cells derived from RSTRs showed an early differentiation phenotype as well as enrichment of Th17-like transcriptional programs compared to LTBI, which were characterized by Th1*-like effector programs. The paper was interesting, very well written and the data presented built on nicely from previous research from this group. The inclusion of community controls, and comparison with other models, provides valuable information on potential mechanisms of protection in this unique cohort.

We appreciate this positive assessment of our work.

Outlined below are some minor concerns/points for clarity:

Introduction:

1. You mention other cohorts of 'resisters' but these are all quite different from your long-term resister cohort. I would suggest differentiating between 'early clearance' and resistance mechanisms as they are likely to involve different cell types.

This is an excellent suggestion. We have revised the Introduction to more clearly communicate the differences among these cohorts.

Lines 81-85: Other studies have also reported highly exposed individuals who demonstrate negative TST and/or IGRA among healthcare workers, miners, and household contacts⁸. *The consistency of these reports is striking despite variations in timing and strength of exposure, duration of follow-up, and frequency of testing*^{7, 9}. Whether these individuals have controlled *M.tb* infection is unknown.

Lines 91-92: However, T cells likely play a role in mediating 'resistance' to established *M.tb* infection, *in particular in long-term 'resistance' to established M.tb infection*^{11, 12}.

Lines 424-427: Though our experiments focused on characterizing T cell phenotypes *in long-term resistance*, our results are not inconsistent with a model in which *M.tb* is recognized and eliminated without the assistance of T cells *at early stages of exposure and protection from disease*.

2. You mention the SA cohort were long-term non-progressors but they were non-progressors from disease, not infection. This needs to be clarified as it's very likely different mechanisms will be at play.

In response to this comment, we have extensively revised the manuscript to reflect the associations between T cell phenotypes and the full spectrum of human tuberculosis, ranging from 'resistance' to infection and non-

progression or progression to active disease. It is in this context that comparisons with the South African cohort are most informative. We have made the following changes to the text:

Fig 7 Legend Title. Lines 913-914: **Gene programs enriched in RSTRs were associated with lack of progression to active TB in South African adolescents as well as bacterial control in non-human primates.**

Abstract Lines 64-69: *Th17-like functional programs were also associated with a lack of progression to active TB among South African adolescents with LTBI, as well as bacterial control in published non-human primate studies.* Together, these data suggest that 'resisters' may successfully control *M.tb* after exposure and immune priming and establish a set of T cell biomarkers to facilitate further study of this important clinical phenotype.

Introduction Lines 86-90: Early events after *M.tb* infection are incompletely modeled and poorly understood. Most animal models seek to recapitulate active disease, so the immune mechanisms underlying protection from *M.tb* infection rather than disease are currently undefined¹⁰. *In addition, there is increasing appreciation that human TB occurs along an immunologic spectrum with overt disease on one end, 'resistance' on the other end, and asymptomatic infection and disease in between².*

Lines 123-129: These included Th17-like as well as an early differentiation memory phenotype that were validated in an independent set of samples using flow cytometry and multiplex cytokine analysis. *We also show that these T cell phenotypes are enriched within a published cohort of South African adolescents who fail to progress to active disease relative to those who do. Finally, we provide additional context for our results using published non-human primate studies examining the natural history of M.tb infection or the protective efficacy of intravenous BCG.*

Results Line 353: Gene programs enriched in RSTRs are also associated with long-term LTBI non-progressors

Line 377-379: Together, these data reveal the relative enrichment of early differentiation, T-regulatory, and Th17-like phenotypes among a cohort of adolescents *with LTBI* who appear to resist progression to active TB disease.

Methods:

3. How did you determine the community controls were low exposure?

This was known from prior epidemiologic profiling of the district . We have revised the text and provided the reference.

Lines 497-499: *Of the five divisions of Kampala, a low TB incidence division was identified by the Kampala Capital City Authority based on low TB transmission rates²⁷.*

4. Why did you focus solely on antigen-specific T cells and not B cells etc.?

Indeed, this is the subject of ongoing work. In a recent study of South African gold miners, we reported that *M.tb* specific antibodies from 'resisters' harbored unique Fc galactosylation and sialylation profiles (PMID: 37379655).

However, the current work is meant to directly follow up our original study which reported IFN- γ -independent T cell responses in both RSTR and LTBI subjects (PMID: 31110348). We have added the following text in the Introduction to clarify this:

Lines 113-118: Consistent with this literature, we have previously shown that nearly all Ugandan household contacts with a high probability of *M.tb* exposure display IFN- γ independent T cell responses to *M.tb*-specific antigens, including IL-2, TNF, and CD154⁷. *In a study of South African miners, we also showed that RSTRs harbored M.tb-specific antibodies with unique Fc receptor profiles*²⁶. However, it is currently unknown whether *M.tb*-specific T cells from RSTRs harbor any unique anti-bacterial T cell functional programs compared to LTBI controls.

5. Why did you use peptide pools rather than WCL for e.g. Or Mtb300 – this would allow a much broader response to be analysed.

We initially focused on characterizing T cell responses to ESAT-6 and CFP-10 because those antigens form the bedrock of clinical TB diagnosis and are incorporated into multiple commercial assays. However, in response to this critique as well as that of Reviewers 2 and 3, we generated new data using *M.tb* whole cell lysate, which contains many more antigens including several that are not specific for *M.tb*. Specifically, we conducted a follow up experiment in which we analyzed a discovery cohort of RSTRs (n=16) and LTBI (n=16) using index sorting and targeted transcriptional profiling followed by a validation cohort of RSTRs (n=17) and LTBI (n=20) using flow cytometry and Luminex after whole *M.tb* lysate stimulation (see revised Fig 1a study schema below). Overall, we did not observe qualitative differences in the transcriptional profiles of CD4 T cells between the two groups. However, we did replicate the association between Th17-like and T-regulatory T cells and RSTRs seen with ESAT-6/CFP-10 stimulation. The data have been incorporated into new Figure 6 and associated Results text, which are copied below in full.

Lines 767-774:

Fig. 1a study schema. We analyzed endemic controls with LTBI (n = 19) and matched TST-/IGRA- (n = 17) participants, as well as RSTR (n = 17) and LTBI (n = 20) subjects enrolled through our household contact study. The functional diversity of T cells specific for ESAT-6/CFP-10 or whole M.tb lysate was assessed by intracellular cytokine staining (ICS), targeted transcriptional profiling, and SELECT-Seq. Results were validated experimentally in an independent set of RSTRs (n = 17) and LTBI (n = 20) using flow cytometry and Luminex.

Lines 890-911:

Fig. 6

Fig. 6 T cells activated by M.tb lysate exhibit phenotypes similar to those specific to ESAT-6/CFP-10. (a) Expression of RORγt and T-bet among M.tb-specific CD4 T cells as defined by CD154+ and/or CD137+ expression after M.tb lysate stimulation. Representative staining from one RSTR and one LTBI donor from the validation cohort with M.tb lysate stimulation is shown. (b) Frequencies of T cells expressing RORγt and/or T-bet among M.tb lysate-specific CD4 T cells in the validation cohort after M.tb lysate stimulation. (c-e) Frequencies of T cells expressing a Th1* phenotype (CXCR3+CCR6+) (c), IFN-γ (d), or IL-17A (e) among M.tb lysate-specific CD4 T cells in the validation cohort after M.tb lysate stimulation. (f-g) Background-corrected mean fluorescence intensity (MFI) of IL-17A (f) and IL-23 (g) (open circles) in conditioned supernatants as determined by multiplex cytokine analysis at 24 hours and 6 hours, respectively, post-stimulation with whole M.tb lysate. (h) Expression of Treg markers (Foxp3 and CD25) among M.tb-specific CD4 T cells as defined by CD154+ and/or CD137+ expression after M.tb lysate stimulation. Representative staining from one RSTR and one LTBI donor from the validation cohort is shown. (i) Frequencies T cells expressing a Treg phenotype (Foxp3+CD25+) among total CD4 T cells (left) or of M.tb-specific CD4 T cells (right) in the validation cohort, following stimulation with M.tb lysate. (j) Frequencies of T cells expressing CD25 among M.tb lysate-specific CD4 T cells stratified by group in

the validation cohort. **(k)** Background-corrected mean fluorescence intensity (MFI) of IL-10 (open circles) in conditioned supernatants as determined by multiplex cytokine analysis at 12 hours post-stimulation with whole *M.tb* lysate. Statistical significance in **(b-e)** and **(i-k)** was determined by Wilcoxon rank-sum tests, and unadjusted *p*-values are shown. Statistical significance in **(f-g)** and **(k)** was determined by the Student's *t*-test.

Lines 318-351:

T cells activated by M.tb lysate exhibit phenotypes similar to those seen with ESAT-6/CFP-10

Having established the relative enrichment of several ESAT-6/CFP-10 specific T cell phenotypes among RSTRs compared to LTBI, we next sought to determine whether the same phenotypes were enriched after stimulation with *M.tb* whole cell lysate, which contains a broader set of antigens that are conserved across mycobacteria. Compared to experiments with ESAT-6/CFP-10 stimulation, we examined a larger discovery cohort of RSTRs ($n = 16$) and LTBI ($n = 16$) using index sorting and targeted transcriptional profiling. Overall, we did not observe qualitative differences in CD4 T cell phenotypes between the two groups (Supplementary Fig. 7a). Nevertheless, we attempted to replicate our original findings in an independent set of samples (RSTRs $n = 17$, LTBI $n = 20$) with flow cytometry and Luminex (Fig. 1A, Supplementary Fig. 8, and Table 2). T cells responsive to *M.tb* lysate stimulation that expressed an early differentiation phenotype were present at similar frequencies in RSTRs and LTBI (Supplementary Fig. 6b). However, ROR γ t+T-bet+ T cells were present at higher frequencies among LTBI subjects and ROR γ t+T-bet- T cells were higher among RSTRs, analogous to what we observed with ESAT-6/CFP-10 stimulation ($p=0.039$ and $p=0.011$, respectively) (Fig. 6a and 6b). These cells were also characterized by co-expression of CCR6 and CXCR3, supporting their designation as Th1* T cells (Fig. 6c). As expected, IFN- γ production after *M.tb* lysate stimulation was higher among LTBI subjects when measured by intracellular cytokine staining or Luminex ($p < 0.001$ and $p < 0.001$, respectively) (Fig. 6d). IL-17A-expressing CD4 T cells also trended higher in RSTRs; however, this was not statistically significant (Fig. 6e and 6f). Notably, the Th17-promoting cytokine IL-23 was significantly higher in conditioned supernatants from RSTRs compared to LTBI ($p=0.041$) (Fig. 6g).

We next examined the frequency of *M.tb* lysate-responsive regulatory T cells as defined by the expression of CD25 or co-expression of FoxP3 and CD25. We observed a higher frequency of both CD25+ and FoxP3+CD25+ T cells after *M.tb* lysate among RSTRs compared to LTBI ($p=0.052$ and $p=0.013$, respectively) (Fig. 6h, 6i, and 6j). In addition, IL-10 concentrations in conditioned supernatants trended higher, though this was not statistically significant (Fig. 6k). The ratio of the anti-inflammatory regulatory T cell fraction to the pro-inflammatory T cell fraction, including Th1, Th17, and Th1* cells, was significantly higher in RSTRs compared to LTBI (Supplementary Fig. 7c). Taken together, these data show that enrichment of Th17-like and T-regulatory functional programs observed among RSTRs after stimulation with ESAT-6 and CFP-10 are also recapitulated with a broader set of antigens present in *M.tb* lysate.

Results:

6. In all these studies, analysis is done only after classification is known. It would be good to know if there is prognostic risk for development of LTBI as opposed to simply analyzing the differences once converted. It would be good to determine the cell types in individuals at baseline time-point i.e. in those

who convert and those who don't convert prior to any evidence of infection. This would indicate if the same cell types are dominant in RSTR at future time-points.

We appreciate this comment, and addressing this question is indeed the focus of current work which requires enrolling a completely independent cohort and is beyond the scope of the current manuscript. Dr. Boom is leading those efforts as PI of our IMPAc-TB Consortium but we will not have results for several years. Nevertheless, the current study is still extremely valuable for the field as acknowledged by this Reviewer because it builds on a cohort that was established and followed for nearly 10 years with clear clinical outcomes.

7. For the HIV infected individuals, were they on ART? What were their CD4 counts and what were their QFT and TST readings?

We apologize for the confusion. There were no HIV-infected persons included in our study. We have clarified this in the Methods.

Line 502: We were able to classify 196 *HIV-uninfected* individuals based on TST and QFT concordance.

Lines 522-524: PBMC from a subset of these definite LTBI controls and RSTRs were used for the present study and selected after matching for age, sex, exposure risk score, and documented lack of HIV co-infection.

8. Do you think protection from disease progression (i.e. the ACS cohort) would require the same cell types? Given progressors are infected then I would hypothesise that different mechanisms are required, and likely involve both IFN-dependent and independent responses.

We agree with this comment as well as that from R3 and feel that we can do a better job of comparing and contrasting the associations we report in the two human cohorts. We do acknowledge the difference but want to address the similarity in the IFN- γ -independent response among different cohorts.

We have edited the Discussion section as follows:

Lines 446-456: *There is emerging evidence that IFN- γ dependent T-cell immunity in humans may be a reliable proxy for established infection with *M.tb*. IFN- γ concentrations derived from IGRAs are associated with progression to active TB^{70, 71}. In addition, there is a progressive increase in *M.tb*-specific IFN- γ dependent CD4 T cell responses across the spectrum of IGRA non-converters, reverters, and persistent IGRA+ South African adolescents⁷². The clinical relevance of IFN- γ independent T-cell profiles has not been thoroughly explored. Our data suggest that IL-17 or IL-23 production in the absence of IFN- γ after stimulation with ESAT-6/CFP-10 may also have clinical utility. Expanded analysis of IGRA supernatants has already revealed host biomarkers that distinguish latent from active TB⁷³. As IL-17, IL-21, and IL-23 concentrations are expected to be very low after peptide stimulation, more sensitive assays may be required to realize their full diagnostic potential.*

Lines 482-489: *Surprisingly, we also found an expansion of this Th17-like transcriptional program among a subset of South African adolescents with established LTBI that fail to progress to active TB relative to progressors. Thus, it appears that Th17-like T cells may mediate protection across the spectrum of *M.tb* infection and disease and across species. At a minimum, it supports the notion that the current definition of 'LTBI' is*

heterogeneous and consists of individuals at risk and protected from *M.tb* disease. Future work may eventually render the terms 'RSTR' and 'LTBI' obsolete because they are defined by IFN- γ dependent immunity and replace them with Th17-based, more clinically informative definitions.

9. Suggest looking at ratio of Treg to other cell types as it's normally the overall response with a balance of pro and anti-inflammatory that determines outcome.

This is an excellent suggestion. We performed the analysis as suggested by the reviewer. Among *M.tb*-specific T cells, we found that the ratio of Treg to Th1 and Th17 cells was higher among RSTRs than LTBI subjects. These data have been incorporated into a new Supplementary Figure 7 and we have added the following text to the Results and Discussion:

In the Supplementary Figures PDF:

Supplementary Fig. 7c. Ratio of anti-inflammatory regulatory CD4 T cell fraction to pro-inflammatory CD4 T cell fraction, including Th1, Th17, and Th1* cells, among RSTR and LTBI donors in response to ESAT-6/CFP-10 or *M.tb* lysate stimulation.

Lines 345-349: In addition, IL-10 concentrations in conditioned supernatants trended higher, though this was not statistically significant (Fig. 6k). *The ratio of the anti-inflammatory regulatory T cell fraction to the pro-inflammatory T cell fraction, including Th1, Th17, and Th1* cells, was significantly higher in RSTRs compared to LTBI (Supplementary Fig. 7c).*

10. Could the transcriptional profiles have changed following sorting (and thus activation)?

We appreciate this point but note that we and others have successfully employed this method to successfully identify rare antigen-specific T cells in previously published work (PMID: 32341563, 30992377, 36604540). Per the Methods, the stimulation time was six hours while the time required for sorting was less than 1 hour. Cells were directly sorted into lysis buffers to capture their functional profiles without additional manipulation. We have added the following text to the Methods to clarify this point:

Lines 547-549: *We conducted the SELECT-seq protocol, which includes stimulation and sorting procedures, as described in several recent publications^{31, 55, 79}. In brief, PBMCs from RSTR and LTBI samples were stimulated as described in the ICS methods above.*

11. You say cluster 6 'appears to be' enriched. Please be more precise as to whether it was or not...

We apologize for the confusion. We have clarified this point in the Results.

Lines 187-190: T-distributed stochastic neighbor embedding (T-SNE) visualization revealed that some of these subsets might be uniquely associated with RSTR or LTBI status (Fig. 2c and Supplementary Fig. 3a). For example, cluster 6, expressing *RORC*, *RUNX3*, and *IL17A*, was enriched among RSTRs.

12. Interesting results for RSTR in terms of clonal expansion. Perhaps this lack of skewing to a certain type of response may aid protection depending on which antigen is presented?

Indeed, this is one possible hypothesis and the topic for future work. We have expanded upon this point in the Discussion.

Line 458-468: *Our initial experimental approach was focused on defining the functions of T cells targeting ESAT-6 and CFP-10 because these are specific for M.tb and incorporated into IGRAs. We extended our approach to include M.tb lysate, which contains many more antigens, but at the expense of specificity for M.tb. Nevertheless, we found concordant results when examining Th17-like and T-regulatory T cell phenotypes. In a study comparing M.tb-infected mice and BCG-vaccinated humans, M.tb infection drove ESAT-6 specific T cells to become more differentiated than Ag85B-specific T cells⁷⁴. This was consistent with the observation that M.tb restricts expression of Ag85B but not ESAT-6 during chronic infection^{75,76}. Musvosvi et al. also recently reported preferential targeting of PE13 and CFP-10 by non-progressors compared to progressors⁵⁵. Similarly, RSTRs may preferentially target certain mycobacterial antigens when compared to LTBI, which will be the subject of future work.*

13. Results for higher levels of naïve phenotype in RSTR are not significant, particularly if you were to adjust for multiple comparisons. Please change wording to 'shows a trend'.

We made the change as requested to the text. In addition, we generated new data to demonstrate the functionality of stem cell memory T cells in both RSTRs and LTBI (see response to Reviewer #2, Comment #5 below).

Lines 225-227: *We observed a trend* toward a higher proportion of *M.tb*-specific T cells expressing a naïve-like phenotype (CD45RA+CCR7+) among RSTRs when compared to LTBI (16.85% vs. 13.21%, $p = 0.069$) (Fig. 3f).

Line 239-242: At the time of sorting, we observed a significantly higher proportion of naïve-like T cells co-expressing the T_{SCM} marker CD95+ cells among LTBI compared to RSTR subjects regardless of stimulation (Supplementary Fig. 5d, *M.tb* lysate: $p = 0.048$; ESAT-6/CFP-10: $p = 0.013$).

14. Which time-points were analyzed for Luminex? Could these same time-points be analysed using PBMC in future studies? Note, I am not suggesting further analysis for this manuscript.

We apologize for the confusion. We have clarified the choice of these time points in the Methods and highlight the specific ones displayed in the figure legends (Figure 4J-K and Figure 5J).

Lines 691-699: Background-corrected signals were computed by subtracting the signal from the DMSO condition from the antigen stimulation condition. *To choose the precise time points with which to compare protein secretion (measured by Luminex and ELISA) between the RSTR and LTBI groups, we identified samples exhibiting the most favorable signal-to-noise ratio in response to stimulation (ESAT-6/CFP-10 or M.tb lysate) versus DMSO. This selection was based on applying the T-statistic at each time point assessed. Subsequently, statistical testing between the two groups was performed on a single time point for each analyte using the Students' t-test (Supplementary Fig. 6 and Supplementary Fig. 8).*

15. The NHP data are very interesting. What about data from iv BCG? Do those animals who were protected show similar gene profiles? That data should be publicly available.

This is an excellent suggestion. Around the time this manuscript was originally submitted for review, Liu et al. analyzed whole blood transcriptional profiles in a dose-ranging study of IV BCG in rhesus macaques (Liu et al. Cell Reports Medicine 2023, PMID: 37390827). They identified early innate transcriptomic response as immune correlates of protection in this setting. A parallel study by Darrah et al. examined T cell phenotypes in airway and blood (Darrah et al. Cell Host Microbe 2023, PMID: 37267955). This study identified the combined immune feature of frequency of CD4 T cells producing TNF with IFN- γ , frequency of those producing TNF with IL17, and the number of NK cells as immune correlates of protection.

To directly address this critique, we collaborated with Yiran Liu and Purvesh Khatri (lead authors of PMID: 37390827) to examine whether T cell phenotypes that we observed as associated with RSTR status were also associated with protection after IV BCG. Indeed, we found that a Th17 summary score derived from data reported here was also associated with protection (revised Fig 7i). We also collaborated with Dr. Changqi Wang and Dr. Doug Lauffenberger (authors of PMID: 37267955) to examine whether IFN- γ independent CD4 T cell profiles were associated with protection in the Darrah et al. study. Indeed, we found that CD4 T cells producing IL-17 but not IFN- γ , TNF, or IL-2 were associated with protection after IV BCG vaccination. These data substantially strengthen the original claims in our paper. We have revised Figure 7 as follows and added the following text to the Methods and Results:

Lines 930-939, Fig 7h and 7i legends: **(h)** Whole blood transcriptional profiles from dose-ranging study of intravenous (IV) BCG of rhesus macaques⁵⁷ were used to calculate the Th17 gene module score based on genes identified in Fig. 4. The score was computed as the geometric mean of gene expressions plotted for protected and non-protected groups according to the total CFU M.tb upon necropsy. Statistical testing was performed

using the Wilcoxon rank-sum test. (i) Flow cytometry data from a parallel study examining airway T cells induced by IV BCG vaccination in rhesus macaques⁵⁸ was used to calculate the cell count and the cell fraction of Th17 cells defined as IFN γ -IL2-IL17+TNF- CD4 subset in BALs. The violin plots depict the difference between the protected and non-protected groups according to the total CFU *M.tb* upon necropsy. Statistical testing was performed using the Wilcoxon rank-sum test.

Lines 394-402: The associated gene module scores were significantly higher in RSTRs compared to LTBI (Fig. 7g). Next, we examined blood transcriptomes and T cell phenotypes in rhesus macaques that were vaccinated with intravenous BCG and subsequently challenged with *M.tb* in an effort to identify correlates of protective immunity^{57, 58}. Expression of a Th17 gene module based on genes enriched in T cells from RSTRs was higher among macaques that were protected against *M.tb* challenge compared to those that were not protected (Fig. 7h). Similarly, the absolute number but not the frequency of purified protein derivative (PPD)-specific IFN- γ independent IL-17+ CD4 T cells in bronchoalveolar lavage was higher among protected macaques (Fig. 7i).

Lines 719-738:

Non-human primate cohort analysis

We derived the gene expression patterns identified in 26 granulomas from four cynomolgus macaques obtained ten weeks after low-dose *M.tb* infection and analyzed using single-cell RNA sequencing⁵⁶. We selected the top 15 enriched genes in each of the stem-like cell subsets and T1-T17 population 1 cell subset and calculated their mean expression in RSTR and LTBI donors using the SELECT-seq dataset. Genes that were not expressed or not found were excluded from the analysis. We used the AddModuleScore function from Seurat⁸⁶ to calculate the associated gene module scores.

We examined the expression of the Th17 gene module in the whole blood transcriptome of 34 rhesus macaques from a dose-ranging study of intravenous (IV) BCG vaccination followed by *M.tb* challenge. Data collection and pre-processing have been described in detail previously⁵⁷. We calculated expression summary scores using the geometric mean of the Th17 module genes identified in Fig. 4. We then compared summary scores across all timepoints between macaques that were protected ($n = 18$) versus not protected ($n = 16$) against *M.tb* challenge, as determined by total *M.tb* CFU upon necropsy, using the Wilcoxon rank-sum test. Flow cytometry was performed on freshly collected bronchoalveolar lavage fluid obtained from these same macaques and detailed methods have been previously published⁵⁸. From these data, we extracted the IL17-monofunctional T cell phenotype measured by IFN γ -IL2-IL17+TNF- among CD4 T cells after PPD stimulation.

Discussion:

16. This was really short and perhaps didn't provide enough detail on the limitations of each model. The South African cohort were not resistant to infection and whilst the data provides interesting insight into protection from disease, it's not necessarily the same mechanisms/cell types at play. In addition, adolescents will be closer to the time of BCG vaccination and likely will have some influence on protection from infection due to vaccine status.

We agree with this point. We have expanded this point in the Discussion and include both strengths and potential limitations in making this connection.

Lines 424-428: Though our experiments focused on characterizing T cell phenotypes *in long-term resistance*, our results are not inconsistent with a model in which *M.tb* is recognized and eliminated without the assistance of T cells *at early stages of exposure and protection from disease*. Alveolar macrophages are among the first airway immune cells to encounter *M.tb* and are generally permissive to *M.tb* growth^{59, 60}...

Lines 446-450: *There is emerging evidence that IFN- γ dependent T-cell immunity in humans may be a reliable proxy for established infection with M.tb. IFN- γ concentrations derived from IGRAs are associated with progression to active TB^{70, 71}. In addition, there is a progressive increase in M.tb-specific IFN- γ dependent CD4 T cell responses across the spectrum of IGRA non-converters, reverters, and persistent IGRA+ South African adolescents⁷².*

Lines 480-487: *We report a higher frequency of ROR γ t+T-bet- M.tb-specific T cells among RSTRs compared to household contacts with established LTBI in Uganda. Surprisingly, we also found an expansion of this Th17-like transcriptional program among a subset of South African adolescents with established LTBI that fail to progress to active TB relative to progressors. Thus, it appears that Th17-like T cells may mediate protection across the spectrum of M.tb infection and disease and across species. At a minimum, it supports the notion that the current definition of 'LTBI' is heterogeneous and consists of individuals at risk and protected from M.tb disease.*

17. I think the RSTR cohort is unique in that it is very long term after exposure and therefore would benefit from more discussion on this long-term protection versus protection at early stages of exposure versus protection from disease. In addition, the obvious omission in human studies is lack of analysis at the site of infection and this limitation should also be discussed in relation to the NHP findings.

We agree with these points and have revised the Discussion to address them as follows:

Line 428-444: *Studies of blood-derived myeloid cells have revealed differences in the transcriptional response between RSTRs and LTBI, which we have reported in this cohort as well as in a parallel cohort of gold miners in South Africa⁶¹...These mechanisms may act in concert to reduce the bacterial/antigen load and tune the inflammatory environment to prime the T cell phenotypes that we observed. Further investigation of M.tb-specific T cells in the lungs of RSTRs will be required to confirm the findings we report here using peripheral blood⁶⁹.*

Methods:

18. Why were different staining methods used for the endemic controls and the HHC? Are the results generated therefore comparable?

We apologize for the confusion. In the interest of transparency, we sought to highlight minor methodologic differences between staining methods used in the two cohorts. Rather than distract the reader, we have elected to consolidate the text.

Lines 537-544: *Intracellular cytokine staining (ICS) was performed on samples from the endemic controls as previously described⁷. The same ICS assay and flow cytometry acquisition method were performed on samples from the household contacts with minor modifications. Prior to staining, samples from the household contacts were divided in half to be analyzed using two multiparameter flow cytometry panels, one for the analysis of Tregs and one for Th subsets (Supplementary Table 2). Cells were permeabilized and underwent intracellular staining*

using the eBioscience Foxp3/Transcription Factor Staining Buffer Set (eBioscience, San Diego, CA) *to allow for the analysis of transcription factors.*

19. Please explain why no multiple comparison corrections were used?

We apologize for the confusion. Broadly speaking, we designed our study with a ‘discovery’ and ‘validation’ phase from the outset. The discovery phase was meant to be hypothesis generating with a high tolerance for false discovery. We identified key hypotheses from the discovery data (targeted transcriptional profiling and SELECT-Seq) based not only on statistical significance but biological rationale to follow up in the validation phase using independent methods (flow cytometry and Luminex) and non-overlapping samples (Fig. 1 schema). We also performed targeted testing of our key hypotheses in external data sets using data derived from both human and non-human primates. (Fig. 7). Thus, our overall approach provides several layers of experimental and biological validation without focusing only on statistical significance. Where multiple features were measured and analyzed simultaneously, we did correct for multiple hypothesis testing using the Bonferroni Method, and this is indicated in the Fig legend (Fig. 1d, 3d, 4b, 5c, etc). We have made the following changes to the Figure legends to make this point more explicitly:

Lines 665-667: *We used Wilcoxon rank-sum tests to compare the cell phenotype subset composition between the two groups and corrected for multiple hypothesis testing using the Bonferroni method.*

Lines 788-789, Fig 1 legend: *Reported p-values in (d) were adjusted for multiple hypothesis testing using the Bonferroni method.*

Lines 823-824, Fig 3 legend: *Statistical testing was performed using the Wilcoxon rank-sum tests with corrections for multiple-hypothesis testing using the Bonferroni method.*

Lines 837-838, Fig 4 legend: *Statistical testing was performed using the Wilcoxon rank-sum tests with correction for multiple hypothesis testing using the Bonferroni method.*

Lines 871-873, Fig 5 legend: *Statistical significance was determined using Wilcoxon rank-sum tests with correction for multiple hypothesis testing using the Bonferroni method.*

Reviewer #2:

Remarks to the Author:

Sun et al set out to understand early events of Mtb infection with the main question been if resisters have any unique anti-bacterial functional programs relative to latently infected persons. A long outstanding question in the field of TB is understanding the phenomenon of resisters and whether failure to convert TST or IGRA has a biological association with Mtb clearance. They aimed to measure IFNg-independent T cell signatures specific to household exposure, and to determine genes selectively enriched among Mtb-specific T cells derived from resisters compared to controls. This will fill in knowledge gaps about whether resisters have actual protective immunity or failed to convert TST/IGRAs in addition to the known Epi data. They observed a role of Th17 cells and lower polyfunctionality scores

in TST-/IGRA relative to LTBI, in addition, they showed important information on T cell quality which is novel. They used other cohorts to confirm this and non-human primates. The manuscript is well-written, experiments were well described as well as the results and conclusions which were precise and concise. The study is important and would have a major impact on the field, however, some of the claims are not confirmed resulting in several limitations. I'm requesting for a major revision due to importance of the paper in the field.

We welcome the positive assessment of our work and the opportunity to address these important critiques.

Major limitations

The study is important and will have a major impact on the field, however these points should be addressed before the paper can be considered for publication in Nature Medicine.

1. The authors claim that the observed Th17 phenotype (a known correlate of protection both in human and animal studies); they show is reflective of bacterial burden. However, they have not proven this. I would recommend they measure bacterial burden in the study cohorts to confirm this claim. How do the bacterial burden correlate with the responses observed?

We appreciate this comment but note that, unlike active TB, there are no established methods in the field to quantify bacterial burdens in healthy asymptomatic populations (RSTRs and LTBI). Our overall thesis is that the T cell signatures that we observe in this study may reflect the final result (bacterial control) of a sequence of events that may have taken place nearly 9 years prior to sample collection. To address this, we extended our findings to collaborative studies of non-human primates, where quantitative bacterial burdens are known. As described above in the reply to Reviewer #1, Comment #15, we now include new data that show positive associations between IFN γ -independent and IL17-expressing CD4 T cells and protection from M.tb in non-human primates that have been vaccinated with intravenous BCG (revised Fig 7h and 7i). In addition, we extended our results to a longitudinal human cohort of adolescents with latent M.tb infection that is well-established in the field. In Fig. 7a-e, we show that M.tb-specific Th17 T cells are associated with lack of progression to TB disease. This finding is notable because it reveals that the protective signature is not limited to 'resisters' but may also be seen in a subset of individuals with latent TB, thus highlighting the importance of our findings across the spectrum of TB disease. Changes to the text have been detailed in response to Reviewer #1, Comment #2 above. We have also revised the title to be more conservative and emphasize the associative nature of our findings.

Line 2: T cell phenotypes in M. tuberculosis 'resisters' are associated with bacterial control

2. TB has a diverse set of antigens, the authors used only two antigens, ESAT-6 and CFP which are used in IGRAs. The study participants were classified based on IGRAs and TSTs, and persons who were IGRA negative were classified as resisters. IGRAs measure IFN γ responses from T cells, thus their observed results that the quality of IFN γ response in resisters is low is expected. Is the response observed after ESAT⁶ and CFP10 stimulation the same as in other TB antigens such as MTB300? I'm recommending that the authors test other TB antigens in the same way as ESAT-6/CFP10 to confirm their results.

This was a consistent critique from two of the three Reviewers. In response, we performed new experiments on RSTR (n=17) and LTBI (n=20) and analyzed T cell phenotypes with flow cytometry after stimulation with M.tb

lysate. The findings broadly confirmed our results using ESAT-6/CFP-10 stimulation and have been incorporated into revised Fig. 6. Please see response to Reviewer #1, Comment #5 above for full details.

3. The resistor/LTBI cohort used for targeted sequencing is small. Sample size should be increased to increase the statistical observations made in Figure 2 after index sorting and targeted sequencing.

The targeted sequencing in the discovery phase was meant to be hypothesis generating and followed up in the validation phase using independent methods (flow cytometry and Luminex) and non-overlapping samples (Fig. 1 schema). We have included new targeted transcriptional profiling data on ~16,000 cells derived RSTR (n=16) and LTBI (n=16) after M.tb-lysate stimulation. Please see detailed response to Reviewer #1, Comment #5 above. Overall, we found that the targeted transcriptional profiles were broadly similar among the two groups (new Supp Fig. 7a). Additionally, we have amended the text as follows:

In the Supplementary Figures PDF:

Supplementary Fig 7. M.tb lysate-specific CD4 T cells targeted transcriptomic profile and naïve phenotype with flow cytometry validation. (a) M.tb lysate-specific T cell subsets were visualized using dimensionality reduction (t-SNE) and stratified by group. Here cell subsets were clustered based on flow cytometry mean fluorescence intensities (MFI) and binarized read counts of profiled genes.

Lines 323-330: Compared to experiments with ESAT-6/CFP-10 stimulation, we examined a larger discovery cohort of RSTRs (n = 16) and LTBI (n = 16) using index sorting and targeted transcriptional profiling. Overall, we did not observe qualitative differences in CD4 T cell phenotypes between the two groups (Supplementary Fig. 7a). Nevertheless, we attempted to replicate our original findings in an independent set of samples (RSTRs n = 17, LTBI n = 20) with flow cytometry and Luminex (Fig. 1A, Supplementary Fig. 8, and Table 2). T cells

responsive to M.tb lysate stimulation that expressed an early differentiation phenotype were present at similar frequencies in RSTRs and LTBI (Supplementary Fig. 6b).

4. SELECT-Seq was used to characterize responses observed in the targeted sequencing, cells were sorted on TCRab and activation induced markers. I recommend showing data from the TCR observed in the study to provide more information on the diversity of the T cells involved in protection.

This is a reasonable request. We have now included the TCR sequences from ESAT-6 and CFP-10 stimulated cells as Supplementary Table 4.

Lines 193-194: *We observed clonal expansion of ESAT-6/CFP-10-specific T cells among both RSTRs and LTBI subjects (Fig. 2d, 2e, and Supplementary Table 4)*

Lines 968-970: ***Supplementary Table 4. TCR-sequencing of the ESAT-6/CFP-10-specific T cells. The TCR-sequencing data from 3 RSTR and 4 LTBI using the SELECT-seq approach was analyzed to calculate the clonal expanded T cells.***

5. The authors claim that the T cells from the resistors are more stem cell naïve-like based on expression of a few stem naïve genes. This has not been functionally validated and should be validate via flow using naïve like stem cell markers.

In response to this critique, we performed follow up experiments in which we sorted T_{SCM} cells defined by CD95 expression from RSTRs (n=12) and LTBI (n=12), stimulated with ESAT-6/CFP-10 or M.tb lysate and quantified T_{SCM} frequencies and proliferation as well as examining resulting memory T cell phenotypes. We found that antigen-specific T_{SCM} cells were present at a higher proportion in LTBI compared to RSTRs but proliferated equally in both groups. In addition, we show that T_{SCM} cells stimulated with M.tb lysate skew towards a central memory phenotype while those stimulated with ESAT-6/CFP-10 skew towards an effector memory phenotype. These results have been incorporated into new Supp. Figure 5. We have also modified the Results as follows:

Lines 237-242: *Following stimulation, live, activated (CD154+/CD69+), naïve-like (CCR7+CD45RA+CD45RO-) lymphocytes co-stained with CFSE were enriched via FACS and assessed for proliferation (Supplementary Fig. 5a, 5b, 5c, and Supplementary Table 2). At the time of sorting, we observed a significantly higher proportion of naïve-like T cells co-expressing the T_{SCM} marker CD95+ cells among LTBI compared to RSTR subjects regardless of stimulation (Supplementary Fig. 5d, M.tb lysate: p = 0.048; ESAT-6/CFP-10: p = 0.013).*

In the Supplementary Figures PDF:

Supplementary Fig. 5

Supplementary Fig. 5. RSTR T_{scm} cells proliferate similarly to LTBI T_{scm} cells in response to *M.tb* antigen. (a) RSTR ($n = 12$) or LTBI ($n = 12$) PBMC were thawed, rested for 2 hours, stained with $5 \mu\text{M}$ CFSE, and stimulated with *M.tb* lysate ($100 \mu\text{g}/\text{mL}$) or ESAT-6/CFP-10 ($1 \mu\text{g}/\text{mL}$) overnight. Activated, naive-like lymphocytes ($\text{CD}154^+/\text{CD}69^+/\text{CCR}7^+/\text{CD}45\text{RA}^+/\text{CD}45\text{RO}^-$) were sorted and cultured for seven days, and proliferation (CFSE MFI) was assessed via flow cytometry. (b-d) At T0, the proportion of T_{scm} cells ($\text{CD}95^+/\text{CCR}7^+/\text{CD}45\text{RA}^+/\text{CD}45\text{RO}^-$) was assessed in both RSTRS and LTBI subjects stimulated with *M.tb* lysate and ESAT-6/CFP-10. (c & e) After seven days of culture, the proportion of CFSE low $\text{CD}4^+/\text{CD}8^-/\text{CD}95^+$ was assessed. The phenotype of proliferated cells in response to (f) *M.tb* lysate and (g) ESAT-6/CFP-10 stimulation. Statistical significance in (d-g) was assessed using the Wilcoxon rank-sum test.

6. There are no mechanistic studies which are important in comprehending the underlying mechanisms for T cell protection in TB disease. Proliferative responses of clonally expanded cells are needed to show functional ability of the cells after stimulation.

Please see response to Comment #5 above. In addition, we have added the following text to the Results:

Lines 242-252: *After sorting, the cells were cultured for seven days. The proliferation of T_{SCM} cells, defined as $CD3^+$ and $CD95^+$ and originating from the $CCR7^+CD45RA^+CD45RO^-$ lymphocyte component, were assessed by flow cytometry^{40, 41}. The proportion of T_{SCM} cells that underwent proliferation in response to stimulation with *M.tb* lysate or ESAT-6/CFP-10 was approximately 18% and 20% for *M.tb* lysate and ESAT-6/CFP-10 stimulated cells and similar between RSTRs and LTBI (p=0.61 and p=0.67, respectively) (Supplementary Fig. 5c and 5e). Interestingly, while proliferative responses were similar, the resulting memory phenotypes of the proliferated cells were different. Sorted naïve-like T cells stimulated with *M.tb* lysate preferentially differentiated into central memory T cells, while stimulation with ESAT-6/CFP-10 resulted in a higher proportion of effector memory T cells (Supplementary Fig. 5f and 5g).*

Minor comment

7. The study was validated in a cohort of adolescents based on gene expression data, which is excellent. Cellular data is also available from the same cohort (Scriba et al., 2017, Plos Pathogens) and this should be analyzed to confirm the T cell responses observed in the resisters.

We appreciate the positive assessment of this aspect of our work as well as the excellent suggestion to complement our analysis with additional data. We attempted to obtain flow cytometry data from Scriba et al. PLOS Pathogens 2017 (PMID: 29145483), but were unsuccessful as their data have not been deposited in a public repository (e.g. ImmPort) and supplementary files detailing frequencies of various cell populations were not provided in the paper. However, we would emphasize that Fig. 7c partially addresses this point. We leveraged published data from Musvosvi et al. Nature Medicine 2023 (PMID: 36604540) in which the authors profiled *M.tb* specific cells from the same cohort described in the PLOS Pathogens study using the targeted transcriptional profiling approach we have used here. The findings show that Th17-like cells are associated with a lack of progression to TB. These data extend the findings we report on RSTRs to also include a sub-population of subjects with LTBI and suggest that the associations with protection extend across the clinical spectrum of TB (see response to Reviewer #1, Comment #8 for additional details).

[REDACTED]

[REDACTED]

[REDACTED]

[REDACTED]

[REDACTED]

Decision Letter, first revision:

Dear Dr. Seshadri,

Thank you for your response to the reviewers' comments on your manuscript "T cell phenotypes in M. tuberculosis 'resisters' are associated with bacterial control". We are happy to inform you that if you revise your manuscript appropriately in response to the referees' comments and our editorial requirements your manuscript should be publishable in Nature Immunology.

Please revise your manuscript to clarify the issues raised by the reviewers. At resubmission, please include a point-by-point response to the referees' comments, noting the pages and lines where the changes can be found in the revision. Please highlight the changes in the revised manuscript as well.

We are trying to improve the quality and transparency of methods and statistics reporting in our papers (please see our editorial in the May 2013 issue). Please update the Life Sciences Reporting Summary, and supplements if applicable, with any information relevant to any new experiments and upload it (as a Related Manuscript File) along with the files for your revision. If nothing in the checklist has changed, please upload the current version again.

TRANSPARENT PEER REVIEW

Nature Immunology offers a transparent peer review option for new original research manuscripts submitted from 1st December 2019. We encourage increased transparency in peer review by publishing the reviewer comments, author rebuttal letters and editorial decision letters if the authors agree. Such peer review material is made available as a supplementary peer review file. **Please state in the cover letter 'I wish to participate in transparent peer review' if you want to opt in, or 'I do not wish to participate in transparent peer review' if you don't.** Failure to state your preference will result in delays in accepting your manuscript for publication.

Please note: we allow redactions to authors' rebuttal and reviewer comments in the interest of confidentiality. If you are concerned about the release of confidential data, please let us know specifically what information you would like to have removed. Please note that we cannot incorporate redactions for any other reasons. Reviewer names will be published in the peer review files if the reviewer signed the comments to authors, or if reviewers explicitly agree to release their name. For more information, please refer to our FAQ page.

ORCID

Nature Immunology is committed to improving transparency in authorship. As part of our efforts in this direction, we are now requesting that all authors identified as 'corresponding author' on published papers create and link their Open Researcher and Contributor Identifier (ORCID) with their account on the Manuscript Tracking System (MTS), prior to acceptance. ORCID helps the scientific community achieve unambiguous attribution of all scholarly contributions. For more information please visit www.springernature.com/orcid.

Before resubmitting the final version of the manuscript, if you are listed as a corresponding author on the manuscript, please follow the steps below to link your account on our MTS with your ORCID. If you don't have an ORCID yet, you will be able to create one in minutes. If you are not listed as a

corresponding author, please ensure that the corresponding author(s) comply.

1. From the home page of the MTS click on '**Modify my Springer Nature account**' under '**General tasks**'.
2. In the '**Personal profile**' tab, click on '**ORCID Create/link an Open Researcher Contributor ID(ORCID)**'. This will re-direct you to the ORCID website.
- 3a. If you already have an ORCID account, enter your ORCID email and password and click on '**Authorize**' to link your ORCID with your account on the MTS.
- 3b. If you don't yet have an ORCID, you can easily create one by providing the required information and then click on '**Authorize**'. This will link your newly created ORCID with your account on the MTS.

IMPORTANT: All authors identified as 'corresponding authors' on the manuscript must follow these instructions. Non-corresponding authors do not have to link their ORCIDs, but please note that it will not be possible to add/modify ORCIDs at proof. Thus, if they wish to have their ORCID added to the paper, they must also follow the above procedure prior to acceptance.

To support ORCID's aims, we only allow a single ORCID identifier to be attached to one account. If you have any issues attaching an ORCID identifier to your Manuscript Tracking System account, please contact the Platform Support Helpdesk.

We hope that you will support this initiative and supply the required information. Should you have any query or comments, please do not hesitate to contact me.

Nature Immunology has now transitioned to a unified Rights Collection system which will allow our Author Services team to quickly and easily collect the rights and permissions required to publish your work. Once your paper is accepted, you will receive an email in approximately 10 business days providing you with a link to complete the grant of rights. If you choose to publish Open Access, our Author Services team will also be in touch at that time regarding any additional information that may be required to arrange payment for your article.

In recognition of the time and expertise our reviewers provide to Nature Immunology's editorial process, we would like to formally acknowledge their contribution to the external peer review of your manuscript entitled "T cell phenotypes in M. tuberculosis 'resisters' are associated with bacterial control". For those reviewers who give their assent, we will be publishing their names alongside the published article.

When you are ready to submit your revised manuscript, please use the URL below to submit the revised version:

We hope to receive your revised manuscript in 7 days, by 24th April. Please let us know if circumstances will delay submission beyond this time. If you have any questions please do not hesitate to contact me.

Sincerely,

Ioana Staicu, Ph.D.
Senior Editor
Nature Immunology

Tel: 212-726-9207
Fax: 212-696-9752
www.nature.com/ni

Reviewer #1 (Remarks to the Author):

Sun et al set out to determine the early events associated with Mtb control and/or clearance. The additional experiments including regulatory T cell measurements on flow cytometry and analyses of published datasets. They also included the Mtb H37RV lysate as an alternative TB antigen which showed similar results to the Esat6/CFP10 antigen confirming the results are not skewed due to IGRA results. They have made efforts to further analyze published datasets, such as the Scriba Plos Pathogens dataset. I think they have covered all the previous comments and have clarified ambiguous points in the manuscript.

The manuscript is well-written, the experiments are well described, and I think the results and conclusions match the data shown. I do not have any major comments and have only one minor comment.

Line 712: Typo, please delete "the" before whether.

Reviewer #2 (Remarks to the Author):

This is a well-written paper which provides extensive insight into resister control of infection in this highly unique cohort from Uganda. The South African cohort has been leveraged well to validate findings in terms of Th17-like function.

I previously reviewed this manuscript and have carefully checked the authors rebuttal to my comments. They have incorporated all my suggestions and have performed additional experiments to clarify some comments raised by myself and the other reviewers (namely in terms of the stimulation condition used). Responses were thorough and detailed and the manuscript has improved in clarity and scientific impact.

Two points for clarification. Firstly, in the discussion, lines 339-441. I think the addition of 'were' in "Mtb were exposed, but TST-negative..." may be incorrect as the sentence currently doesn't make sense. Secondly, in the methods for select seq you use anti-CD154 incubation for 30 min to prevent downregulation of CD154 but in the T cell proliferation assay you use anti-CD40. Why the difference?

Author Rebuttal, first revision:

Post-acceptance Rebuttal

Reviewer #1 (Remarks to the Author):

Sun et al set out to determine the early events associated with Mtb control and/or clearance. The additional experiments including regulatory T cell measurements on flow cytometry and analyses of published datasets. They also included the Mtb H37RV lysate as an alternative TB antigen which showed similar results to the Esat6/CFP10 antigen confirming the results are not skewed due to IGRA results. They have made efforts to further analyze published datasets, such as the Scriba Plos Pathogens dataset. I think they have covered all the previous comments and have clarified ambiguous points in the manuscript.

The manuscript is well-written, the experiments are well described, and I think the results and conclusions match the data shown. I do not have any major comments and have only one minor comment.

Line 712: Typo, please delete “the” before whether.

We thank reviewer 1 for the positive assessment of our work, their feedback, acknowledgement of validating datasets. We also thank them for the identification of this typo. The additional “the” on line 712 has been removed and now reads as the following:

Line 712 -713: “...we evaluated whether these modules were differentially expressed between progressors and non-progressors, in advance of sputum conversion.”

Reviewer #2 (Remarks to the Author):

This is a well-written paper which provides extensive insight into resistor control of infection in this highly unique cohort from Uganda. The South African cohort has been leveraged well to validate findings in terms of Th17-like function.

I previously reviewed this manuscript and have carefully checked the authors rebuttal to my comments. They have incorporated all my suggestions and have performed additional experiments to clarify some comments raised by myself and the other reviewers (namely in terms of the stimulation condition used). Responses were thorough and detailed and the manuscript has improved in clarity and scientific impact.

Two points for clarification. Firstly, in the discussion, lines 339-441. I think the addition of 'were' in "Mtb were exposed, but TST-negative..." may be incorrect as the sentence currently doesn't make sense. Secondly, in the methods for select seq you use anti-CD154 incubation for 30 min to prevent downregulation of CD154 but in the T cell proliferation assay you use anti-CD40. Why the difference?

We appreciate reviewers 2 approval of the changes made and new data included in the revised version of our manuscript. Additionally, we thank for the reviewer for pointing out these minor errors/typos: The following have been rectified:

- The additional “were” has been removed and now reads as the following:
Lines 439-441: “Monoclonal antibodies derived from *M.tb* exposed, but TST-negative healthcare workers can confer protection against *M.tb* challenge in mice”.
- Regarding the use of “anti-CD154” vs “anti-CD40”, anti-CD154 was a typo but refers to the use of the same reagent (anti-CD40). We have now standardized the verbiage used to anti-CD40 throughout the manuscript.

Decision Letter, second revision:

26th Apr 2024

Dear Dr. Seshadri,

Thank you for submitting your revised manuscript "T cell phenotypes in *M. tuberculosis* 'resisters' are associated with bacterial control" (NI-A36283B). We are happy to inform you that if you revise your manuscript appropriately according to our editorial requirements, your manuscript should be publishable in Nature Immunology.

I will now pre-edit the current version of your paper. We will also perform detailed checks on your paper and will send you a checklist detailing our editorial and formatting requirements in about two weeks. Please do not upload the final materials and make any revisions until you receive this additional information from us.

If you had not uploaded a Word file for the current version of the manuscript, we will need one before beginning the editing process; please email that to immunology@us.nature.com at your earliest convenience.

In the meantime, please deposit all omic and code data into public repositories so that the accession codes are readily available to be added in the revised manuscript. We cannot accept the paper without the codes. In addition, please check that the ORCID of ALL corresponding authors is linked to their Nature account, as this frequently causes delays at acceptance. Should you have any query or comments about ORCID, please do not hesitate to contact our editorial assistant at

immunology@us.nature.com.

Thank you again for your interest in Nature Immunology. Please do not hesitate to contact me if you have any questions.

Sincerely,

Ioana Staicu, Ph.D.
Senior Editor
Nature Immunology

Tel: 212-726-9207
Fax: 212-696-9752
www.nature.com/ni

Final Decision Letter:

Dear Dr. Seshadri,

I am delighted to accept your manuscript entitled "Specific CD4+ T cell phenotypes associate with bacterial control in people who 'resist' infection with Mycobacterium tuberculosis" for publication in an upcoming issue of Nature Immunology.

Over the next few weeks, your paper will be copyedited to ensure that it conforms to Nature Immunology style. Once your paper is typeset, you will receive an email with a link to choose the appropriate publishing options for your paper and our Author Services team will be in touch regarding any additional information that may be required.

Please note that *Nature Immunology* is a Transformative Journal (TJ). Authors may publish their research with us through the traditional subscription access route or make their paper immediately open access through payment of an article-processing charge (APC). Authors will not be required to

make a final decision about access to their article until it has been accepted. Find out more about Transformative Journals.

Authors may need to take specific actions to achieve compliance with funder and institutional open access mandates. If your research is supported by a funder that requires immediate open access (e.g. according to Plan S principles) then you should select the gold OA route, and we will direct you to the compliant route where possible. For authors selecting the subscription publication route, the journal's standard licensing terms will need to be accepted, including self-archiving policies. Those licensing terms will supersede any other terms that the author or any third party may assert apply to any version of the manuscript.

Your paper will be published online soon after we receive your corrections and will appear in print in the next available issue.

Also, if you have any spectacular or outstanding figures or graphics associated with your manuscript - though not necessarily included with your submission - we'd be delighted to consider them as candidates for our cover. Simply send an electronic version (accompanied by a hard copy) to us with a possible cover caption enclosed.

If you have not already done so, we strongly recommend that you upload the step-by-step protocols used in this manuscript to protocols.io. protocols.io is an open online resource that allows researchers to share their detailed experimental know-how. All uploaded protocols are made freely available and are assigned DOIs for ease of citation. Protocols can be linked to any publications in which they are used and will be linked to from your article. You can also establish a dedicated workspace to collect all your lab Protocols. By uploading your Protocols to protocols.io, you are enabling researchers to more readily reproduce or adapt the methodology you use, as well as increasing the visibility of your

protocols and papers. Upload your Protocols at <https://protocols.io>. Further information can be found at <https://www.protocols.io/help/publish-articles>.

Please note that we encourage the authors to self-archive their manuscript (the accepted version before copy editing) in their institutional repository, and in their funders' archives, six months after publication. Nature Portfolio recognizes the efforts of funding bodies to increase access of the research they fund, and strongly encourages authors to participate in such efforts. For information about our editorial policy, including license agreement and author copyright, please visit www.nature.com/ni/about/ed_policies/index.html

Sincerely,

Ioana Staicu, Ph.D.
Senior Editor
Nature Immunology

Tel: 212-726-9207
Fax: 212-696-9752
www.nature.com/ni